# Constant Rate Scheduling: A General Framework for Optimizing Diffusion Noise Schedule via Distributional Change

**Shuntaro Okada**,* **Kenji Doi, Ryota Yoshihashi, Hirokatsu Kataoka, Tomohiro Tanaka**
*LY Corporation, Japan*

**Reviewed on OpenReview:** *https://openreview.net/forum?id=Pjq6kdvMBj*

## Abstract

We propose a general framework for optimizing noise schedules in diffusion models, applicable to both training and sampling. Our method enforces a constant rate of change in the probability distribution of diffused data throughout the diffusion process, where the rate of change is quantified using a user-defined discrepancy measure. We introduce three such measures, which can be flexibly selected or combined depending on the domain and model architecture. While our framework is inspired by theoretical insights, we do not aim to provide a complete theoretical justification of how distributional change affects sample quality. Instead, we focus on establishing a general-purpose scheduling framework and validating its empirical effectiveness. Through extensive experiments, we demonstrate that our approach consistently improves the performance of both pixel-space and latent-space diffusion models, across various datasets, samplers, and a wide range of number of function evaluations from 5 to 250. In particular, when applied to both training and sampling schedules, our method achieves a state-of-the-art FID score of 2.03 on LSUN Horse 256×256, without compromising mode coverage.

## 1 Introduction

Diffusion models are a class of probabilistic generative models that have attracted significant attention due to their ability to generate high-quality samples across a variety of domains, including image generation (Saharia et al., 2022b; Nichol et al., 2022b; Podell et al., 2024a), image super-resolution (Saharia et al., 2023; Doi et al., 2024), video synthesis (Ho et al., 2022b;a), and audio generation (Kong et al., 2021; Popov et al., 2021).

Diffusion models were originally introduced in Sohl-Dickstein et al. (2015), and denoising diffusion probabilistic models (DDPMs) (Ho et al., 2020) demonstrated that diffusion models can achieve state-of-the-art performance in image generation. A typical diffusion model consists of two Markovian processes: a forward process and a reverse process. The forward process gradually adds Gaussian noise to the input data, transforming its distribution toward a Gaussian. The reverse process aims to reconstruct data from Gaussian noise by tracing the forward process in the reverse direction.

The noise levels applied at each step of these processes are treated as hyperparameters and are collectively referred to as the noise schedule. The noise schedule has a critical impact on both the quality and efficiency of training and sampling (Chen, 2023). Importantly, the schedules used during training and sampling do not have to be identical, and recent studies have shown that using different schedules for training and sampling can improve performance (Karras et al., 2022; Kingma & Gao, 2023; Hang et al., 2025). However, designing effective noise schedules remains a non-trivial and largely empirical process.

In this work, we propose a general framework for optimizing both training and sampling schedules by enforcing a constant rate of change in the probability distribution of diffused data throughout the diffusion

---

*Correspondence to: shuokada@lycorp.co.jp

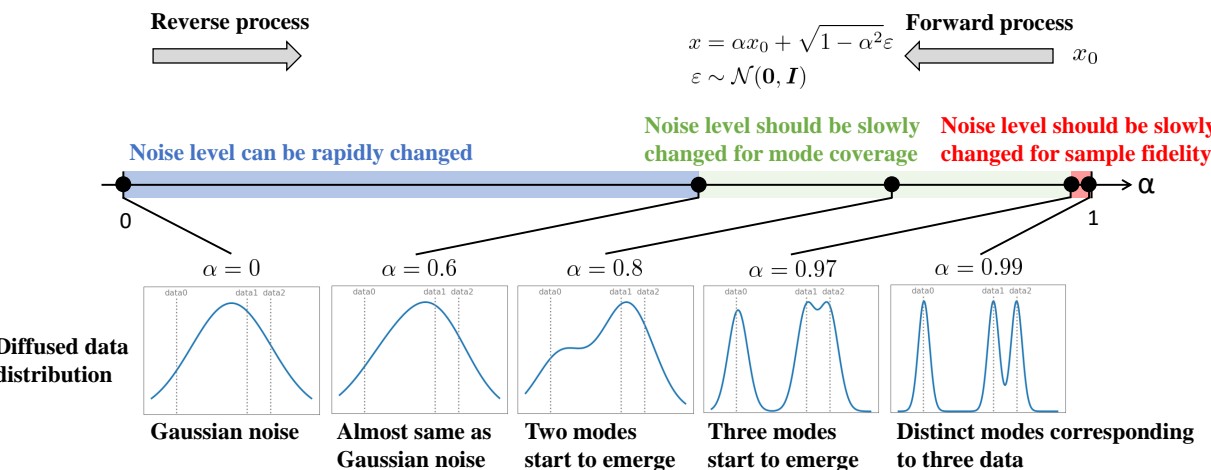

Figure 1: Toy example of diffused data distributions with three data points in one-dimensional data space. Probability distributions barely change when $\alpha \lesssim 0.6$, and we can rapidly change noise level. Three modes corresponding to data points emerge when $0.6 \lesssim \alpha \lesssim 0.97$, and noise level should be changed slowly for mode coverage. Three modes become distinct when $\alpha \gtrsim 0.97$, requiring careful control of noise level for sample fidelity.

process. We refer to this framework as constant rate scheduling (CRS). Since the reverse process is expected to accurately trace the forward process, the key motivation is to determine noise schedules that can enhance the traceability of the forward process .

Figure 1 presents a toy example of diffused data distributions with three data points in one-dimensional data space (see Appendix A for details). Here, $\alpha$ represents the decay rate of data at each step of the diffusion process. In this example, the diffusion process can be divided into three regions: 1) where distributions barely change and noise can be decreased rapidly ($\alpha \lesssim 0.6$), 2) where modes corresponding to each data point emerge and require fine control of noise to avoid mode dropping ($0.6 \lesssim \alpha \lesssim 0.97$), and 3) where mode peaks become sharp and noise must be reduced gradually for better fidelity ($\alpha \gtrsim 0.97$). This example illustrates that the distributional dynamics of the diffusion process are highly non-uniform. We hypothesize that the probability-distributional change represents the traceability of the diffusion process, and controlling the rate of distributional change can improve the quality and stability of the generative process.

Recent advances in generative modeling have also emphasized the importance of well-behaved distributional evolution. For example, flow matching (Lipman et al., 2023) and rectified flow transformers (Esser et al., 2024) formulate generation as a deterministic flow between data and noise distributions, highlighting that stable distributional trajectories can lead to improved sample quality. Although these approaches differ from diffusion models in their formulation, they share the high-level motivation of designing generative processes with controlled and stable distributional dynamics.

CRS offers a unified and flexible framework for designing noise schedules, with the following key properties: 1) it supports both training and sampling schedules, 2) it allows the use of arbitrary discrepancy measures to quantify distributional change, enabling domain-specific tailoring and independent schedules for training and sampling, and 3) many existing schedules can be viewed as special cases of CRS under different discrepancy measures.

Our contributions are summarized as follows:

1. We propose a general framework for optimizing noise schedules in diffusion models by enforcing a constant rate of distributional change throughout the diffusion process. Our framework supports both training and sampling schedules and allows the use of any user-specified discrepancy measure.

2. We introduce three practical discrepancy measures for quantifying distributional change, each capturing different aspects of the diffusion process. These measures can be flexibly selected or combined, depending on the domain or model type (e.g., pixel-space or latent-space diffusion models).

3. While theoretically motivated, this work does not aim to provide a full formal analysis of the link between distributional change and sample quality. Instead, we focus on empirically validating the effectiveness of CRS across diverse experimental settings. Through extensive experiments, we demonstrate that CRS consistently improves the performance of both pixel-space and latent-space diffusion models across various datasets, samplers, and numbers of function evaluations (NFEs), ranging from 5 to 250.

The remainder of this paper is organized as follows: Sec. 2 reviews related work. Sec. 3 briefly introduces DDPMs. Sec. 4 presents our proposed framework. Sec. 5 provides the experimental results. Sec. 6 discusses limitations, and Sec. 7 concludes the paper.

## 2 Related work

The performance of generative models is typically evaluated along three axes: sampling speed, fidelity, and mode coverage. Diffusion models are known to excel in terms of fidelity and mode coverage, but suffer from slow sampling speed due to the iterative nature of the reverse process (Xiao et al., 2022). To address this limitation, a wide range of acceleration strategies have been proposed. These approaches can be broadly categorized into five groups:

**Conditional generation:** This strategy improves sampling speed by reducing sample diversity through conditioning (e.g., class labels or text prompts) (Dhariwal & Nichol, 2021; Ho & Salimans, 2021; Rombach et al., 2022; Saharia et al., 2022a; Nichol et al., 2022a; Podell et al., 2024b), which guides the model to generate specific outputs and avoid unnecessary exploration.

**Dimensionality reduction:** Latent diffusion models (LDMs) (Rombach et al., 2022) employ pre-trained autoencoders to compress data representations, allowing the diffusion process to operate in a lower-dimensional latent space. This improves both training and sampling efficiency.

**Samplers:** These methods improve the update rule of diffused data for the reverse process. Denoising diffusion implicit models (DDIMs) (Song et al., 2021a) introduce a non-Markovian process to generalize DDPMs and propose a deterministic sampler. In continuous-time diffusion models, the forward and reverse processes are formulated as stochastic differential equations (SDEs) and ordinary differential equations (ODEs) (Song et al., 2021b). Numerous studies have developed efficient solvers tailored to these formulations (Jolicoeur-Martineau et al., 2021; Liu et al., 2022; Lu et al., 2022a;b; Zhang & Chen, 2023; Zhao et al., 2023; Zheng et al., 2023b; Zhou et al., 2024).

**Noise schedules:** The choice of noise schedule significantly affects training and sampling efficiency. Early schedules, such as linear (Ho et al., 2020) and cosine (Nichol & Dhariwal, 2021), were chosen heuristically. More recently, principled approaches for optimizing sampling schedules have been proposed (Chen et al., 2024; Sabour et al., 2024; Williams et al., 2024; Park et al., 2025; Tong et al., 2025). Among them, Align Your Steps (AYS) (Sabour et al., 2024) determines a noise schedule to minimize the discretization errors measured using Kullback-Leibler divergence. One of the discrepancy measures in our approach can be interpreted as a simplified variant of Kullback-Leibler divergence upper bound (KLUB) introduced in AYS, and thus CRS generalizes AYS and related methods as special cases.

In contrast, elucidating diffusion models (EDMs) (Karras et al., 2022) and variational diffusion models++ (VDM++) (Kingma & Gao, 2023) have proposed optimization of the training schedule. These works also highlight that using different schedules for training and sampling can improve performance. However, most prior work has focused predominantly on sampling schedules, and optimization of training schedules remains relatively underexplored. CRS is designed to optimize both schedules within a unified framework, supporting the use of arbitrary and even composite discrepancy measures. This flexibility allows practitioners to adapt scheduling strategies based on empirical behavior and task-specific requirements.

**Distillation:** Distillation-based approaches are another major direction for fast sampling (Luhman & Luhman, 2021; Salimans & Ho, 2022; Song et al., 2023; Yin et al., 2024b;a; Frankel et al., 2025). These methods compress multi-step diffusion into fewer steps. CRS is orthogonal to this line of work and can provide better teacher models for distillation, as it can improve training schedules.

## 3    Background

We briefly introduce DDPMs (Ho et al., 2020).

In the forward process, Gaussian noise is gradually added to the data through a sequence of timesteps, transforming the data distribution into an isotropic Gaussian. This process is typically modeled as a Markov chain that starts from a data sample $\boldsymbol{x}_0 \sim q(\boldsymbol{x}_0)$:

$$q(\boldsymbol{x}_{0:T}) = \left( \prod_{t=1}^{T} q(\boldsymbol{x}_t | \boldsymbol{x}_{t-1}) \right) q(\boldsymbol{x}_0), \quad \text{where} \quad q(\boldsymbol{x}_t | \boldsymbol{x}_{t-1}) = \mathcal{N}(\boldsymbol{x}_t; \beta_t \boldsymbol{x}_{t-1}, \delta_t^2 \boldsymbol{I}). \tag{1}$$

Here, $T$ denotes the number of timesteps, $\beta_t$ is a decay factor controlling the signal strength, $\delta_t$ represents the added noise level, and $q(\boldsymbol{x}_0)$ is the data distribution.

It is straightforward to show that $q(\boldsymbol{x}_t | \boldsymbol{x}_0)$ takes the following form:

$$q(\boldsymbol{x}_t | \boldsymbol{x}_0) = \mathcal{N}(\boldsymbol{x}_t; \alpha_t \boldsymbol{x}_0, \sigma_t^2 \boldsymbol{I}) \tag{2}$$

where the coefficients satisfy the recursive relations: $\alpha_t = \beta_t \alpha_{t-1}$ and $\sigma_t^2 = \delta_t^2 + \beta_t^2 \sigma_{t-1}^2$. This formulation allows sampling of $\boldsymbol{x}_t$ from $\boldsymbol{x}_0$ in a single step. In this work, we parameterize the forward process using $(\alpha_t, \sigma_t)$ instead of $(\beta_t, \delta_t)$, and adopt a variance-preserving formulation in which both $\alpha_t^2 + \sigma_t^2 = 1$ and $\beta_t^2 + \delta_t^2 = 1$ hold. The sequence $\{\alpha_t\}_{t=0}^{T}$ defines the noise schedule, which controls how rapidly the signal component of the data decays during diffusion. Our goal is to optimize this noise schedule to enable more effective and efficient training and sampling in diffusion models.

In the reverse process, data samples are generated from Gaussian noise by tracing the forward process in the reverse direction. This process is also modeled as a Markov chain and is defined as:

$$p_\theta(\boldsymbol{x}_{0:T}) = \left( \prod_{t=1}^{T} p_\theta(\boldsymbol{x}_{t-1} | \boldsymbol{x}_t) \right) p(\boldsymbol{x}_T), \quad \text{where} \quad p(\boldsymbol{x}_T) = \mathcal{N}(\boldsymbol{x}_T; \boldsymbol{0}, \boldsymbol{I}). \tag{3}$$

Here, $\theta$ denotes the learnable parameters of the diffusion model. In the limit of $T \to \infty$, the reverse transitions can be well approximated by the Gaussian distributions (Feller, 1949):

$$p_\theta(\boldsymbol{x}_{t-1} | \boldsymbol{x}_t) = \mathcal{N}\big(\boldsymbol{x}_{t-1}; \boldsymbol{\mu}_\theta(\boldsymbol{x}_t, t), \nu_t^2 \boldsymbol{I}\big). \tag{4}$$

DDPMs employ a noise prediction model $\boldsymbol{\varepsilon}_\theta(\boldsymbol{x}_t, t)$, which predicts the noise added at timestep $t$. The predicted mean $\boldsymbol{\mu}_\theta(\boldsymbol{x}, t)$ is given by:

$$\boldsymbol{\mu}_\theta(\boldsymbol{x}, t) = \frac{1}{\beta_t} \left( \boldsymbol{x}_t - \frac{\delta_t^2}{\sigma_t} \boldsymbol{\varepsilon}_\theta(\boldsymbol{x}, t) \right). \tag{5}$$

The noise prediction model is trained by maximizing a simplified version of the variational lower bound, which yields a practical and stable learning objective.

## 4    General Approach for Optimizing Noise Schedule

We explain our motivation and present CRS.

### 4.1 Motivation

As stated in Sec. 3, when $T$ is sufficiently large, it is a good approximation to assume that the reverse transitions $p_\theta(\boldsymbol{x}_{t-1}|\boldsymbol{x}_t)$ follow a Gaussian distribution, as expressed in Eq. (4). This implies that the distributional change between adjacent timesteps must be small enough to be well approximated by a Gaussian distribution. In diffusion models, sample quality depends on how faithfully the reverse process can trace the forward process. Large distributional changes can lead to poor approximation, resulting in artifacts or mode dropping. This observation motivates us to minimize the maximum distributional change during the diffusion process, in order to maintain the validity of Eq. (4) throughout the trajectory.

The importance of controlling distributional change can also be understood from the perspective of noise conditional score networks (NCSNs) (Song & Ermon, 2019; 2020; Song et al., 2021b), which are the conceptual predecessors of diffusion models. According to the manifold hypothesis, natural data typically lie on a low-dimensional manifold embedded in high-dimensional space. This leads to two main challenges when generating samples using only the score function $\nabla_{\boldsymbol{x}} \log q(\boldsymbol{x})$ from a randomly initialized point $\boldsymbol{x}_{\mathrm{rnd}}$ (Song & Ermon, 2019): first, $\boldsymbol{x}_{\mathrm{rnd}}$ is likely to fall in regions where $q(\boldsymbol{x}_{\mathrm{rnd}}) \ll 1$, making it difficult to estimate accurate gradients; second, even if the score is correctly estimated, it tends to be small in such regions, leading to slow mixing.

To address these challenges, NCSNs interpolate between the data distribution and a Gaussian distribution, whose support spans the entire space. If the distributional change between adjacent steps is kept small during this interpolation, each reverse step is likely to remain within regions of high probability density. This improves the reliability of local updates and enhances overall sample quality.

These theoretical motivations form the basis for our proposed framework: to design noise schedules that ensure a constant rate of change in the distribution of the diffused data across timesteps.

### 4.2 Constant Rate Scheduling

As described above, our goal is to optimize noise schedules by minimizing the maximum distributional change between adjacent timesteps in the diffusion process. Let $D(t, t + \Delta t)$ denote a discrepancy measure between the distributions of diffused data $q(\boldsymbol{x}_t)$ and $q(\boldsymbol{x}_{t+\Delta t})$. Our criterion can be formulated as:

$$\underset{\alpha(t)}{\mathrm{argmin}} \left( \max_t D(t, t + \Delta t) \right). \tag{6}$$

The optimal schedule $\alpha(t)$ must satisfy the following condition:

$$D(t, t + \Delta t) = \frac{D(t, t + \Delta t) - D(t, t)}{\Delta t} \Delta t \simeq \left. \frac{\partial D(t, t')}{\partial t'} \right|_{t'=t} \Delta t = \mathrm{const.}, \tag{7}$$

where we used $D(t, t) = 0$.

To explicitly reflect the dependence on the noise schedule, we rewrite $D(t, t')$ as $\tilde{D}(\alpha(t), \alpha(t')) \equiv \tilde{D}(\alpha, \alpha')$, where $\alpha(t)$ is the noise schedule to be optimized. This leads to:

$$D(t, t + \Delta t) \simeq \left. \frac{\partial D(t, t')}{\partial t'} \right|_{t'=t} \Delta t = v(\alpha) \frac{d\alpha(t)}{dt} \Delta t = \mathrm{const.}, \tag{8}$$

where $v(\alpha)$ represents the rate of change in the probability distribution of the diffused data.

Therefore, the noise schedule must satisfy $\frac{d\alpha(t)}{dt} \propto v(\alpha)^{-1}$. The proportionality constant is computed from the boundary conditions $\alpha(0) = \alpha_{\max}$ and $\alpha(1) = \alpha_{\min}$, yielding:

$$-\frac{d\alpha(t)}{dt} = Cv(\alpha)^{-\xi}, \quad \text{where} \quad C = \int_{\alpha_{\min}}^{\alpha_{\max}} v(\alpha)^\xi d\alpha. \tag{9}$$

Here, we introduce a hyperparameter $\xi > 0$ to control the dependence of $\alpha(t)$ on $v(\alpha)$. Larger values of $\xi$ allocate more timesteps to regions where $v(\alpha)$ is large, i.e., where the distribution changes rapidly.

The noise schedule optimization using CRS consists of the following steps:

1. Choose a discrepancy measure to quantify distributional change.

2. Compute $v(\alpha)$ using the selected discrepancy measure.

3. Solve Eq. (9) to obtain $\alpha(t)$.

4. Apply the resulting schedule as follows:

   - For training, use $\alpha(t)$ directly.
   - For sampling, discretize the schedule uniformly in time: $\{\alpha(t/T)|t = 0, 1, ..., T\}$ where $T$ is the number of sampling steps.

We propose three discrepancy measures for computing $v(\alpha)$, which are detailed in Section 4.3.

The derivation of Eqs. (7) and (8) relies on a first-order Taylor expansion with respect to $\Delta t$, which is accurate when the NFE is sufficiently large. However, when optimizing the sampling schedule at small NFEs, $\Delta t$ becomes large and the approximation error may no longer be negligible. Empirically, we observe that CRS with $\xi = 1$ remains effective for moderate NFEs, while the importance of tuning $\xi$ increases as the NFE decreases (see Table 27 in Appendix F.2). We attribute this increased sensitivity to the growing discretization error of the first-order expansion.

For small NFEs, the optimal value of $\xi$ tends to be greater than 1, resulting in more timesteps being allocated to regions where $v(\alpha)$ is large. Since discretization error is expected to be more pronounced in regions with larger $v(\alpha)$, it is intuitively reasonable that values of $\xi > 1$ become optimal when the NFE is small.

Although introducing $\xi$ partially compensates for the discretization error, it does not fully resolve the mismatch between the continuous-time formulation and the discrete-time implementation. A fully principled solution would require directly optimizing the sampling schedule in the discrete-time domain, without relying on the continuous-time assumption underlying Eq. (7). We consider this an important direction for future work.

Similar to NCSNs inspired by simulated annealing (Kirkpatrick et al., 1983), our formulation draws additional motivation from the adiabatic theorem (Morita & Nishimori, 2008) in quantum annealing (Kadowaki & Nishimori, 1998). Quantum annealing solves combinatorial optimization problems by interpolating between a target cost function and an initial one that has a trivial optimum. The optimization starts from the known optimum of the initial function and aims to trace the instantaneous optimums along the interpolation path. It is crucial to accurately trace the optimum of the instantaneous cost function, and the adiabatic theorem provides the traceability of the optimum at each step. Let $\alpha(t)$ be the weight of the target cost function and $1 - \alpha(t)$ be the weight of the initial cost function. In this setting, the inverse traceability is expressed in the form of $\frac{d\alpha(t)}{dt}v(\alpha)$, which takes the same form as Eq. (8). Prior work has shown that minimizing the maximum value of this inverse traceability yields the optimal $\alpha(t)$ (Roland & Cerf, 2002), that ensures consistent traceability throughout the interpolation. In CRS, we regard the rate of probability-distributional change as the inverse traceability and minimize its maximum value to derive an optimal noise schedule.

### 4.3 Discrepancy Measures

Any discrepancy measure is applicable to CRS. We introduce three discrepancy measures.

#### 4.3.1 Discrepancy measure based on FID

FID is a widely used metric for evaluating sample fidelity in the field of image generation. It computes the Fréchet distance (Heusel et al., 2017) between the feature distributions of real and generated images under a Gaussian assumption, where the features are extracted using the Inception-V3 network (Szegedy et al., 2016) trained on ImageNet classification.

When used as a discrepancy measure in our framework, $v(\alpha)$ tends to take large values in the region where sample fidelity significantly degrades (i.e., $\alpha \simeq 1$). As a result, CRS allocates more timesteps to this region, thereby improving sample fidelity.

However, a limitation of the FID-based measure is that it fails to accurately capture distributional changes in high-noise regions ($\alpha \simeq 0$), since the feature extractor is not trained on heavily diffused images and may not provide meaningful representations in that regime. Therefore, while the FID-based measure is effective for capturing fidelity-related changes in low-noise regions, it may benefit from being complemented by other discrepancy measures that can better capture high-noise dynamics.

Once the forward process is defined, $v(\alpha)$ can be computed by simulating the forward diffusion using the training dataset (see Algorithm 1 in Appendix C.1). Since this procedure is independent of model parameters, it only needs to be executed once prior to training the diffusion model.

The computation time of $v(\alpha)$ is proportional to the number of training images, which may be prohibitively expensive for large-scale training datasets. To address the scalability issue for large-scale datasets, we introduce a practical alternative in Sec. 5.7.

We refer to $v(\alpha)$ computed using FID as $v_{\text{FID}}(\alpha)$.

### 4.3.2 Discrepancy measure based on data prediction

We introduce a discrepancy measure that leverages a trained diffusion model as follows:

$$v_x(\alpha) = \sqrt{\left.\frac{\partial \overline{D_x^2}(\alpha, \alpha')}{\partial \alpha'}\right|_{\alpha'=\alpha}}, \quad \text{where} \quad \overline{D_x^2}(\alpha, \alpha') = \mathbb{E}_{\boldsymbol{x}_\alpha, \boldsymbol{x}_{\alpha'} \sim q(\boldsymbol{x}_\alpha, \boldsymbol{x}_{\alpha'})} \left[\left\|\boldsymbol{x}_\theta(\boldsymbol{x}_\alpha, \alpha) - \boldsymbol{x}_\theta(\boldsymbol{x}_{\alpha'}, \alpha')\right\|_2^2\right]. \quad (10)$$

Here, $q(\boldsymbol{x}_\alpha, \boldsymbol{x}_{\alpha'})$ is the joint distribution of diffused data at different noise levels, and $\boldsymbol{x}_\theta(\boldsymbol{x}, \alpha)$ denotes the data prediction, which is related to the noise prediction $\boldsymbol{\varepsilon}_\theta(\boldsymbol{x}, \alpha)$ via $\boldsymbol{x} = \alpha \boldsymbol{x}_\theta(\boldsymbol{x}, \alpha) + \sigma \boldsymbol{\varepsilon}_\theta(\boldsymbol{x}, \alpha)$.

This measure is designed to allocate more computational resources to regions where the data prediction changes rapidly with respect to $\alpha$. As shown in Appendix B, $v_x(\alpha)$ can be interpreted as a simplified variant of KLUB, introduced in AYS. Importantly, $v_x(\alpha)$ can be used not only to determine the sampling schedule after training, but also to adaptively optimize the training schedule during learning.

We further interpret $v_x(\alpha)$ as measuring the probability-distributional change of the diffused data. According to prior work (Ambrogioni, 2024), the data prediction $\boldsymbol{x}_\theta(\boldsymbol{x}, \alpha)$ can be viewed as a weighted average over training data points $\boldsymbol{x}_0$, where the weighting depends on the noise level $\alpha$. When $\alpha = 0$, the weights are approximately uniform across the dataset. As $\alpha \to 1$, only a few data points contribute significantly, indicating sharper and more specific representations. The change in these implicit weights reflects how sample trajectories diverge as noise decreases, and thus $v_x(\alpha)$ provides a proxy for the rate of distributional change.

The pseudocode for optimizing training and sampling schedules is provided in Appendix C.2. For sampling schedule optimization, the computation time of $v_x(\alpha)$ is negligible compared to the overall training time. In contrast, training schedule optimization requires updating $\overline{D_x^2}(\alpha, \alpha')$ during training as the model parameters evolve, which increases the total training time by approximately 20–30%. However, our experiments show that the schedule typically stabilizes early in training. Therefore, fixing the schedule after a few epochs or reducing the update frequency of $\overline{D_x^2}(\alpha, \alpha')$ may mitigate the computational cost without compromising performance. Developing a more efficient implementation remains an important direction for future work.

### 4.3.3 Discrepancy measure based on noise prediction

In the experiments presented in Sec. 5, we trained the noise prediction model. In this setting, it is also effective to use the noise predictions themselves to quantify distributional change, leading to the following definition:

$$v_\varepsilon(\alpha) = \sqrt{\left.\frac{\partial \overline{D_\varepsilon^2}(\alpha, \alpha')}{\partial \alpha'}\right|_{\alpha'=\alpha}}, \quad \text{where} \quad \overline{D_\varepsilon^2}(\alpha, \alpha') = \mathbb{E}_{\boldsymbol{x}_\alpha, \boldsymbol{x}_{\alpha'} \sim q(\boldsymbol{x}_\alpha, \boldsymbol{x}_{\alpha'})} \left[\left\|\boldsymbol{\varepsilon}_\theta(\boldsymbol{x}_\alpha, \alpha) - \boldsymbol{\varepsilon}_\theta(\boldsymbol{x}_{\alpha'}, \alpha')\right\|_2^2\right]. \quad (11)$$

We refer to CRS using $v_{\text{FID}}(\alpha)$, $v_x(\alpha)$, and $v_\varepsilon(\alpha)$ as CRS-$v_{\text{FID}}$, CRS-$v_x$, and CRS-$v_\varepsilon$, respectively, depending on the choice of discrepancy measure used to compute $v(\alpha)$.

### 4.4 Combining Multiple Discrepancy Measures

Each discrepancy measure has its own strengths and weaknesses. For example, CRS-$v_{\text{FID}}$ effectively captures distributional changes relevant to sample fidelity, but fails to represent changes accurately in high-noise regions ($\alpha \simeq 0$), as the feature extractor is not trained on diffused images. On the other hand, CRS-$v_x$ captures distributional change even in high-noise regions, but it is not necessarily optimal for improving sample fidelity. Therefore, it is considered effective to complementarily combining multiple discrepancy measures for optimizing noise schedules.

We introduce the following strategy for combining multiple discrepancy measures:

$$v(\alpha) = \sum_{m=1}^{M} w_m \tilde{v}_m(\alpha), \quad \text{where} \quad \tilde{v}_m(\alpha) = \frac{1}{C_m} v_m(\alpha)^{\xi_m} \quad \text{and} \quad C_m = \int_{\alpha_{\min}}^{\alpha_{\max}} v_m(\alpha)^{\xi_m}. \tag{12}$$

where $M$ is the number of discrepancy measures, $v_m(\alpha)$ is the rate of change computed using the $m$-th measure, $w_m$ is its corresponding weight (satisfying $\sum_{m=1}^{M} w_m = 1$), and $\xi_m$ controls the emphasis on $v_m(\alpha)$. Since the value ranges of different discrepancy measures can vary significantly, we normalize each $v_m(\alpha)$ as in Eq. (12) before combining them.

This formulation also enables conventional noise schedules to be interpreted and incorporated within the CRS framework. By converting a given noise schedule $\alpha(t)$ into its corresponding form of $v(\alpha)$, we can treat it as an implicit discrepancy measure. For example, if $\alpha(t) = \cos\left(\frac{\pi t}{2}\right)$, then:

$$v(\alpha) \propto -\left(\frac{d\alpha(t)}{dt}\right)^{-1} \propto \sin^{-1}\left(\frac{\pi t}{2}\right) = \frac{1}{\sqrt{1-\alpha^2}}. \tag{13}$$

This observation implies that any noise schedule can be viewed as the optimization result of CRS with an associated (possibly implicit) discrepancy measure, thus providing a unified perspective on the design of noise schedules.

While combining multiple discrepancy measures increases the flexibility of CRS, it also introduces additional hyperparameters. In our experiments, we found that tuning these hyperparameters is required only when optimizing sampling schedules in pixel-space diffusion models. In all other cases—training schedules for both pixel-space and latent-space models, and sampling schedules for latent-space models—CRS-$v_x$ with $\xi = 1$ consistently performed well and can serve as a default setting.

In the next section, we show that CRS-$v_x + v_{\text{FID}}$, which combines $v_x(\alpha)$ and $v_{\text{FID}}(\alpha)$, is effective for optimizing the sampling schedule of pixel-space diffusion models. CRS-$v_x + v_{\text{FID}}$ has four hyperparameters: $w_x$, $w_{\text{FID}}$, $\xi_x$, and $\xi_{\text{FID}}$. The procedure for tuning these hyperparameters is described in Appendix D, and the hyperparameter settings used in all experiments are summarized in Table 12. For configurations with NFE $\leq 50$, Table 12 shows that $w_x = w_{\text{FID}} = 0.5$ is optimal in most cases. These observations suggest that, in practical use, it is generally sufficient to tune only $\xi_x$ and $\xi_{\text{FID}}$.

## 5 Experiments

We experimentally demonstrate that CRS broadly improves the performance of diffusion models.

### 5.1 Experimental Settings

Below, we provide a brief overview of the components and settings used in our experiments.

**Dataset:** We use six image datasets: LSUN (Church, Bedroom, Horse) (Yu et al., 2016), ImageNet (Deng et al., 2009), FFHQ (Karras et al., 2019), and CIFAR10 (Krizhevsky, 2009). LSUN, FFHQ, and CIFAR10 are used for unconditional image generation, while ImageNet is used for class-conditional image generation.

**Noise Prediction Model:** We adopt the U-Net architecture introduced in the ablated diffusion model (ADM) (Dhariwal & Nichol, 2021) as the noise prediction model. Hyperparameter settings are listed in Table 13 (Appendix E).

**Training Schedule:** We compare four training noise schedules: linear (Ho et al., 2020), shifted cosine (Hoogeboom et al., 2023), the adaptive schedule proposed in VDM++ (Kingma & Gao, 2023), and our proposed CRS. The CRS hyperparameter $\xi$ is fixed to 1 in all experiments.

**Loss Function:** Except for the adaptive schedule in VDM++, we use the simplified loss function from DDPMs:

$$L = \frac{1}{2}\mathbb{E}_{\boldsymbol{x}_0\sim\mathcal{D},t\sim\mathcal{U}(0,1),\boldsymbol{\varepsilon}\sim\mathcal{N}(\boldsymbol{0},\boldsymbol{I})}\left[\left\|\boldsymbol{\varepsilon}_\theta(\boldsymbol{x},\alpha(t))-\boldsymbol{\varepsilon}\right\|_2^2\right], \tag{14}$$

where $\boldsymbol{x} = \alpha(t)\boldsymbol{x}_0 + \sqrt{1-\alpha(t)^2}\boldsymbol{\varepsilon}$, $\mathcal{D}$ denotes the training dataset, and $\alpha(t)$ is the training schedule.

When using the adaptive schedule from VDM++, the following loss function is used:

$$L = \frac{1}{2}\mathbb{E}_{\boldsymbol{x}_0\sim\mathcal{D},\lambda\sim p(\lambda),\boldsymbol{\varepsilon}\sim\mathcal{N}(\boldsymbol{0},\boldsymbol{I})}\left[\frac{\omega(\lambda)}{p(\lambda)}\left\|\boldsymbol{\varepsilon}_\theta\left(\boldsymbol{x},\sqrt{\text{sigmoid}(\lambda)}\right)-\boldsymbol{\varepsilon}\right\|_2^2\right], \tag{15}$$

where $\lambda = \log\left(\frac{\alpha^2}{\sigma^2}\right)$ is the log signal-to-noise ratio (log-SNR), $\omega(\lambda)$ is the weighting function, and $p(\lambda) = -\frac{d\lambda}{dt}$ denotes the noise schedule. We adopt the EDM monotonic function as $\omega(\lambda)$, and update $p(\lambda)$ at each iteration to approximately satisfy:

$$p(\lambda) \propto \mathbb{E}_{\boldsymbol{x}_0\sim\mathcal{D},\boldsymbol{\varepsilon}\sim\mathcal{N}(\boldsymbol{0},\boldsymbol{I})}\left[\omega(\lambda)\left\|\boldsymbol{\varepsilon}_\theta\left(\boldsymbol{x},\sqrt{\text{sigmoid}(\lambda)}\right)-\boldsymbol{\varepsilon}\right\|_2^2\right]. \tag{16}$$

**Sampling Schedule:** We evaluate four sampling schedules: linear, shifted cosine, EDM (Karras et al., 2022), and CRS. For pixel-space diffusion models, we additionally evaluate CRS-$v_x + v_{\text{FID}}$, which combines $v_x(\alpha)$ and $v_{\text{FID}}(\alpha)$ to leverage their complementary strengths, as discussed in Sec. 5.5. Unless otherwise stated, the hyperparameter $\xi$ for CRS-$v_x$, CRS-$v_\varepsilon$, and CRS-$v_{\text{FID}}$ is fixed to $\xi = 1$. For the hyperparameter tuning procedure of CRS–$v_x + v_{\text{FID}}$ and the exact values used in our experiments, please refer to Appendix D. Following prior work (Karras et al., 2022; Kingma & Gao, 2023), we allow training and sampling schedules to differ. We thus evaluate all combinations of training and sampling schedules. Details of the conventional schedules are summarized in Table 16 (Appendix E).

**Sampling Methods:** To assess the generality across samplers, we employ two stochastic samplers and three deterministic samplers. The stochastic samplers include Stochastic DDIM ($\eta = 1$) (Song et al., 2021a) and SDE-DPM-Solver++(2M) (Lu et al., 2022b). The deterministic samplers include DDIM (Song et al., 2021a), DPM-Solver++(2M) (Lu et al., 2022b), and pseudo numerical methods (PNDM) (Liu et al., 2022). At low NFE regimes, we also evaluate improved PNDM (iPNDM) (Zhang & Chen, 2023) and the unified predictor-corrector framework (UniPC) (Zhao et al., 2023).

**Evaluation Metrics:** We use four evaluation metrics: FID (Heusel et al., 2017), spatial FID (sFID) (Nash et al., 2021), and improved precision and recall (Sajjadi et al., 2018; Kynkäänniemi et al., 2019). FID, sFID, and precision measure sample fidelity, while recall quantifies mode coverage.

## 5.2 Results on Pixel-Space Diffusion Models

We trained continuous-time diffusion models on LSUN Horse 256×256 and LSUN Bedroom 256×256 for unconditional image generation. Detailed experimental results are provided in Appendix F.1.

**LSUN Horse 256×256:** The evaluation results are summarized in Table 1. Rows shaded in gray represent results obtained under different training schedules. Although sample fidelity and mode coverage are generally considered to be in a trade-off relationship, CRS-$v_x$ achieved the highest scores across all evaluation metrics. This indicates that CRS-$v_x$ does not merely balance fidelity and diversity but leads to an overall improvement in sample quality.

The unshaded rows represent the dependence on sampling schedules. CRS-$v_x + v_{\text{FID}}$ outperformed all baselines across all metrics, indicating a fundamental improvement in generation quality. Notably, our model achieved an FID score of 2.03, surpassing the previous state-of-the-art score of 2.11 reported in Kumari et al. (2022).

Table 1: Performance of pixel-space diffusion models on LSUN Horse 256×256 with NFE = 250, using SDE-DPM-Solver++(2M) as the sampler. Gray-shaded rows indicate variations in the training schedule, while unshaded rows correspond to different sampling schedules. Bold values highlight the best performance for each metric under varying training or sampling schedules.

| Training schedule | Sampling schedule | Metrics | | | |
|---|---|---|---|---|---|
| | | FID ↓ | sFID ↓ | Precision ↑ | Recall ↑ |
| Linear | Linear | 2.90 | 6.82 | 0.66 | **0.56** |
| Shifted cosine | | 2.95 | 6.43 | **0.67** | **0.56** |
| VDM++ | | 3.82 | 7.40 | 0.66 | 0.54 |
| CRS-$v_\varepsilon$ | | 2.91 | 6.56 | 0.66 | **0.56** |
| CRS-$v_x$ | | **2.73** | **6.40** | **0.67** | **0.56** |
| CRS-$v_x$ | Linear | 2.73 | 6.40 | 0.67 | **0.56** |
| | Shifted cosine | 3.13 | 7.09 | 0.67 | 0.54 |
| | EDM | 2.09 | 6.16 | **0.69** | **0.56** |
| | CRS-$v_\varepsilon$ | 2.87 | 6.71 | 0.66 | 0.56 |
| | CRS-$v_x$ | 5.46 | 8.34 | 0.62 | 0.52 |
| | CRS-$v_x + v_{\text{FID}}$ | **2.03** | **6.06** | **0.69** | **0.56** |

Table 2: FID scores for LSUN Bedroom 256×256 in pixel-space diffusion models under low NFE conditions. Our results are compared with prior work from Chen et al. (2024). Bold entries indicate the best FID achieved at each NFE setting.

| Model | Sampler | Sampling schedule | NFE = 5 | NFE = 10 |
|---|---|---|---|---|
| EDM | UniPC (Zhao et al., 2023) | EDM | 23.34 | 5.75 |
| | AMED-Solver (Zhou et al., 2024) | | – | 5.38 |
| | iPNDM (Zhang & Chen, 2023) | GITS (Chen et al., 2024) | 15.85 | 5.28 |
| Ours | DPM-Solver++(2M) | EDM | 23.72 | 4.73 |
| | | CRS-$v_x + v_{\text{FID}}$ | **14.02** | 4.88 |
| | UniPC | EDM | 82.52 | 4.94 |
| | | CRS-$v_x + v_{\text{FID}}$ | 21.42 | **3.30** |

**LSUN Bedroom 256×256:** We further validated the effectiveness of CRS in low-NFE regimes by comparing it with strong baselines reported in GITS (Chen et al., 2024). We trained continuous-time diffusion models with CRS-$v_x$ used for optimizing the training schedule. Table 2 lists the results. At NFE = 5 and NFE = 10, the best FID scores were achieved by using CRS-$v_x$ and CRS-$v_x + v_{\text{FID}}$ for optimizing training and sampling schedules, respectively.

As shown in Table 20 in Appendix F.1, we also tuned the $\rho$ parameter in the EDM sampling schedule. Even with the optimal $\rho$ setting, CRS-$v_x + v_{\text{FID}}$ consistently outperformed EDM.

### 5.3 Results on Latent-Space Diffusion Models

We trained latent-space diffusion models using the pre-trained autoencoder from LDMs (Rombach et al., 2022). In our experiments, we adopted the VQ-regularized variant for all datasets, which is the configuration primarily used in the publicly available pretrained models on GitHub (`https://github.com/CompVis/latent-diffusion`). Although a downsampling factor of $f = 8$ is also commonly used, we set $f = 4$ (VQ-f4), as it is expected to yield better FID scores according to Table 10 in the LDMs paper. LSUN Church and LSUN Bedroom were used for unconditional image generation, and ImageNet was used for class-conditional image generation. All images were resized to 256×256 and encoded into 64×64 latent representations. Here, we present the evaluation results on LSUN Church 256×256. Detailed results are provided in Appendix F.2.

**LSUN Church 256×256:** Table 3 summarizes the results. Rows shaded in gray show the effect of different training schedules, where CRS-$v_x$ achieved the best performance on all metrics except sFID. Unshaded rows show the dependence on the sampling schedule, and again, CRS-$v_x$ achieved the best performance on all metrics except precision.

Table 3: Performance of latent-space diffusion models on LSUN Church 256×256 with NFE = 30, using DPM-Solver++(2M) as the sampler. Gray-shaded rows indicate variations in the training schedule, while unshaded rows correspond to different sampling schedules. Bold values highlight the best performance for each metric under varying training or sampling schedules.

| Training schedule | Sampling schedule | Metrics | | | |
|---|---|---|---|---|---|
| | | FID ↓ | sFID ↓ | Precision ↑ | Recall ↑ |
| Linear | Linear | 3.82 | 10.28 | **0.60** | 0.58 |
| VDM++ | | 4.00 | **10.07** | 0.58 | **0.59** |
| CRS-$v_{\text{FID}}$ | | 3.78 | 10.51 | **0.60** | 0.58 |
| CRS-$v_\varepsilon$ | | 3.87 | 10.30 | **0.60** | **0.59** |
| CRS-$v_x$ | | **3.69** | 10.23 | **0.60** | **0.59** |
| CRS-$v_x$ | Linear | 3.69 | 10.23 | **0.60** | 0.59 |
| | EDM | 4.31 | 11.31 | **0.60** | 0.56 |
| | CRS-$v_{\text{FID}}$ | 3.98 | 9.61 | 0.58 | **0.60** |
| | CRS-$v_\varepsilon$ | 3.63 | 9.91 | 0.59 | **0.60** |
| | CRS-$v_x$ | **3.59** | **9.51** | 0.59 | **0.60** |

Table 4: FID scores on CIFAR10 32×32 and FFHQ 64×64 evaluated using the pretrained model provided by EDM (Karras et al., 2022). Bold numbers indicate the best sampling schedule for each combination of dataset, sampler, and NFE. For reference, the FID scores of AYS (Sabour et al., 2024) and LD3 (Tong et al., 2025) are taken directly from their original papers.

| Sampler | Sampling schedule | CIFAR10 32×32 | | FFHQ 64×64 | |
|---|---|---|---|---|---|
| | | NFE = 10 | NFE = 20 | NFE = 10 | NFE = 20 |
| DPM-Solver++(2M) | AYS | **2.98** | **2.10** | 5.43 | 3.29 |
| | GITS | 4.03 | 2.32 | 5.51 | 3.81 |
| | LD3 | 3.38 | 2.36 | **3.98** | **2.89** |
| | CRS-$v_x + v_{\text{FID}}$ | 3.92 | 2.27 | 5.35 | 3.10 |
| UniPC(3M) | GITS | 3.50 | **1.99** | 4.52 | 2.71 |
| | LD3 | 2.84 | – | **3.27** | – |
| | CRS-$v_x + v_{\text{FID}}$ | **2.52** | **1.99** | 5.23 | **2.64** |
| iPNDM(3M) | GITS | 2.73 | **2.08** | 3.96 | 3.09 |
| | LD3 | **2.38** | – | **3.25** | – |
| | CRS-$v_x + v_{\text{FID}}$ | 2.87 | 2.11 | 4.07 | **2.75** |
| iPNDM(4M) | GITS | 2.49 | **2.02** | 3.62 | 3.00 |
| | CRS-$v_x + v_{\text{FID}}$ | **2.46** | **2.02** | **3.45** | **2.53** |

These results indicate that CRS is also effective in latent-space diffusion models. Although CRS-$v_x$ performs best in most settings, we observe that depending on the sampler and NFEs, CRS-$v_\varepsilon$ can be preferable for sampling.

### 5.4 Comparison with Recent Sampling Schedules

Using the pretrained models released with EDM (Karras et al., 2022), we compare the performance of CRS as a sampling schedule against AYS (Sabour et al., 2024), GITS (Chen et al., 2024), and learning to discretize denoising diffusion ODEs (LD3) (Tong et al., 2025). All evaluations in this subsection are conducted using the official repository provided in the GITS paper, ensuring full consistency with their experimental protocol.

It is important to note that EDM adopts a variance-exploding formulation, whereas all of our previous experiments have been conducted under the variance-preserving setting. However, CRS can still be applied to variance-exploding processes by converting between $\alpha$ and the EDM noise level $\sigma_{\text{EDM}}$ via the log-SNR, as shown below:

$$\log \text{SNR} = \log\left(\frac{\alpha^2}{1-\alpha^2}\right) = \log\left(\frac{1}{\sigma_{\text{EDM}}^2}\right) \quad \Rightarrow \quad \alpha = \frac{1}{\sqrt{1+\sigma_{\text{EDM}}^2}}. \tag{17}$$

We also note that, whereas all earlier experiments in this paper used the noise-prediction parameterization, EDM adopts the data-prediction parameterization with preconditioning.

The evaluation results are presented in Table 4. The optimal sampling schedule varies across datasets, samplers, and NFE configurations. On average, LD3 achieves the highest performance, while CRS consistently outperforms GITS and can therefore be regarded as the second-best sampling schedule overall. Although CRS underperforms LD3 as a sampling schedule, a key advantage of CRS is that it also provides a unified framework for optimizing the training schedule, which LD3 does not address. Further improving the performance of CRS will likely require identifying more effective discrepancy measures, and exploring such alternatives is an important direction for future work.

For completeness and fairness, we note that CRS-$v_x + v_{\mathrm{FID}}$ requires both the computation of $v_{\mathrm{FID}}(\alpha)$ and hyperparameter tuning, and the time needed to optimize the sampling schedule is substantially larger than that of GITS or LD3. However, sampling-schedule optimization is performed only once after training the diffusion model. Therefore, as long as the tuning cost remains small relative to the total training time, we believe that performance under the best hyperparameter configuration is the most meaningful criterion for comparison.

## 5.5 Comparison of Noise Schedules

Figure 2 shows examples of CRS-based noise schedules in pixel-space diffusion models, computed using different discrepancy measures. For CRS-$v_x + v_{\mathrm{FID}}$, the hyperparameters were set to $w_x = w_{\mathrm{FID}} = 0.5$ and $\xi_x = \xi_{\mathrm{FID}} = 1.0$.

Since $v_{\mathrm{FID}}(\alpha)$ tends to be large in regions where sample fidelity degrades (i.e., near $\alpha \simeq 1$), CRS-$v_{\mathrm{FID}}$ allocates more timesteps to these regions. In contrast, CRS-$v_x$ allocates more timesteps to regions with smaller $\alpha$. This is due to the formulation of the data prediction $\boldsymbol{x}_\theta(\boldsymbol{x}, \alpha) = \frac{\boldsymbol{x} - \sigma \boldsymbol{\varepsilon}_\theta(\boldsymbol{x}, \alpha)}{\alpha}$, where $\alpha$ appears in the denominator, making the data prediction more sensitive to small changes in $\alpha$ when $\alpha$ is small. Thus, CRS-$v_{\mathrm{FID}}$ and CRS-$v_x$ exhibit complementary behaviors, and combining both yields an effective and balanced sampling schedule.

Furthermore, we observed that the noise schedule derived from CRS-$v_\varepsilon$ can be approximately reproduced by appropriately tuning the hyperparameters of CRS-$v_x + v_{\mathrm{FID}}$. Therefore, in practice, $v_x(\alpha)$ and $v_{\mathrm{FID}}(\alpha)$ serve as the two principal components for constructing robust sampling schedules.

Figure 3 illustrates CRS-based noise schedules for latent-space diffusion models. The optimized schedules differ notably between pixel-space and latent-space models. In latent-space diffusion, fewer timesteps are allocated to the region where $\alpha \simeq 1$. This difference can be attributed to the nature of latent-space diffusion models: perceptual expression is largely reconstructed by the autoencoder, and the autoencoder is robust to minor deviations in the generated latent variables. Consequently, precise control near $\alpha \simeq 1$ is less critical. In contrast, pixel-space models directly generate the final image, so the schedule must allocate sufficient timesteps near $\alpha \simeq 1$ to preserve high-fidelity details.

Because all discrepancy measures result in similar noise schedules for latent-space models, we did not evaluate combinations of multiple discrepancy measures.

## 5.6 Computational Overhead of CRS

CRS requires computing $v(\alpha)$. Below, we report the computational overhead in scenarios where the cost is expected to be most significant.

**Cost of computing** $v_{\mathrm{FID}}(\alpha)$**:** Although $v_{\mathrm{FID}}(\alpha)$ needs to be computed only once before training the diffusion model, its computation requires processing the entire training set to compute feature statistics, and therefore the cost scales linearly with the number of images.

Table 5 reports both the cost relative to a single training epoch and the cost relative to the total training time. Because the computation of $v_{\mathrm{FID}}(\alpha)$ and the duration of one training epoch both scale proportionally with dataset size, the cost relative to one training epoch remains nearly constant across datasets for both

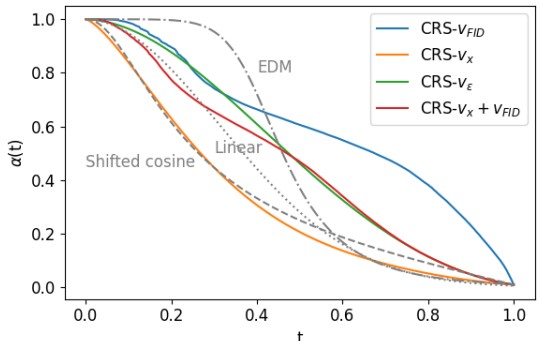 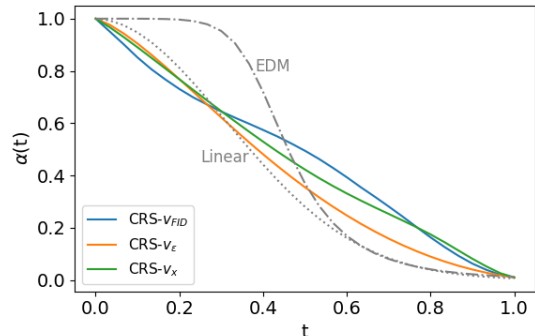

Figure 2: Noise schedules computed using different discrepancy measures on LSUN Horse 256×256 in the pixel-space diffusion model.

Figure 3: Noise schedules computed using different discrepancy measures on LSUN Church 256×256 in the latent-space diffusion model.

Table 5: Wall-clock computational cost of computing $v_{\mathrm{FID}}(\alpha)$. We report both the cost relative to one training epoch and the cost relative to the total training time for each diffusion setting.

| Diffusion type | Dataset | Computational cost | |
| --- | --- | --- | --- |
| | | Relative to one training epoch | Relative to total training time |
| Pixel space | LSUN Horse | ∼5 epochs | ∼10% |
| | LSUN Bedroom | ∼5 epochs | ∼10% |
| Latent space | LSUN Church | ∼14 epochs | ∼1.4% |
| | LSUN Bedroom | ∼14 epochs | ∼14% |
| | ImageNet | ∼14 epochs | ∼2.8% |

Table 6: Comparison of wall-clock training time when using linear and CRS-$v_x$ as the training schedule. CRS-$v_x$ requires evaluating $\overline{D_x^2}(\alpha, \alpha')$ during training, resulting in a 20—30% increase in overall training time.

| Diffusion type | Dataset | Linear | CRS-$v_x$ | Increase |
| --- | --- | --- | --- | --- |
| Pixel space | LSUN Horse | 170 hours | 215 hours | +26.5% |
| Latent space | LSUN Church | 85 hours | 105 hours | +23.5% |
| | LSUN Bedroom | 190 hours | 245 hours | +28.9% |
| | ImageNet | 430 hours | 525 hours | +22.1% |

pixel-space and latent-space diffusion models. The relative cost appears larger for latent-space diffusion models simply because one latent-space epoch is much faster than one pixel-space epoch. When compared to the total training time, the overhead of computing $v_{\mathrm{FID}}(\alpha)$ remains below 15% even in the worst case.

**Overhead of computing $v_x(\alpha)$ during training:** Table 6 compares the training time when using the linear versus CRS-$v_x$ as the training schedule. Because CRS-$v_x$ requires evaluating $\overline{D_x^2}(\alpha, \alpha')$ during training, the overall training time increases by approximately 20–30%. Nevertheless, CRS-$v_x$ typically accelerates the convergence of FID scores during training (see Figs. 12 and 13 in Appendix F for details), which can largely compensate for the per-epoch overhead.

### 5.7 A Practical Alternative to FID-Based Scheduling

In pixel-space diffusion models, CRS-$v_x + v_{\mathrm{FID}}$ consistently achieves the best performance as a sampling schedule. However, for large-scale training datasets, computing $v_{\mathrm{FID}}(\alpha)$ becomes prohibitively expensive.

As shown in the subsection 5.5, CRS-$v_{\mathrm{FID}}$ allocates a large number of timesteps to regions near $\alpha \simeq 1$, resulting in a schedule shape similar to that of the cosine schedule. Motivated by this observation, we evaluate a lightweight alternative: CRS-$v_x + v_{\cos}$, which combines CRS-$v_x$ with the cosine schedule (see Eq. 13 for $v(\alpha)$ of the cosine schedule).

Table 7: Evaluation of CRS-$v_x + v_{\cos}$ as a practical alternative to CRS-$v_x + v_{\text{FID}}$ for sampling in pixel-space diffusion models on LSUN Bedroom 256×256. Despite not requiring the computation of $v_{\text{FID}}(\alpha)$, CRS-$v_x + v_{\cos}$ achieves similar FID scores.

| Sampler | Sampling Schedule | NFE = 5 | NFE = 10 |
|---|---|---|---|
| DPM-Solver++(2M) | CRS-$v_x + v_{\text{FID}}$ | 14.02 | 4.88 |
| | CRS-$v_x + v_{\cos}$ | 14.10 | 4.46 |
| UniPC | CRS-$v_x + v_{\text{FID}}$ | 21.42 | 3.30 |
| | CRS-$v_x + v_{\cos}$ | 20.77 | 3.30 |

Table 7 compares the FID scores of CRS-$v_x + v_{\text{FID}}$ and CRS-$v_x + v_{\cos}$ on LSUN Bedroom 256×256. In all settings, CRS-$v_x + v_{\cos}$ achieves FID scores comparable to CRS-$v_x + v_{\text{FID}}$, demonstrating that CRS-$v_x + v_{\cos}$ is a practical and effective alternative when $v_{\text{FID}}(\alpha)$ is costly to compute.

## 6 Limitations

This paper focuses on developing CRS and empirically validating its effectiveness under fundamental experimental settings. While the results are promising, there are several directions for future work to further enhance the generality, applicability, and performance of CRS. Below, we summarize six key limitations and avenues for future work.

**(1) Theoretical analysis.** The goal of this work is to provide a theoretically motivated and empirically effective framework capable of incorporating a wide range of discrepancy measures. However, we do not provide a theoretical guarantee that minimizing the maximum discrepancy necessarily leads to improved sample quality. Establishing a formal connection between distributional change and sample quality remains an important direction for future research.

**(2) Evaluation on more complex tasks.** Our experiments focus on fundamental settings—unconditional and class-conditional image generation. We did not evaluate CRS in more complex practical scenarios such as text-conditional generation or classifier-free guidance (CFG). CFG is known to improve FID at the cost of reduced recall, and an adaptive noise schedule may help mitigate this trade-off. Evaluating CRS together with CFG or other conditioning mechanisms could further reveal the benefits of schedule optimization.

**(3) Applicability beyond image generation.** Although our experiments are limited to image generation, CRS is derived from general diffusion-model principles and is therefore expected to generalize to other domains. In this work, $v_{\text{FID}}(\alpha)$ proved effective as a sampling discrepancy measure for pixel-space diffusion models; however, it is specific to image generation. Applying CRS to non-image modalities will require identifying discrepancy measures that reflect the characteristics of each target domain.

**(4) Improving discrepancy measures.** As a sampling schedule optimizer, CRS currently performs slightly worse than LD3, suggesting that further improvements may be achieved by designing better discrepancy measures. A promising direction is to build on advances in schedule-optimization theory such as AYS. As shown in Appendix B, CRS-$v_x$ can be interpreted as employing a simplified variant of the KLUB introduced in AYS. Thus, future theoretical progress in schedule optimization may naturally lead to new discrepancy measures compatible with CRS.

With respect to image-specific metrics, although our experiments primarily relied on FID, many feature-based perceptual metrics have been proposed for image generation. For example, LPIPS (Zhang et al., 2018) is a strong candidate for capturing visually meaningful differences. Since FID and LPIPS depend on feature extractors, a key challenge is designing robust feature representations for noised data, which are essential for accurately assessing distributional change. Self-supervised learning on noised samples is a promising direction, as it requires no labels and can be made noise-level aware (e.g., by predicting or conditioning on $\alpha$). Such learned embeddings may lead to more expressive and domain-adaptive discrepancy measures.

**(5) Generality across prediction parameterizations and architectures.** Our experiments primarily evaluate CRS under the noise-prediction parameterization. However, the data- and velocity-prediction formulations are also widely used in recent diffusion models, and the generality of CRS under these parameterizations has not been thoroughly assessed.

Furthermore, modern diffusion architectures such as Diffusion Transformers (Peebles & Xie, 2023) were not evaluated. Because CRS is grounded in the dynamics of distributional evolution rather than in model architecture, we expect architectural dependence to be limited, but empirical validation remains an important direction for future work.

**(6) Scalability to dataset size and image resolution.** Although CRS-$v_x + v_{\mathrm{FID}}$ is effective for optimizing sampling schedules in pixel-space diffusion models, it requires both the computation of $v_{\mathrm{FID}}(\alpha)$ and hyperparameter tuning. In particular, the computation of $v_{\mathrm{FID}}(\alpha)$ scales linearly with the number of training images, making this variant impractical for large-scale datasets. Although Section 5.7 demonstrates that CRS-$v_x + v_{\cos}$ can serve as a practical alternative, fundamentally improving scalability will require developing new discrepancy measures that are both computationally efficient and robust to noise-level variation.

Furthermore, all of our experiments were conducted at a resolution of 256×256. Validating whether CRS remains effective for higher-resolution image generation, and analyzing how the optimal noise schedule depends on image size, are compelling directions for future research.

## 7 Conclusion

We proposed a general framework called CRS for optimizing noise schedules in diffusion models. CRS enforces a constant rate of distributional change throughout the diffusion process and provides a unified perspective for designing both training and sampling schedules. Our framework supports arbitrary user-specified discrepancy measures, and we introduced three practical options that capture different aspects of the distributional dynamics.

Through extensive experiments, we demonstrated that CRS consistently improves the performance of both pixel-space and latent-space diffusion models across various datasets, samplers, and NFEs ranging from 5 to 250. In particular, for sampling in pixel-space diffusion models, CRS-$v_x + v_{\mathrm{FID}}$ with tuned hyperparameters achieved the best results. In all other settings, CRS-$v_x$ with $\xi = 1$ performed consistently well and is recommended as a default configuration due to its simplicity and robustness.

Although this study has limitations, as discussed in Sec. 6—including the lack of a complete theoretical analysis and the need for further validation in other domains—we believe that CRS provides a solid and extensible foundation for future research on schedule optimization in diffusion models. We hope that this work encourages broader exploration of discrepancy-based scheduling, and that future research will further generalize CRS and extend its applicability beyond image generation.

### Acknowledgments

We thank the anonymous reviewers for their constructive and insightful comments, which helped improve the quality of this paper. We are also grateful to the team responsible for operating and maintaining the AI Cloud Platform, which provides the internal GPU infrastructure used in this work. Without their support, this research would not have been possible.

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

## A  Probability Distribution of Diffused Data

We derive the probability distribution of the diffused data used in the toy example (Fig. 1).

The distribution of the diffused data is given by:

$$q(\boldsymbol{x}_\alpha) = \int q(\boldsymbol{x}_\alpha|\boldsymbol{x}_0)q(\boldsymbol{x}_0)d\boldsymbol{x}_0, \tag{18}$$

$$q(\boldsymbol{x}_\alpha|\boldsymbol{x}_0) = \mathcal{N}(\boldsymbol{x}_\alpha; \alpha\boldsymbol{x}_0, \sigma^2\boldsymbol{I}), \tag{19}$$

where $q(\boldsymbol{x}_0)$ is the probability distribution of the target dataset.

We approximate $q(\boldsymbol{x}_0)$ by the empirical distribution:

$$q(\boldsymbol{x}_0) = \frac{1}{N}\sum_{n=1}^{N}\delta(\boldsymbol{x}_0 - \boldsymbol{x}_0^n), \tag{20}$$

where $N$ is the number of data samples, $\boldsymbol{x}_0^n$ is the $n$-th sample, and $\delta$ denotes the Dirac delta function. Substituting Eq. (20) into Eq. (18) yields:

$$q(\boldsymbol{x}_\alpha) = \frac{1}{N}\sum_{n=1}^{N}\mathcal{N}(\boldsymbol{x}_\alpha; \alpha\boldsymbol{x}_0^n, \sigma^2\boldsymbol{I}). \tag{21}$$

This derivation uses the following formula of Dirac's delta function:

$$\int f(\boldsymbol{x})\delta(\boldsymbol{x} - \boldsymbol{y})d\boldsymbol{x} = f(\boldsymbol{y}). \tag{22}$$

The resulting probability distribution is a Gaussian mixture distribution. We used Eq. (21) in the toy example.

## B  Derivation of Discrepancy Measure Based on Data Prediction from KLUB

We show that $v_x(\alpha)$ can be derived as a simplified variant of the KLUB introduced in AYS.

### B.1 KLUB in Variance-Preserving Process

The forward SDE in continuous-time diffusion models (Song et al., 2021b) is defined as:

$$d\boldsymbol{x} = f(t)\boldsymbol{x}dt + g(t)d\boldsymbol{\omega}, \tag{23}$$

where $f(t)\boldsymbol{x}$ is the drift coefficient, $g(t)$ is the diffusion coefficient, and $\boldsymbol{\omega}$ is the standard Wiener process. The corresponding reverse SDE is given by:

$$d\boldsymbol{x} = \left[ f(t)\boldsymbol{x} - g(t)^2 \nabla_{\boldsymbol{x}} \log p_t(\boldsymbol{x}) \right] dt + g(t)d\boldsymbol{\omega}. \tag{24}$$

To approximate the score $\nabla_{\boldsymbol{x}} \log p_t(\boldsymbol{x})$, we train a noise prediction model $\boldsymbol{\varepsilon}_\theta(\boldsymbol{x}, \alpha)$:

$$\boldsymbol{\varepsilon}_\theta(\boldsymbol{x}, \alpha) \simeq -\sigma \nabla_{\boldsymbol{x}} \log p_t(\boldsymbol{x}). \tag{25}$$

The data prediction $\boldsymbol{x}_\theta(\boldsymbol{x}, \alpha)$ is related to the noise prediction via:

$$\boldsymbol{x} = \alpha \boldsymbol{x}_\theta(\boldsymbol{x}, \alpha) + \sigma \boldsymbol{\varepsilon}_\theta(\boldsymbol{x}, \alpha). \tag{26}$$

By substituting Eq. (25) into Eq. (24) and using Eq. (26), the reverse SDE can be rewritten as

$$d\boldsymbol{x} = \boldsymbol{f}(\boldsymbol{x}, t)dt + g(t)d\boldsymbol{\omega}, \tag{27}$$

$$\boldsymbol{f}(\boldsymbol{x}, t) = \left( f(t) + \frac{1}{\sigma(t)^2} g(t)^2 \right) \boldsymbol{x} - \frac{\alpha(t)}{\sigma(t)^2} g(t)^2 \boldsymbol{x}_\theta(\boldsymbol{x}, \alpha(t)). \tag{28}$$

Stochastic DDIM numerically solves this reverse SDE. It exactly integrates the linear term and approximates the remaining term, introducing discretization error (Lu et al., 2022a;b). According to Sabour et al. (2024), in the interval $[t_{i-1}, t_i]$, Stochastic DDIM solves the discretized SDE:

$$d\boldsymbol{x} = \boldsymbol{f}'(\boldsymbol{x}, t)dt + g(t)d\boldsymbol{\omega}, \tag{29}$$

$$\boldsymbol{f}'(\boldsymbol{x}, t) = \left( f(t) + \frac{1}{\sigma(t)^2} g(t)^2 \right) \boldsymbol{x} - \frac{\alpha(t)}{\sigma(t)^2} g(t)^2 \boldsymbol{x}_\theta(\boldsymbol{x}_{t_i}, \alpha(t_i)). \tag{30}$$

Furthermore, the KLUB between probability distributions of $\boldsymbol{x}_{t_{i-1}}$ obtained by solving the true reverse SDE and the descretized one is introduced as

$$\text{KLUB}(t_{i-1}, t_t) = \frac{1}{2} \int_{t_{i-1}}^{t_i} \mathop{\mathbb{E}}_{\substack{\boldsymbol{x}_0 \sim q(\boldsymbol{x}_0) \\ \boldsymbol{x}_t \sim q(\boldsymbol{x}_t | \boldsymbol{x}_0) \\ \boldsymbol{x}_{t_i} \sim q(\boldsymbol{x}_{t_i} | \boldsymbol{x}_t)}} \left[ \frac{\|\boldsymbol{f}(\boldsymbol{x}_t, t) - \boldsymbol{f}'(\boldsymbol{x}_t, t)\|_2^2}{g(t)^2} \right] dt, \tag{31}$$

$$= \frac{1}{2} \int_{t_{i-1}}^{t_i} \mathop{\mathbb{E}}_{\substack{\boldsymbol{x}_0 \sim q(\boldsymbol{x}_0) \\ \boldsymbol{x}_t \sim q(\boldsymbol{x}_t | \boldsymbol{x}_0) \\ \boldsymbol{x}_{t_i} \sim q(\boldsymbol{x}_{t_i} | \boldsymbol{x}_t)}} \left[ \frac{\alpha(t)^2}{\sigma(t)^4} g(t)^2 \|\boldsymbol{x}_\theta(\boldsymbol{x}_t, \alpha(t)) - \boldsymbol{x}_\theta(\boldsymbol{x}_{t_i}, \alpha(t_i))\|_2^2 \right] dt. \tag{32}$$

In the variance-preserving setting, the diffusion coefficient satisfies:

$$\frac{1}{2} g(t)^2 = -\frac{\dot{\alpha}}{\alpha}. \tag{33}$$

Substituting Eq. (33) into Eq. (32) gives:

$$\text{KLUB}(t_{i-1}, t_i) = -\int_{t_{i-1}}^{t_i} dt \dot{\alpha}(t) \frac{\alpha(t)}{\sigma(t)^4} \mathop{\mathbb{E}}_{\substack{\boldsymbol{x}_0 \sim q(\boldsymbol{x}_0) \\ \boldsymbol{x}_t \sim q(\boldsymbol{x}_t | \boldsymbol{x}_0) \\ \boldsymbol{x}_{t_i} \sim q(\boldsymbol{x}_{t_i} | \boldsymbol{x}_t)}} \left[ \|\boldsymbol{x}_\theta(\boldsymbol{x}_t, \alpha(t)) - \boldsymbol{x}_\theta(\boldsymbol{x}_{t_i}, \alpha(t_i))\|_2^2 \right]. \tag{34}$$

### B.2 KLUB as Discrepancy Measure of CRS

To apply KLUB within the CRS framework, we reinterpret it as a discrepancy measure over adjacent steps in the diffusion process. Because the KLUB is introduced to evaluate the discretization error when solving the reverse SDE, it is reasonable to regard the KLUB as the traceability measure of the diffusion process.

We approximate the integral in the KLUB using Simpson's rule as follows:

$$\text{KLUB}(t, t + \Delta t) = \frac{\dot{\alpha}(t)\alpha(t)}{2\sigma(t)^4}\overline{\mathcal{D}_x^2}(t, t + \Delta t)\Delta t, \tag{35}$$

$$\overline{\mathcal{D}_x^2}(t, t') = \mathbb{E}_{\substack{\boldsymbol{x}_0 \sim q(\boldsymbol{x}_0) \\ \boldsymbol{x}_t \sim q(\boldsymbol{x}_t | \boldsymbol{x}_0) \\ \boldsymbol{x}_{t'} \sim q(\boldsymbol{x}_{t'} | \boldsymbol{x}_t)}} \left[ \|\boldsymbol{x}_\theta(\boldsymbol{x}_t, \alpha(t)) - \boldsymbol{x}(\boldsymbol{x}_{t'}, \alpha(t'))\|_2^2 \right]. \tag{36}$$

Then, to treat $\overline{\mathcal{D}_x^2}(t, t')$ as a function of $\alpha$ rather than $t$, we introduce $\overline{D_x^2}(\alpha(t), \alpha(t')) = \overline{\mathcal{D}_x^2}(t, t')$:

$$\overline{D_x^2}(\alpha, \alpha') = \mathbb{E}_{\boldsymbol{x}_\alpha, \boldsymbol{x}_{\alpha'} \sim q(\boldsymbol{x}_\alpha, \boldsymbol{x}_{\alpha'})} \left[ \|\boldsymbol{x}_\theta(\boldsymbol{x}_\alpha, \alpha) - \boldsymbol{x}_\theta(\boldsymbol{x}_{\alpha'}, \alpha')\|_2^2 \right], \tag{37}$$

Here, $q(\boldsymbol{x}_\alpha, \boldsymbol{x}_{\alpha'})$ is given by

$$q(\boldsymbol{x}_\alpha, \boldsymbol{x}_{\alpha'}) = \int q(\boldsymbol{x}_{\alpha'} | \boldsymbol{x}_\alpha) q(\boldsymbol{x}_\alpha | \boldsymbol{x}_0) q(\boldsymbol{x}_0) d\boldsymbol{x}_0, \tag{38}$$

$$q(\boldsymbol{x}_\alpha | \boldsymbol{x}_0) = \mathcal{N}(\boldsymbol{x}_\alpha; \alpha\boldsymbol{x}_0, \sigma^2 \boldsymbol{I}), \tag{39}$$

$$q(\boldsymbol{x}_{\alpha'} | \boldsymbol{x}_\alpha) = \mathcal{N}(\boldsymbol{x}_{\alpha'}; \beta\boldsymbol{x}_\alpha, \delta^2 \boldsymbol{I}), \tag{40}$$

where $\beta = \frac{\alpha'}{\alpha}$, $\sigma = \sqrt{1 - \alpha^2}$, and $\delta = \sqrt{1 - \beta^2}$. Using $\overline{D_x^2}(\alpha, \alpha')$, we can rewrite $\text{KLUB}(t, t + \Delta t)$ as

$$\text{KLUB}(t, t + \Delta t) = \frac{\dot{\alpha}\alpha}{2\sigma^4}\overline{D_x^2}(\alpha, \alpha + \Delta\alpha)\Delta t, \tag{41}$$

$$= \frac{\dot{\alpha}\alpha}{2\sigma^4} \frac{\overline{D_x^2}(\alpha, \alpha + \Delta\alpha) - \overline{D_x^2}(\alpha, \alpha)}{\Delta\alpha} \frac{\Delta\alpha}{\Delta t}(\Delta t)^2, \tag{42}$$

$$\simeq \left\{ \frac{d\alpha}{dt} \cdot \frac{\sqrt{\alpha}}{\sqrt{2}\sigma^2} \sqrt{\left. \frac{\partial \overline{D_x^2}(\alpha, \alpha')}{\partial \alpha'} \right|_{\alpha'=\alpha}} \Delta t \right\}^2. \tag{43}$$

We use $\overline{D_x^2}(\alpha, \alpha) = 0$ in the second equality. Finally, by adopting $\sqrt{\text{KLUB}(t, t')}$ as the discrepancy measure $D(t, t')$ in CRS, we obtain the following equation:

$$D(t, t + \Delta t) = \sqrt{\text{KLUB}(t, t + \Delta t)} = -\frac{d\alpha}{dt} \cdot \underbrace{\frac{\sqrt{\alpha}}{\sqrt{2}\sigma^2} \sqrt{\left. \frac{\partial \overline{D_x^2}(\alpha, \alpha')}{\partial \alpha'} \right|_{\alpha'=\alpha}}}_{v_{\text{KLUB}}(\alpha)} \Delta t. \tag{44}$$

which corresponds to the general form of CRS in Eq. (8). We can interpret $v_x(\alpha)$ as a simplified variant of $v_{\text{KLUB}}(\alpha)$, in which the multiplicative weighting term $\frac{\sqrt{\alpha}}{\sqrt{2}\sigma^2}$ is omitted.

Although it is possible to optimize the noise schedule using $v_{\text{KLUB}}(\alpha)$ directly, this can lead to undesirable behavior, as the weighting term $\frac{\sqrt{\alpha}}{\sqrt{2}\sigma^2}$ grows very large when $\alpha \simeq 1$ (i.e., $\sigma \simeq 0$), which can overwhelm the influence of $\overline{D_x^2}(\alpha, \alpha')$. Since $\overline{D_x^2}(\alpha, \alpha')$ can be interpreted as capturing the distributional change between diffused data distributions (as discussed in Section 4.3), the simplified variant $v_x(\alpha)$ is more aligned with the objective of CRS.

## C   Implementation of CRS

We describe how to compute $v(\alpha)$ using each discrepancy measure.

Table 8: Hyperparameters for computing $v_{\text{FID}}(\alpha)$.

| Parameter | Description | Used Value |
|---|---|---|
| $T$ | Number of timesteps for simulating the forward process | $10^3$ |
| $\{\alpha_t\}_{t=0,\dots,T}$ | Grid of $\alpha$-values for evaluating $v(\alpha)$ | $\alpha_t = 1 - \left(\frac{t}{T}\right)^p$, where $p = \begin{cases} 2, & \text{pixel space} \\ 1, & \text{latent space} \end{cases}$ |
| $\phi$ | Feature extractor for computing FID | Pixel space: Inception-V3 model used in the EDM repository for FID evaluation |
| | | Latent space: ResNet-50 trained by us on ImageNet classification in the latent space |

## C.1 Discrepancy Measure Based on FID

Algorithm 1 presents the pseudocode for computing $v_{\text{FID}}(\alpha)$, and hyperparameters are summarized in Tabel 8. Once the forward process is defined, $v(\alpha)$ can be computed by simulating the forward diffusion using the training dataset. Since this procedure is independent of model parameters, it only needs to be executed once prior to training the diffusion model.

It is important to note that, in latent-space diffusion models, the probability-distributional change must be measured in the latent space embedded by the autoencoder, as the Gaussian approximation in Eq. (4) applies to the diffusion process defined in the latent space. Therefore, for latent-space models, we trained a ResNet-50 (He et al., 2016) classifier on ImageNet in the autoencoder's latent space and use it as the feature extractor $\phi$.

To compute $v(\alpha)$, we discretize the range of $\alpha$ using a one-dimensional grid $\{\alpha_t\}_{t=0,\dots,T}$ and approximate the continuous function via linear interpolation. In our experiments, we define the discretized grid of $\alpha$ values as:

$$\alpha_t = 1 - \left(\frac{t}{T}\right)^p, \tag{45}$$

where $T = 1000$. For pixel-space diffusion models, where $v(\alpha)$ exhibits sharp variation near $\alpha \simeq 1$, we set $p = 2$ to allocate more evaluation points in that region. For latent-space models, where $v(\alpha)$ is more uniformly distributed, we use $p = 1$.

The computational cost of $v(\alpha)$ is proportional to the number of training images. For the pixel-space diffusion process on LSUN Horse 256×256 with 2M images, the computation takes approximately 35 hours using eight A100 GPUs. For the latent-space process on ImageNet 256×256 with 1.28M images, it takes about 12 hours using eight A100 GPUs. To address the scalability issue for large-scale datasets, we introduce a practical alternative in Sec. 5.7.

## C.2 Discrepancy Measure Based on Data Prediction

Algorithm 2 presents the pseudocode for computing $v_x(\alpha)$ using a trained diffusion model. As in the case of $v_{\text{FID}}(\alpha)$, this involves simulating the forward process on the training data and measuring changes in the predicted data across adjacent timesteps. Hyperparameters are summarized in Table 9. We use the same configuration for all our experiments, regardless of the diffusion type or dataset.

The computation time for obtaining $v_x(\alpha)$ is negligible compared with the training time of the diffusion model. It takes approximately 5.5 hours using eight H100 GPUs in pixel-space diffusion models. In latent-space diffusion models, it takes approximately 1.5 hours using eight A100 GPUs.

Algorithm 3 shows how $v_x(\alpha)$ can be used to adaptively optimize the training schedule. Since the data prediction is expressed as $\boldsymbol{x}_\theta(\boldsymbol{x}, \alpha) = \frac{\boldsymbol{x} - \sigma \boldsymbol{\varepsilon}_\theta(\boldsymbol{x}, \alpha)}{\alpha}$, $v_x(\alpha)$ can become numerically unstable as $\alpha \to 0$. To prevent this, we restrict evaluation to the range $\alpha_{\text{th}} \leq \alpha \leq \alpha_{\max}$ and fix $v_x(\alpha) = v_x(\alpha_{\text{th}})$ for $\alpha < \alpha_{\text{th}}$.

---

**Algorithm 1** $v(\alpha)$ on basis of FID

---

**Require:** training dataset $\{\boldsymbol{x}^{(n)}\}_{n=0,\ldots,N-1}$, timesteps $T$, noise levels $\{\alpha_t\}_{t=0,\ldots,T}$, feature model $\boldsymbol{\phi}$
**Ensure:** LinearInterp1d($\{\alpha_t\}_{t=0,\ldots,T}, \{v_t\}_{t=0,\ldots,T}$)
 1: # Compute feature vectors of diffused data
 2: **for** $n = 0$ to $N - 1$ **do**
 3:     $\boldsymbol{x}_0^{(n)} \leftarrow \boldsymbol{x}^{(n)}$
 4:     **if** is_latent_space **then**
 5:         $\boldsymbol{x}_0^{(n)} \leftarrow \text{autoencoder}(\boldsymbol{x}_0^{(n)})$
 6:     **end if**
 7:
 8:     # Simulate the forward process
 9:     **for** $t = 1$ to $T$ **do**
10:         $\beta_t \leftarrow \frac{\alpha_t}{\alpha_{t-1}}$
11:         $\delta_t \leftarrow \sqrt{1 - \beta_t^2}$
12:         $\boldsymbol{x}_t^{(n)} \leftarrow \beta_t \boldsymbol{x}_{t-1}^{(n)} + \delta_t \mathcal{N}(\boldsymbol{0}, \boldsymbol{I})$
13:     **end for**
14: **end for**
15:
16: # Compute statistics of feature vectors
17: $\boldsymbol{\mu}_t \leftarrow \text{mean}\left(\left\{\phi\left(\boldsymbol{x}_t^{(n)}\right)\right\}_{n=0,\ldots,N-1}\right), \ \forall t \in \{0,\ldots,T\}$
18: $\boldsymbol{\Sigma}_t \leftarrow \text{cov}\left(\left\{\phi\left(\boldsymbol{x}_t^{(n)}\right)\right\}_{n=0,\ldots,N-1}\right), \ \forall t \in \{0,\ldots,T\}$
19:
20: # Compute $v(\alpha)$ based on FID
21: **for** $t = 0$ to $T - 1$ **do**
22:     $v_t \leftarrow \dfrac{\text{FID}(\boldsymbol{\mu}_t, \boldsymbol{\Sigma}_t, \boldsymbol{\mu}_{t+1}, \boldsymbol{\Sigma}_{t+1})}{\alpha_t - \alpha_{t+1}}$
23: **end for**
24: $v_T \leftarrow v_{T-1}$

---

Table 9: Hyperparameters used to compute $v_x(\alpha)$ for optimizing sampling schedule.

| Parameter | Description | Used Value |
|---|---|---|
| $T$ | Number of timesteps for simulating the forward process | $10^3$ |
| $S$ | Number of samples used to estimate $\overline{D_x^2}(\alpha, \alpha')$ | $10^4$ |
| $\alpha_s$ | Maximum $\alpha$ used to evaluate $v(\alpha)$ | 1.0 |
| $\alpha_e$ | Minimum $\alpha$ used to evaluate $v(\alpha)$ | 0.0 |

Similar to VDM++ (Kingma & Gao, 2023), we divide the range of $\alpha$ into $B$ uniform bins and estimate $\overline{D_x^2}$ for each bin using an exponential moving average (EMA).

Hyperparameters are summarized in Table 10. We use the same configuration for all our experiments, regardless of the diffusion type or dataset. Although not explicitly included in the pseudocode, the update of $\overline{D_b^2}$ starts after 1000 iterations, and the noise schedule $\alpha(t)$ is updated every 100 iterations.

While this adaptive scheme slightly increases training time (by 20–30% compared to fixed schedules; see Table 15 in Appendix E), we find that the schedule stabilizes early: for example, on LSUN Horse 256×256, $v_x(\alpha)$ changes very little after 10 epochs. Thus, fixing the schedule after a few epochs or reducing the frequency of $\overline{D_x^2}$ updates may mitigate the cost without degrading performance. Developing a more efficient implementation remains an important direction for future work.

---

**Algorithm 2** $v_x(\alpha)$ for optimizing sampling schedule.

---

**Require:** training dataset $\{\boldsymbol{x}^{(n)}\}_{n=0,\ldots,N-1}$, trained diffusion model $\boldsymbol{\varepsilon}_\theta(\boldsymbol{x}, \alpha)$, number of timesteps $T$, number of samples for mean $S$, range of $\alpha$ $(\alpha_s, \alpha_e)$
**Ensure:** $\text{LinearInterp1d}\big(\{\alpha_t\}_{t=0,\ldots,T}, \{v_t\}_{t=0,\ldots,T}\big)$

1:  # Evaluate $\mathbb{E}_{\boldsymbol{x}_\alpha, \boldsymbol{x}_{\alpha'}}\left[\|\boldsymbol{x}_\theta(\boldsymbol{x}_\alpha, \alpha) - \boldsymbol{x}_\theta(\boldsymbol{x}_{\alpha'}, \alpha')\|_2^2\right]$

2:  $\overline{D_t^2} \leftarrow 0, \ \forall t \in \{0, 1, \ldots, T\}$

3:  $\Delta\alpha \leftarrow \frac{1}{T}(\alpha_s - \alpha_e)$

4:  **for** $n$ **in** $\text{randperm}(N)[:S]$ **do**

5:     $\boldsymbol{x}_0 \leftarrow \boldsymbol{x}^{(n)}$

6:     **if** model operates in latent space **then**

7:         $\boldsymbol{x}_0 \leftarrow \text{autoencoder}(\boldsymbol{x}_0)$

8:     **end if**

9:     $\boldsymbol{y}_0 \leftarrow \boldsymbol{x}_0$

10:

11:     # Simulate the forward process

12:     **for** $t = 1$ to $T$ **do**

13:         $\alpha_t \leftarrow \alpha_s - \Delta\alpha\, t, \quad \sigma_t \leftarrow \sqrt{1 - \alpha_t^2}$

14:         $\beta_t \leftarrow \frac{\alpha_t}{\alpha_{t-1}}, \quad \delta_t \leftarrow \sqrt{1 - \beta_t^2}$

15:         $\boldsymbol{x}_t \leftarrow \beta_t \boldsymbol{x}_{t-1} + \delta_t \mathcal{N}(\boldsymbol{0}, \boldsymbol{I})$

16:         $\boldsymbol{y}_t \leftarrow \frac{1}{\alpha_t}(\boldsymbol{x}_t - \sigma_t\, \boldsymbol{\varepsilon}_\theta(\boldsymbol{x}_t, \alpha_t))$

17:         $\overline{D_t^2} \leftarrow \overline{D_t^2} + \frac{1}{S}\|\boldsymbol{y}_t - \boldsymbol{y}_{t-1}\|_2^2$

18:     **end for**

19:  **end for**

20:

21:  # Compute $v(\alpha)$ using $\overline{D_t^2}$

22:  **for** $t = 1$ to $T$ **do**

23:     $v_t \leftarrow \sqrt{\frac{1}{\Delta\alpha}\overline{D_t^2}}$

24:  **end for**

25:  $v_0 \leftarrow v_1$

---

Table 10: Hyperparameters used for optimizing the training schedule with CRS-$v_x$.

| Parameter | Description | Used Value |
|---|---|---|
| $\alpha_{\min}$ | Minimum value of $\alpha$ for the training schedule | 0.0 |
| $\alpha_{\max}$ | Maximum value of $\alpha$ for the training schedule | 1.0 |
| $\Delta\alpha$ | Small step used for approximating the derivative | $10^{-3}$ |
| $\alpha_{\text{th}}$ | Threshold of $\alpha$ for clipping $v(\alpha)$ | 0.01 |
| $\xi$ | A parameter controlling the dependence of $\alpha(t)$ on $v(\alpha)$ | 1 |
| $B$ | Number of bins used to divide the range of $\alpha$ | 100 |
| $e$ | EMA decay rate for approximately evaluating $\overline{D_x^2}(\alpha, \alpha')$ | 0.995 |
| $\alpha(t)$ | Initial noise schedule | $1 - t$ |

## D   Hyperparameter Tuning for Combining Multiple Discrepancy Measures

CRS-$v_x + v_{\text{FID}}$ achieves the best performance for sampling in pixel-space diffusion models. CRS-$v_x + v_{\text{FID}}$ involves four hyperparameters: $w_x$, $w_{\text{FID}}$, $\xi_x$, and $\xi_{\text{FID}}$. The effects of these hyperparameters on the resulting noise schedules are illustrated in Figs. 4, 5, and 6. Increasing either $w_{\text{FID}}$ or $\xi_{\text{FID}}$ amplifies the influence of $v_{\text{FID}}(\alpha)$, resulting in more timesteps being allocated to the region where $\alpha \simeq 1$ (see Figs. 4 and 5).

---

**Algorithm 3** Adaptive optimization of training schedule using $v_x(\alpha)$.

---

**Require:** training dataset $\{\boldsymbol{x}^{(n)}\}_{n=0,\ldots,N}$, initial noise schedule $\alpha(t)$, number of bins $B$, decay rate of EMA $e$, range of $\alpha$ $(\alpha_{\max}, \alpha_{\min})$, derivative step $\Delta\alpha$, minimum value for evaluating $v(\alpha)$ $\alpha_{\text{th}}$, hyperparameter of CRS $\xi$

**Ensure:** noise prediction model $\boldsymbol{\varepsilon}_\theta(\boldsymbol{x}, \alpha)$

1:  $\alpha_b \leftarrow \alpha_{\text{th}} + (\alpha_{\max} - \alpha_{\text{th}})\dfrac{b}{B}, \quad \forall b \in \{0, \ldots, B\}$

2:  $\overline{D_b^2} \leftarrow 10^{-6}, \quad \forall b \in \{0, \ldots, B-1\}$

3:  **for** $n$ **in** randperm($N$) **do**

4:      # Compute loss for training diffusion models

5:      $\boldsymbol{x}_0 \leftarrow \boldsymbol{x}^{(n)}, \quad t \sim \mathcal{U}(0,1), \quad \boldsymbol{\varepsilon} \sim \mathcal{N}(\boldsymbol{0}, \boldsymbol{I})$

6:      $\alpha \leftarrow \alpha(t), \quad \sigma \leftarrow \sqrt{1 - \alpha^2}$

7:      $\boldsymbol{\varepsilon}_\alpha \leftarrow \boldsymbol{\varepsilon}_\theta(\alpha\boldsymbol{x}_0 + \sigma\boldsymbol{\varepsilon}, \alpha)$

8:      $loss \leftarrow \dfrac{1}{2}\|\boldsymbol{\varepsilon}_\alpha - \boldsymbol{\varepsilon}\|_2^2$

9:

10:     # Evaluate $\overline{D_b^2}$ using EMA (without gradient)

11:     **if** $\alpha \geq \alpha_{\text{th}}$ **then**

12:        $\alpha' \leftarrow \alpha - \Delta\alpha, \quad \sigma' \leftarrow \sqrt{1 - \alpha'^2}, \quad \beta' \leftarrow \dfrac{\alpha'}{\alpha}, \quad \delta' \leftarrow \sqrt{1 - \beta'^2}$

13:        $\boldsymbol{x}_{\alpha'} \leftarrow \beta'\boldsymbol{x}_\alpha + \delta'\mathcal{N}(\boldsymbol{0}, \boldsymbol{I})$

14:        $\boldsymbol{\varepsilon}_{\alpha'} \leftarrow \boldsymbol{\varepsilon}_\theta(\boldsymbol{x}_{\alpha'}, \alpha')$

15:        $\boldsymbol{y}_\alpha \leftarrow \dfrac{1}{\alpha}(\boldsymbol{x}_\alpha - \sigma\boldsymbol{\varepsilon}_\alpha), \quad \boldsymbol{y}_{\alpha'} \leftarrow \dfrac{1}{\alpha'}(\boldsymbol{x}_{\alpha'} - \sigma'\boldsymbol{\varepsilon}_{\alpha'})$

16:        $b \leftarrow \min\left(\left\lfloor \dfrac{\alpha - \alpha_{\text{th}}}{\alpha_{\max} - \alpha_{\text{th}}}B \right\rfloor, B-1\right)$

17:        $\overline{D_b^2} \leftarrow e\overline{D_b^2} + (1-e)\|\boldsymbol{y}_\alpha - \boldsymbol{y}_{\alpha'}\|_2^2$

18:        $v_b \leftarrow \sqrt{\dfrac{1}{\Delta\alpha}\overline{D_b^2}}$

19:     **end if**

20:

21:     # Update training schedule $\alpha(t)$

22:     $\mathcal{X} \leftarrow \{\alpha_{\min}, \alpha_0, \alpha_1, \ldots, \alpha_B\}$

23:     $\mathcal{Y} \leftarrow \{v_0, v_0, v_1, \ldots, v_{B-1}, v_{B-1}\}$

24:     $v(\alpha) \leftarrow \text{LinearInterp1d}(\mathcal{X}, \mathcal{Y})$

25:     **Update noise schedule $\alpha(t)$ by solving Eq. (9)**

26:

27:     # Update model parameters $\theta$

28:     **loss.backward()**

29:     **optimizer.step()**

30: **end for**

---

Conversely, increasing either $w_x$ or $\xi_x$ strengthens the contribution of $v_x(\alpha)$, leading to more timesteps being assigned to the region where $\alpha \simeq 0$ (see Figs. 4 and 6).

We tuned these hyperparameters in two steps. First, we fixed $\xi_x = \xi_{\text{FID}} = 1$ and searched for appropriate values of $w_x$ and $w_{\text{FID}}$ under the constraint $w_x + w_{\text{FID}} = 1$. Then, we fixed the weights and adjusted $\xi_x$ and $\xi_{\text{FID}}$. Although further iterations between tuning weights and exponents are possible, we found that just one iteration per step was typically sufficient to outperform conventional noise schedules.

The detailed tuning results on LSUN Horse 256×256 are presented in Table 11. In the first step, adjusting $w_x$ and $w_{\text{FID}}$ led to substantial changes in FID at both NFE = 250 and NFE = 50. In the second step, tuning $\xi_x$ and $\xi_{\text{FID}}$ had little impact at NFE = 250, but resulted in a significant FID improvement at NFE = 50. This suggests that hyperparameter sensitivity increases as the NFEs decrease.

Table 11: Detailed results of hyperparameter tuning for CRS-$v_x + v_{\text{FID}}$ on LSUN Horse 256×256 in pixel-space diffusion model. DPM-Solver++(2M) was used as sampler. To reduce computational cost, FID scores were calculated using 10K samples.

| NFE | Tuning step | $w_{\text{FID}}$ | $x_x$ | $\xi_{\text{FID}}$ | $\xi_x$ | FID (10K) |
|---|---|---|---|---|---|---|
| 250 | 1st | 0.9 | 0.1 | 1.0 | 1.0 | 3.78 |
| | | 0.7 | 0.3 | | | **3.70** |
| | | 0.5 | 0.5 | | | **3.70** |
| | | 0.3 | 0.7 | | | 3.73 |
| | | 0.1 | 0.9 | | | 3.76 |
| | 2nd | 0.7 | 0.3 | 0.5 | 0.5 | 3.77 |
| | | | | | 1.0 | 4.30 |
| | | | | | 1.2 | 3.75 |
| | | | | | 1.4 | 3.94 |
| | | | | 1.0 | 0.5 | 3.76 |
| | | | | | 1.0 | 3.70 |
| | | | | | 1.2 | **3.69** |
| | | | | | 1.4 | 3.78 |
| | | | | 1.2 | 0.5 | 3.82 |
| | | | | | 1.0 | **3.69** |
| | | | | | 1.2 | 3.78 |
| | | | | | 1.4 | 3.77 |
| | | | | 1.4 | 0.5 | 3.83 |
| | | | | | 1.0 | 3.78 |
| | | | | | 1.2 | 3.77 |
| | | | | | 1.4 | 3.77 |
| 50 | 1st | 0.9 | 0.1 | 1.0 | 1.0 | 6.13 |
| | | 0.7 | 0.3 | | | 4.42 |
| | | 0.5 | 0.5 | | | **4.38** |
| | | 0.3 | 0.7 | | | 4.78 |
| | | 0.1 | 0.9 | | | 7.03 |
| | 2nd | 0.5 | 0.5 | 0.5 | 0.5 | 7.42 |
| | | | | | 1.0 | 7.44 |
| | | | | | 1.2 | 7.48 |
| | | | | | 1.4 | 7.79 |
| | | | | 1.0 | 0.5 | 4.85 |
| | | | | | 1.0 | 4.38 |
| | | | | | 1.2 | 4.32 |
| | | | | | 1.4 | 4.27 |
| | | | | 1.2 | 0.5 | 4.57 |
| | | | | | 1.0 | 4.09 |
| | | | | | 1.2 | 3.96 |
| | | | | | 1.4 | 3.93 |
| | | | | 1.4 | 0.5 | 4.51 |
| | | | | | 1.0 | 3.95 |
| | | | | | 1.2 | 3.85 |
| | | | | | 1.4 | **3.82** |

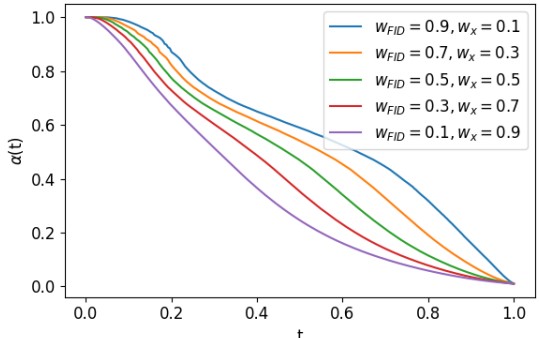

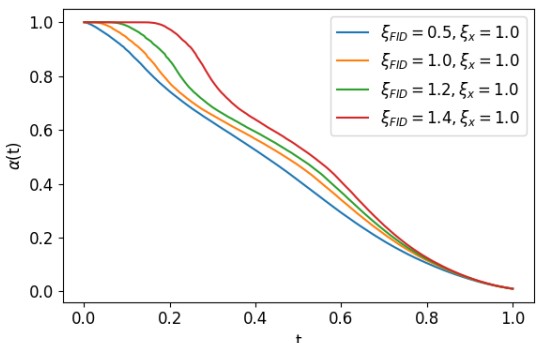

Figure 4: Effect of the weights $w_x$ and $w_{\text{FID}}$ on the noise schedule generated by CRS-$v_x + v_{\text{FID}}$, with fixed exponents $\xi_x = \xi_{\text{FID}} = 1$.

Figure 5: Effect of the exponent $\xi_{\text{FID}}$ on the noise schedule generated by CRS-$v_x + v_{\text{FID}}$, with fixed weights $w_x = w_{\text{FID}} = 0.5$.

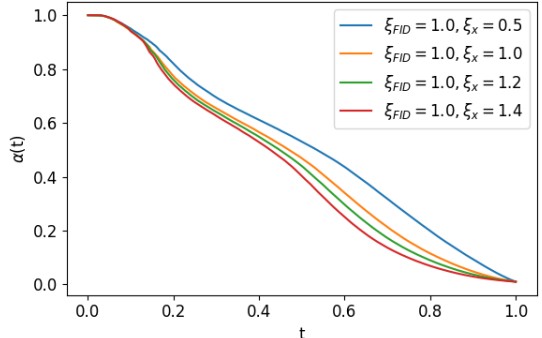

Figure 6: Effect of the exponent $\xi_x$ on the noise schedule generated by CRS-$v_x + v_{\text{FID}}$, with fixed weights $w_x = w_{\text{FID}} = 0.5$.

The final hyperparameter settings used in our experiments are summarized in Table 12. The resulting noise schedules $\alpha(t)$ for CRS-$v_x + v_{\text{FID}}$ are visualized in Figs. 7, 8, 9, 10, and 11.

# E  Implementation Details

## E.1  U-Net Architecture

The hyperparameters of the U-Net model are listed in Table 13. For latent-space diffusion models, we use the same settings as those for ImageNet 64×64 in ADM (Dhariwal & Nichol, 2021). For pixel-space diffusion models, we adopt the same settings as those for LSUN 256×256 in ADM.

## E.2  Training Hyperparameters

The training hyperparameters are summarized in Table 14. We save ten checkpoints for each model. FID scores are computed using 10K samples for all checkpoints, and we report the results from the checkpoint with the best FID.

The training times for each model configuration are provided in Table 15. When using CRS to optimize the training schedule, training time increases by 20–30% compared to conventional noise schedules, due to the overhead of computing $v_x(\alpha)$ during training. In practice, however, we observe that the noise schedule stabilizes early; for example, on LSUN Horse 256×256, $v_x(\alpha)$ shows minimal change beyond 10 epochs. This suggests that freezing the schedule after a small number of epochs or reducing the frequency of $\overline{D_x^2}$ updates

Table 12: Hyperparameter settings used for CRS-$v_x + v_{\text{FID}}$ in our experiments on LSUN Horse 256×256, LSUN Bedroom 256×256, FFHQ 64×64, and CIFAR10 32×32.

| Dataset | Sampler | NFE | $w_{\text{FID}}$ | $w_x$ | $\xi_{\text{FID}}$ | $\xi_x$ |
|---|---|---|---|---|---|---|
| LSUN Horse 256×256 | SDE-DPM-Solver++(2M) | 250 | 0.9 | 0.1 | 1.4 | 1.0 |
| | | 50 | 0.7 | 0.3 | 1.4 | 1.2 |
| | PNDM | 250 | 0.9 | 0.1 | 1.0 | 0.5 |
| | | 50 | 0.5 | 0.5 | 1.2 | 0.5 |
| | DPM-Solver++(2M) | 250 | 0.7 | 0.3 | 1.0 | 1.2 |
| | | 50 | 0.5 | 0.5 | 1.4 | 1.4 |
| LSUN Bedroom 256×256 | DPM-Solver++(2M) | 10 | 0.5 | 0.5 | 1.0 | 1.0 |
| | | 5 | 0.5 | 0.5 | 1.0 | 1.2 |
| | UniPC | 10 | 0.5 | 0.5 | 1.0 | 1.5 |
| | | 5 | 0.3 | 0.7 | 1.0 | 1.0 |
| FFHQ 64×64 | DPM-Solver++(2M) | 20 | 0.5 | 0.5 | 1.7 | 1.4 |
| | | 10 | 0.5 | 0.5 | 1.6 | 1.4 |
| | UniPC(3M) | 20 | 0.5 | 0.5 | 1.7 | 1.4 |
| | | 10 | 0.5 | 0.5 | 1.6 | 1.6 |
| | iPNDM(3M) | 20 | 0.5 | 0.5 | 1.6 | 1.2 |
| | | 10 | 0.5 | 0.5 | 1.6 | 1.2 |
| | iPNDM(4M) | 20 | 0.5 | 0.5 | 1.4 | 1.4 |
| | | 10 | 0.5 | 0.5 | 1.6 | 1.4 |
| CIFAR10 32×32 | DPM-Solver++(2M) | 20 | 0.5 | 0.5 | 1.8 | 1.2 |
| | | 10 | 0.5 | 0.5 | 1.7 | 1.2 |
| | UniPC(3M) | 20 | 0.5 | 0.5 | 2.3 | 1.0 |
| | | 10 | 0.5 | 0.5 | 1.9 | 1.5 |
| | iPNDM(3M) | 20 | 0.5 | 0.5 | 1.7 | 1.4 |
| | | 10 | 0.5 | 0.5 | 1.7 | 1.4 |
| | iPNDM(4M) | 20 | 0.5 | 0.5 | 1.6 | 1.6 |
| | | 10 | 0.5 | 0.5 | 1.6 | 1.6 |

Table 13: Hyperparameters of UNet model.

| | Latent-space diffusion models | Pixel-space diffusion models |
|---|---|---|
| Resolution | $64 \times 64$ | $256 \times 256$ |
| Number of parameters | 296M | 552M |
| Channels | 192 | 256 |
| Depth | 3 | 2 |
| Channels multiple | 1,2,3,4 | 1,1,2,2,4,4 |
| Heads channels | 64 | 64 |
| Attention resolution | 32,16,8 | 32,16,8 |
| BigGAN up/downsample | True | True |
| Dropout | 0.1 | 0.1 |

Table 14: Hyperparameters for training diffusion models. Same configuration was used for training latent-space and pixel-space diffusion models.

| Item | Value |
|---|---|
| EMA decay | 0.9999 |
| Optimizer | Adam |
| $\beta_1$ | 0.9 |
| $\beta_2$ | 0.999 |
| Learning rate | 1e-4 |
| Batch size | $32 \times 8$ |

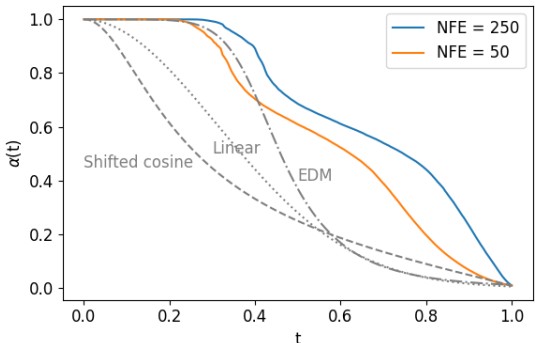

Figure 7: Noise schedules generated by CRS-$v_x + v_{\mathrm{FID}}$ for LSUN Horse 256×256, used with SDE-DPM-Solver++(2M) during sampling.

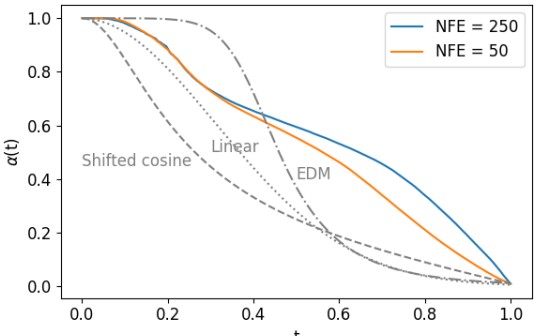

Figure 8: Noise schedules generated by CRS-$v_x + v_{\mathrm{FID}}$ for LSUN Horse 256×256, used with PNDM during sampling.

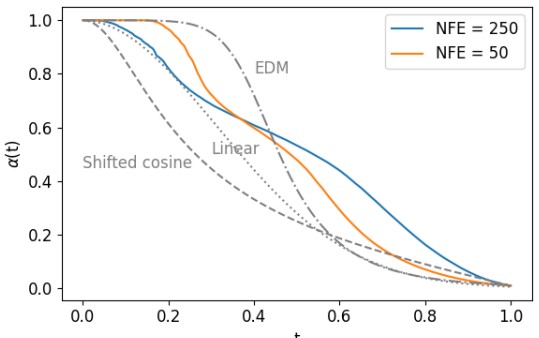

Figure 9: Noise schedules generated by CRS-$v_x + v_{\mathrm{FID}}$ for LSUN Horse 256×256, used with DPM-Solver++(2M) during sampling.

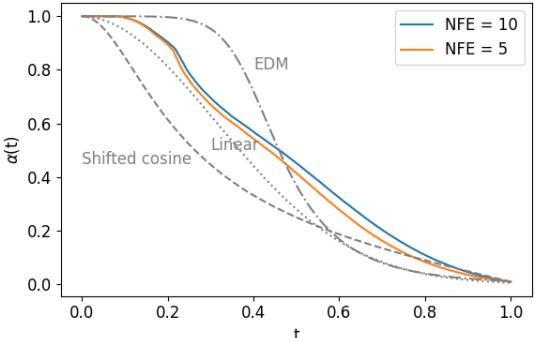

Figure 10: Noise schedules generated by CRS-$v_x + v_{\mathrm{FID}}$ for LSUN Bedroom 256×256, used with DPM-Solver++(2M) during sampling.

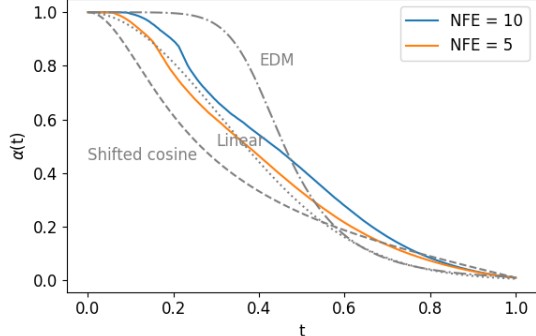

Figure 11: Noise schedules generated by CRS-$v_x + v_{\mathrm{FID}}$ for LSUN Bedroom 256×256, used with UniPC during sampling.

Table 15: Training time of diffusion models.

| Diffusion type | Dataset | GPUs | Epochs | Training schedule | Training time |
|---|---|---|---|---|---|
| Latent space | LSUN Church | A100 $\times$ 8 | 1000 | Linear VDM++ CRS-$v_{\text{FID}}$ | 85 hours |
| | | | | CRS-$v_\varepsilon$ CRS-$v_x$ | 105 hours |
| | LSUN Bedroom | A100 $\times$ 8 | 100 | CRS-$v_{\text{FID}}$ | 190 hours |
| | | | | CRS-$v_\varepsilon$ CRS-$v_x$ | 245 hours |
| | ImageNet | A100 $\times$ 8 | 500 | Linear VDM++ CRS-$v_{\text{FID}}$ | 430 hours |
| | | | | CRS-$v_\varepsilon$ CRS-$v_x$ | 525 hours |
| Pixel space | LSUN Horse | H100 $\times$ 8 | 50 | Linear Shifted cosine VDM++ | 170 hours |
| | | | | CRS-$v_\varepsilon$ CRS-$v_x$ | 215 hours |
| | LSUN Bedroom | H100 $\times$ 8 | 50 | CRS-$v_x$ | 325 hours |

can mitigate training overhead without degrading performance. Developing a more efficient implementation remains an important direction for future work.

### E.3 Noise Schedules for Training and Sampling

The conventional noise schedules used in this study are summarized in Table 16.

## F Detailed Experimental Results

We provide detailed experimental results to confirm the effectiveness of CRS for optimizing both training and sampling schedules.

### F.1 Results on Pixel-Space Diffusion Models

**LSUN Horse 256$\times$256:** The results showing the dependence on training and sampling schedules at NFE = 250 are presented in Table 17. To assess the generality across samplers, we evaluated three different ones: SDE-DPM-Solver++(2M), PNDM, and DPM-Solver++(2M). As shown in the gray-shaded rows, using CRS-$v_x$ for training consistently improved multiple evaluation metrics, including FID, regardless of the sampler used.

Figure 12 shows the evolution of the FID score with 10k samples during training, evaluated using DPM-Solver++ (2M) with NFE = 250. In this experiment, we compare CRS-$v_x$ with shifted cosine, which was the strongest-performing baseline training schedule. The results demonstrate that using CRS-$v_x$ as the training schedule not only improves the final FID score but also leads to faster convergence throughout training.

Table 18 presents the effect of different sampling schedules at NFE = 250 and NFE = 50. When using SDE-DPM-Solver++(2M), CRS-$v_x + v_{\text{FID}}$ improved all four metrics at NFE = 250, achieving a new state-of-the-art FID score. With PNDM and DPM-Solver++(2M), differences among sampling schedules were minor at NFE = 250; however, at NFE = 50, performance became more sensitive to the sampling schedule. In this low-NFE regime, CRS-$v_x + v_{\text{FID}}$ achieved the best FID score.

Table 16: Conventional noise schedules evaluated in this study. $T_s$ denotes number of timesteps for sampling.

| Name | Phase | Noise schedule |
|---|---|---|
| Linear | Training | $\alpha_{\text{linear}}(t) = \text{LinearInterp1d}\left(\left\{\frac{i}{T}\right\}_{i=0,\dots,T}, \{\alpha_i\}_{i=0,\dots,T}\right),$ 
 $\alpha_0 = 1,$ 
 $\alpha_{1 \le i \le T} = \sqrt{1 - \tilde{\beta}_i}\alpha_{i-1},$ 
 $\tilde{\beta}_i = \tilde{\beta}_{\min} + \frac{\tilde{\beta}_{\max} - \tilde{\beta}_{\min}}{T-1}(i-1),$ 
 $T = 1000, \tilde{\beta}_{\min} = 10^{-4}, \tilde{\beta}_{\max} = 0.02.$ |
| | Sampling | $\left\{\alpha_{\text{linear}}\left(\frac{t}{T_s}\right) \middle| t = 0, \dots, T_s\right\}$ |
| Shifted cosine | Training | $\alpha_{\text{shifted}}(t) = \sqrt{\text{sigmoid}(\lambda(t))},$ 
 $\lambda(t) = -2\log\tan\left(\frac{\pi t}{2}\right) + 2\log\left(\frac{64}{d}\right),$ 
 where $d$ is resolution in diffusion process. |
| | Sampling | $\left\{\tilde{\alpha}_{\text{shifted}}\left(\frac{t}{T_s}\right) \middle| t = 0, \dots, T_s\right\},$ 
 $\tilde{\alpha}_{\text{shifted}}(t) = \alpha_{\min} + (\alpha_{\max} - \alpha_{\min})\alpha_{\text{shifted}}(t),$ 
 $\alpha_{\min} = 0.01, \alpha_{\max} = 1.0.$ |
| VDM++ | Training | $p(\lambda) \propto \mathbb{E}_{\boldsymbol{x}_0 \sim \mathcal{D}, \boldsymbol{\varepsilon} \sim \mathcal{N}(\boldsymbol{0}, \boldsymbol{I})}\left[\omega(\lambda)\|\boldsymbol{\varepsilon}_\theta(\boldsymbol{x}, \tilde{\alpha}(\lambda)) - \boldsymbol{\varepsilon}\|_2^2\right],$ 
 $\omega(\lambda) = \begin{cases} \max_\lambda \tilde{\omega}(\lambda), & \lambda < \operatorname{argmax}_\lambda \tilde{\omega}(\lambda) \\ \tilde{\omega}(\lambda), & \lambda \ge \operatorname{argmax}_\lambda \tilde{\omega}(\lambda), \end{cases}$ 
 $\tilde{\omega}(\lambda) = \mathcal{N}(\lambda; \lambda_\mu, \lambda_\sigma^2)(e^{-\lambda} + c^2),$ 
 $\mu_\lambda = 2.4, \sigma_\lambda = 2.4, c = 0.5.$ |
| | Sampling | Only for training. |
| EDM | Training | Not evaluated in this study. |
| | Sampling | $\alpha_0 = 1, \alpha_{t \ge 1} = \alpha_{\text{EDM}}\left(\frac{t}{T_s}\right),$ 
 $\alpha_{\text{EDM}}(t) = \sqrt{\text{sigmoid}(\lambda(t))},$ 
 $\lambda(t) = -2\log\sigma(t),$ 
 $\sigma(t) = \left\{\sigma_{\max}^{\frac{1}{\rho}} + \left(\sigma_{\min}^{\frac{1}{\rho}} - \sigma_{\max}^{\frac{1}{\rho}}\right)(1-t)\right\}^\rho,$ 
 $\sigma_{\max} = 80, \sigma_{\min} = 0.002, \rho = 7.$ |

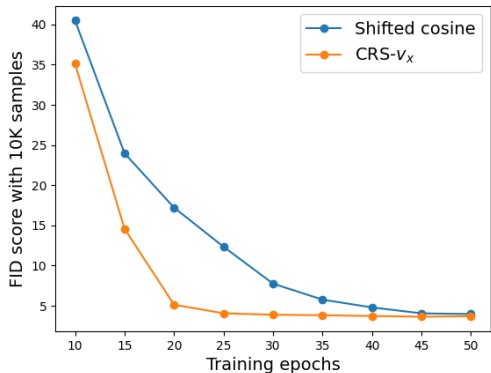

Figure 12: Evolution of the FID score with 10k samples during training on LSUN Horse 256×256 in the pixel-space diffusion model. Using CRS-$v_x$ as the training schedule not only improves the final FID score but also leads to faster convergence compared to shifted cosine. Evaluation is performed using DPM-Solver++(2M) with NFE = 250.

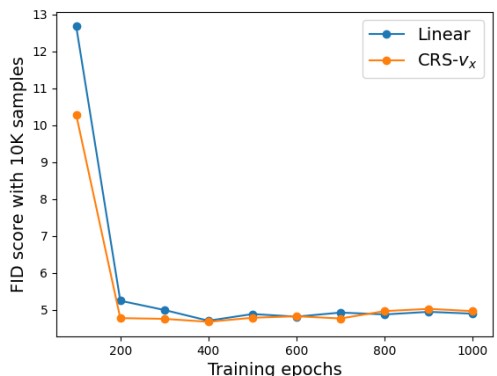

Figure 13: Evolution of the FID score with 10k samples during training on LSUN Church 256×256 in the latent-space diffusion model. CRS-$v_x$ achieves a slightly better best FID score and consistently lower FID scores before reaching the best value compared to the linear schedule. Evaluation is performed using DPM-Solver++(2M) with NFE = 30.

**LSUN Bedroom 256×256:** Since only FID scores are reported in the main text (Table 2), we include sFID, precision, and recall in Table 19 for reference.

Table 20 compares the FID scores of CRS-$v_x$ + $v_{\text{FID}}$ and EDM sampling under various values of the hyperparameter $\rho$. While the default setting for EDM uses $\rho = 7$, tuning $\rho$ can further improve its performance. However, even with the optimal $\rho$, EDM sampling did not outperform CRS-$v_x$ + $v_{\text{FID}}$ in terms of FID.

### F.2 Results on Latent-Space Diffusion Models

**LSUN Church 256×256:** The results showing the dependence on training and sampling schedules at NFE = 30, using stochastic and deterministic samplers, are summarized in Tables 21 and 22, respectively. To evaluate generality across samplers, we tested five different ones: Stochastic DDIM, SDE-DPM-Solver++(2M), DDIM, PNDM, and DPM-Solver++(2M). As shown in the gray-shaded rows, CRS-$v_x$ consistently improved FID scores across all samplers except for SDE-DPM-Solver++(2M).

Figure 13 shows the evolution of the FID score with 10k samples during training, evaluated using DPM-Solver++ (2M) with NFE = 30. In this experiment, we compare CRS-$v_x$ with linear, which was the strongest-performing baseline training schedule. Both training schedules reach their best FID scores at around 400 epochs, with CRS-$v_x$ achieving a slightly better best FID score. Furthermore, CRS-$v_x$ also showed consistently lower FID scores before reaching the best value, indicating its effectiveness in terms of convergence speed as well.

The effects of different sampling schedules are shown separately for stochastic and deterministic samplers in Tables 23 and 24, respectively. CRS-$v_x$ or CRS-$v_\varepsilon$ achieved the best FID scores in all cases except when using SDE-DPM-Solver++(2M) at NFE = 20. In most cases, multiple evaluation metrics—including sFID, precision, and recall—were simultaneously improved.

**LSUN Bedroom 256×256:** The results showing the dependence on sampling schedules using stochastic and deterministic samplers are presented in Tables 25 and 26, respectively. The training schedule was fixed to CRS-$v_x$. CRS achieved the best FID scores in all cases, except when using SDE-DPM-Solver++(2M) at NFE = 30 and NFE = 20. In most cases, multiple evaluation metrics were simultaneously improved.

FID scores evaluated at small NFE settings are listed in Table 27. Compared to the results reported in DPM-Solver-v3 (Zheng et al., 2023a), better FID scores are consistently achieved by using CRS to optimize both training and sampling schedules across all NFE values. As with other evaluations in this paper, we

set $\xi = 1$ by default for CRS-$v_x$ and CRS-$v_\varepsilon$, except at NFE = 5, where $\xi$ was manually tuned to further improve performance. In this experiment, we observed that the importance of tuning $\xi$ increases as the NFE decreases. We attribute this behavior to the discretization error introduced by the continuous-time approximation used in Eq. (7). A comprehensive set of evaluation metrics, including FID, sFID, precision, and recall, is reported in Table 28. CRS-based noise schedules consistently outperform the linear schedule across all metrics, except for recall at NFE = 20.

**ImageNet 256×256:** We demonstrated the effectiveness of CRS in class-conditional image generation on ImageNet. It is important to note that our experiments do not use classifier-free guidance (CFG), and therefore the FID scores are naturally higher than those obtained under CFG-enabled settings. However, as shown in Table 29, our model achieves better FID score than the LDM-4 results reported in the original LDM paper (Rombach et al., 2022). These observations indicate that our ImageNet results represent a reasonable level of performance for class-conditional generation without CFG, and provide a meaningful basis for evaluating the impact of CRS in this setting.

The results showing the dependence on training and sampling schedules at NFE = 30, using both stochastic and deterministic samplers, are summarized in Tables 30 and 31, respectively. As shown in the gray-shaded rows, CRS-$v_{\text{FID}}$ achieved the best performance for training schedule optimization, while CRS-$v_x$, which consistently performed well on other datasets, was the second-best. This result may be attributed to the fact that the feature extractor used in FID computation is pre-trained on ImageNet, which may advantage CRS-$v_{\text{FID}}$ in this specific setting.

The impact of different sampling schedules using stochastic and deterministic samplers is shown in Tables 32 and 33, respectively. Here, the training schedule was fixed to CRS-$v_x$, which outperformed conventional noise schedules by a significant margin on ImageNet and delivered strong results across a wide range of datasets. CRS achieved the best FID scores in all cases, except when using PNDM at NFE = 20 and DPM-Solver++(2M) at NFE = 30. In most settings, CRS-based schedules also led to improvements in other evaluation metrics, including sFID, precision, and recall.

## G   Generated Samples

Generated samples are shown in the following figures:

- LSUN Horse 256×256 in the pixel-space diffusion model: Figs. 14 and 15.

- LSUN Bedroom 256×256 in the pixel-space diffusion model: Figs. 16 and 17.

- LSUN Church 256×256 in the latent-space diffusion model: Fig. 18.

- LSUN Bedroom 256×256 in the latent-space diffusion model: Fig. 19.

- ImageNet 256×256 in the latent-space diffusion model: Fig. 20.

All models are trained using CRS-$v_x$ for the training schedule optimization.

Table 17: Performance of pixel-space diffusion models on LSUN Horse 256×256 at NFE = 250, evaluated under various training and sampling schedules. To assess generality across samplers, we used three different samplers: SDE-DPM-Solver++(2M), PNDM, and DPM-Solver++(2M). Gray-shaded rows indicate results for different training schedules (with fixed sampling), while unshaded rows show results for different sampling schedules (with fixed training). Bold values indicate the best performance for each metric. Note that "cosine" refers to the shifted cosine schedule.

| Sampler | Training schedule | Sampling schedule | Metrics | | | |
| --- | --- | --- | --- | --- | --- | --- |
| | | | FID ↓ | sFID ↓ | Precision ↑ | Recall ↑ |
| SDE-DPM-Solver++(2M) | Linear | Linear | 2.90 | 6.82 | 0.66 | **0.56** |
| | Cosine | | 2.95 | 6.43 | **0.67** | **0.56** |
| | VDM++ | | 3.82 | 7.40 | 0.66 | 0.54 |
| | CRS-$v_\varepsilon$ | | 2.91 | 6.56 | 0.66 | **0.56** |
| | CRS-$v_x$ | | **2.73** | **6.40** | **0.67** | **0.56** |
| | CRS-$v_x$ | Linear | 2.73 | 6.40 | 0.67 | **0.56** |
| | | Cosine | 3.13 | 7.09 | 0.67 | 0.54 |
| | | EDM | 2.09 | 6.16 | **0.69** | **0.56** |
| | | CRS-$v_\varepsilon$ | 2.87 | 6.71 | 0.66 | 0.56 |
| | | CRS-$v_x$ | 5.46 | 8.34 | 0.62 | 0.52 |
| | | CRS-$v_x + v_{\text{FID}}$ | **2.03** | **6.06** | **0.69** | **0.56** |
| PNDM | Linear | Linear | 4.06 | 6.23 | 0.57 | **0.61** |
| | Cosine | | 2.36 | 5.59 | 0.63 | 0.60 |
| | VDM++ | | 70.18 | 36.78 | 0.20 | 0.53 |
| | CRS-$v_{\text{FID}}$ | | 8.20 | 7.92 | 0.51 | 0.60 |
| | CRS-$v_x$ | | **2.09** | **5.43** | **0.65** | 0.58 |
| | CRS-$v_x$ | Linear | **2.09** | **5.43** | 0.65 | **0.58** |
| | | Cosine | 2.19 | 5.59 | **0.67** | 0.57 |
| | | EDM | 2.35 | 5.85 | **0.67** | 0.56 |
| | | CRS-$v_\varepsilon$ | 2.11 | 5.58 | 0.66 | **0.58** |
| | | CRS-$v_x$ | 2.50 | 5.94 | 0.64 | **0.58** |
| | | CRS-$v_x + v_{\text{FID}}$ | 2.12 | 5.50 | **0.67** | 0.57 |
| DPM-Solver++(2M) | Linear | Linear | 3.17 | 5.98 | 0.60 | **0.61** |
| | Cosine | | 2.31 | 5.63 | 0.63 | 0.60 |
| | VDM++ | | 47.63 | 29.77 | 0.26 | 0.56 |
| | CRS-$v_\varepsilon$ | | 7.12 | 7.45 | 0.54 | 0.60 |
| | CRS-$v_x$ | | **2.08** | **5.47** | **0.65** | 0.59 |
| | CRS-$v_x$ | Linear | **2.08** | **5.47** | 0.65 | **0.59** |
| | | Cosine | 2.18 | 5.72 | 0.66 | 0.58 |
| | | EDM | 2.35 | 5.85 | **0.67** | 0.56 |
| | | CRS-$v_x$ | 2.12 | 5.65 | 0.66 | 0.58 |
| | | CRS-$v_\varepsilon$ | 2.87 | 6.15 | 0.63 | 0.58 |
| | | CRS-$v_x + v_{\text{FID}}$ | 2.19 | 5.56 | **0.67** | 0.57 |

Table 18: Performance of pixel-space diffusion models on LSUN Horse 256×256 under different sampling schedules. Training schedule was fixed to CRS-$v_x$. Bold values indicate the best sampling schedule for each evaluation metric.

| Sampler | NFE | Sampling schedule | Metrics | | | |
|---|---|---|---|---|---|---|
| | | | FID ↓ | sFID ↓ | Precision ↑ | Recall ↑ |
| SDE-DPM-Solver++(2M) | 250 | Linear | 2.73 | 6.40 | 0.67 | 0.56 |
| | | Shifted cosine | 3.13 | 7.09 | 0.67 | 0.54 |
| | | EDM | 2.09 | 6.16 | 0.69 | 0.56 |
| | | CRS-$v_\varepsilon$ | 2.87 | 6.71 | 0.66 | 0.56 |
| | | CRS-$v_x$ | 5.46 | 8.34 | 0.62 | 0.52 |
| | | CRS-$v_x + v_{\text{FID}}$ | **2.03** | **6.06** | **0.69** | **0.56** |
| | 50 | Linear | 7.54 | 9.73 | 0.59 | 0.48 |
| | | Shifted cosine | 11.37 | 12.46 | 0.53 | 0.43 |
| | | EDM | **2.75** | **6.91** | **0.68** | **0.55** |
| | | CRS-$v_\varepsilon$ | 6.08 | 8.75 | 0.61 | 0.50 |
| | | CRS-$v_x$ | 12.99 | 12.59 | 0.50 | 0.41 |
| | | CRS-$v_x + v_{\text{FID}}$ | 3.84 | 8.16 | 0.67 | 0.52 |
| PNDM | 250 | Linear | **2.09** | **5.43** | 0.65 | **0.58** |
| | | Shifted cosine | 2.19 | 5.59 | **0.67** | 0.57 |
| | | EDM | 2.35 | 5.85 | **0.67** | 0.56 |
| | | CRS-$v_\varepsilon$ | 2.11 | 5.58 | 0.66 | **0.58** |
| | | CRS-$v_x$ | 2.50 | 5.94 | 0.64 | **0.58** |
| | | CRS-$v_x + v_{\text{FID}}$ | 2.12 | 5.50 | **0.67** | 0.57 |
| | 50 | Linear | 2.50 | 5.76 | 0.62 | **0.59** |
| | | Shifted cosine | 3.92 | 6.77 | 0.60 | 0.57 |
| | | EDM | 2.36 | 5.87 | **0.67** | 0.56 |
| | | CRS-$v_\varepsilon$ | 2.46 | 5.97 | 0.64 | 0.58 |
| | | CRS-$v_x$ | 6.03 | 7.32 | 0.57 | 0.56 |
| | | CRS-$v_x + v_{\text{FID}}$ | **2.04** | **5.60** | 0.66 | 0.58 |
| DPM-Solver++(2M) | 250 | Linear | **2.08** | **5.47** | 0.65 | **0.59** |
| | | Shifted Cosine | 2.18 | 5.72 | 0.66 | 0.58 |
| | | EDM | 2.35 | 5.85 | **0.67** | 0.56 |
| | | CRS-$v_\varepsilon$ | 2.12 | 5.65 | 0.66 | 0.58 |
| | | CRS-$v_x$ | 2.87 | 6.15 | 0.63 | 0.58 |
| | | CRS-$v_x + v_{\text{FID}}$ | 2.19 | 5.56 | **0.67** | 0.57 |
| | 50 | Linear | 3.17 | 5.98 | 0.60 | **0.59** |
| | | Shifted cosine | 5.27 | 7.12 | 0.57 | 0.56 |
| | | EDM | 2.43 | **5.87** | **0.67** | 0.56 |
| | | CRS-$v_\varepsilon$ | 3.10 | 6.18 | 0.62 | 0.58 |
| | | CRS-$v_x$ | 7.44 | 7.83 | 0.55 | 0.55 |
| | | CRS-$v_x + v_{\text{FID}}$ | **2.26** | 5.90 | 0.66 | 0.57 |

Table 19: Evaluation results on LSUN Bedroom 256×256 in pixel-space diffusion model. CRS-$v_x$ was used for optimizing training schedule.

| Sampler | NFE | Sampling schedule | Metrics | | | |
|---|---|---|---|---|---|---|
| | | | FID ↓ | sFID ↓ | Precision ↑ | Recall ↑ |
| DPM-Solver++(2M) | 10 | EDM | **4.73** | **7.75** | **0.53** | 0.51 |
| | | CRS-$v_x + v_{\text{FID}}$ | 4.88 | 8.67 | 0.49 | **0.52** |
| | 5 | EDM | 23.72 | 19.24 | 0.20 | 0.28 |
| | | CRS-$v_x + v_{\text{FID}}$ | **14.02** | **17.49** | **0.28** | **0.41** |
| UniPC | 10 | EDM | 4.94 | 8.20 | 0.53 | 0.40 |
| | | CRS-$v_x + v_{\text{FID}}$ | **3.30** | **6.60** | **0.55** | **0.53** |
| | 5 | EDM | 82.52 | 33.35 | 0.06 | 0.08 |
| | | CRS-$v_x + v_{\text{FID}}$ | **21.42** | **11.26** | **0.21** | **0.44** |

Table 20: Comparison of FID scores for CRS-$v_x + v_{\text{FID}}$ and EDM sampling on LSUN Bedroom 256×256 in the pixel-space diffusion model. Performance of EDM is reported under various values of the hyperparameter $\rho$. Even with the optimal $\rho$, EDM does not outperform CRS-$v_x + v_{\text{FID}}$.

| Sampler | Sampling schedule | NFE = 5 | NFE = 10 |
|---|---|---|---|
| DPM-Solver++(2M) | EDM: $\rho = 1$ | 190.32 | 102.10 |
| | EDM: $\rho = 3$ | 30.24 | 13.12 |
| | EDM: $\rho = 5$ | 22.97 | 7.92 |
| | EDM: $\rho = 7$ | 23.72 | **4.73** |
| | CRS-$v_x + v_{\text{FID}}$ | **14.02** | 4.88 |
| UniPC | EDM: $\rho = 1$ | 178.24 | 90.30 |
| | EDM: $\rho = 3$ | 23.45 | 4.43 |
| | EDM: $\rho = 5$ | 55.06 | 3.96 |
| | EDM: $\rho = 7$ | 82.52 | 4.94 |
| | CRS-$v_x + v_{\text{FID}}$ | **21.42** | **3.30** |

Table 21: Performance of latent-space diffusion models on LSUN Church 256×256 at NFE = 30, evaluated under various training and sampling schedules using stochastic samplers. Two samplers were used: Stochastic DDIM and SDE-DPM-Solver++(2M). Gray-shaded rows indicate variations in training schedules (with fixed sampling), while unshaded rows show variations in sampling schedules (with fixed training). Bold values indicate the best result for each evaluation metric.

| Sampler | Training schedule | Sampling schedule | Metrics | | | |
|---|---|---|---|---|---|---|
| | | | FID ↓ | sFID ↓ | Precision ↑ | Recall ↑ |
| Stochastic DDIM | Linear | Linear | 10.73 | 18.31 | 0.57 | 0.33 |
| | VDM++ | | 11.05 | 17.33 | 0.56 | **0.35** |
| | CRS-$v_{\text{FID}}$ | | 10.85 | 18.77 | 0.57 | 0.34 |
| | CRS-$v_\varepsilon$ | | 10.43 | 17.81 | 0.57 | 0.34 |
| | CRS-$v_x$ | | **9.82** | **16.39** | **0.58** | 0.34 |
| | CRS-$v_x$ | Linear | 9.82 | 16.39 | 0.58 | 0.34 |
| | | EDM | 21.73 | 29.33 | 0.43 | 0.22 |
| | | CRS-$v_{\text{FID}}$ | 7.36 | 13.04 | 0.60 | 0.39 |
| | | CRS-$v_\varepsilon$ | 7.53 | 13.73 | 0.61 | 0.38 |
| | | CRS-$v_x$ | **6.77** | **12.57** | **0.63** | **0.40** |
| SDE-DPM-Solver++(2M) | Linear | Linear | 3.49 | 10.19 | 0.63 | **0.57** |
| | VDM++ | | **3.33** | **9.79** | 0.63 | **0.57** |
| | CRS-$v_{\text{FID}}$ | | 3.50 | 10.69 | 0.63 | **0.57** |
| | CRS-$v_\varepsilon$ | | 3.57 | 10.26 | **0.64** | 0.56 |
| | CRS-$v_x$ | | 3.38 | 10.09 | **0.64** | 0.55 |
| | CRS-$v_x$ | Linear | 3.38 | 10.09 | **0.64** | 0.55 |
| | | EDM | 4.20 | 11.69 | 0.63 | 0.54 |
| | | CRS-$v_{\text{FID}}$ | 3.61 | 9.51 | 0.61 | 0.57 |
| | | CRS-$v_\varepsilon$ | **3.19** | 9.56 | 0.62 | **0.59** |
| | | CRS-$v_x$ | 3.20 | **9.14** | 0.62 | 0.58 |

Table 22: Performance of latent-space diffusion models on LSUN Church 256×256 at NFE = 30, evaluated under various training and sampling schedules using deterministic samplers. Three samplers were used: DDIM, PNDM, and DPM-Solver++(2M). Gray-shaded rows indicate variations in training schedules (with fixed sampling), while unshaded rows show variations in sampling schedules (with fixed training). Bold values indicate the best result for each evaluation metric.

| Sampler | Training schedule | Sampling schedule | Metrics FID ↓ | sFID ↓ | Precision ↑ | Recall ↑ |
|---|---|---|---|---|---|---|
| DDIM | Linear | Linear | 6.18 | 13.16 | **0.59** | 0.51 |
| | VDM++ | | 6.50 | 12.67 | 0.56 | **0.54** |
| | CRS-$v_{\mathrm{FID}}$ | | 6.06 | 13.69 | **0.59** | 0.51 |
| | CRS-$v_{\varepsilon}$ | | 6.05 | 12.49 | 0.58 | 0.50 |
| | CRS-$v_x$ | | **5.67** | **12.33** | **0.59** | 0.51 |
| | CRS-$v_x$ | Linear | 5.67 | 12.33 | **0.59** | 0.51 |
| | | EDM | 9.15 | 16.86 | 0.56 | 0.44 |
| | | CRS-$v_{\mathrm{FID}}$ | 5.35 | **10.70** | 0.57 | 0.53 |
| | | CRS-$v_{\varepsilon}$ | 4.95 | 11.01 | **0.59** | 0.53 |
| | | CRS-$v_x$ | **4.81** | 10.79 | **0.59** | **0.55** |
| PNDM | Linear | Linear | 3.74 | 10.48 | 0.60 | 0.58 |
| | VDM++ | | 3.88 | **10.20** | 0.59 | **0.59** |
| | CRS-$v_{\mathrm{FID}}$ | | 3.59 | 10.61 | **0.61** | **0.59** |
| | CRS-$v_{\varepsilon}$ | | 3.71 | 10.36 | 0.60 | **0.59** |
| | CRS-$v_x$ | | **3.56** | 10.37 | **0.61** | 0.58 |
| | CRS-$v_x$ | Linear | 3.56 | 10.37 | **0.61** | 0.58 |
| | | EDM | 3.88 | 10.76 | 0.60 | 0.57 |
| | | CRS-$v_{\mathrm{FID}}$ | 3.59 | **9.69** | 0.59 | **0.61** |
| | | CRS-$v_{\varepsilon}$ | **3.48** | 9.79 | 0.59 | 0.59 |
| | | CRS-$v_x$ | 3.62 | 10.15 | 0.59 | **0.61** |
| DPM-Solver++(2M) | Linear | Linear | 3.82 | 10.28 | **0.60** | 0.58 |
| | VDM++ | | 4.00 | **10.07** | 0.58 | **0.59** |
| | CRS-$v_{\mathrm{FID}}$ | | 3.78 | 10.51 | **0.60** | 0.58 |
| | CRS-$v_{\varepsilon}$ | | 3.87 | 10.30 | **0.60** | **0.59** |
| | CRS-$v_x$ | | **3.69** | 10.23 | **0.60** | **0.59** |
| | CRS-$v_x$ | Linear | 3.69 | 10.23 | **0.60** | 0.59 |
| | | EDM | 4.31 | 11.31 | **0.60** | 0.56 |
| | | CRS-$v_{\mathrm{FID}}$ | 3.98 | 9.61 | 0.58 | **0.60** |
| | | CRS-$v_{\varepsilon}$ | 3.63 | 9.91 | 0.59 | **0.60** |
| | | CRS-$v_x$ | **3.59** | **9.51** | 0.59 | **0.60** |

Table 23: Performance of latent-space diffusion models on LSUN Church 256×256 using stochastic samplers, evaluated under different sampling schedules. The training schedule was fixed to CRS-$v_x$. Bold values indicate the best sampling schedule for each evaluation metric.

| Sampler | NFE | Sampling schedule | Metrics FID ↓ | sFID ↓ | Precision ↑ | Recall ↑ |
|---|---|---|---|---|---|---|
| Stochastic DDIM | 50 | Linear | 6.23 | 11.94 | **0.65** | 0.43 |
| | | EDM | 10.05 | 17.14 | 0.60 | 0.34 |
| | | CRS-$v_{\mathrm{FID}}$ | 5.17 | **10.49** | **0.65** | 0.44 |
| | | CRS-$v_\varepsilon$ | 5.26 | 10.94 | **0.65** | 0.44 |
| | | CRS-$v_x$ | **4.82** | 11.91 | 0.62 | **0.51** |
| | 30 | Linear | 9.82 | 16.39 | 0.58 | 0.34 |
| | | EDM | 21.73 | 29.33 | 0.43 | 0.22 |
| | | CRS-$v_{\mathrm{FID}}$ | 7.36 | 13.04 | 0.60 | 0.39 |
| | | CRS-$v_\varepsilon$ | 7.53 | 13.73 | 0.61 | 0.38 |
| | | CRS-$v_x$ | **6.77** | **12.57** | **0.63** | **0.40** |
| | 20 | Linear | 18.24 | 25.36 | 0.45 | 0.24 |
| | | EDM | 48.46 | 52.53 | 0.25 | 0.11 |
| | | CRS-$v_{\mathrm{FID}}$ | 12.42 | 18.38 | 0.51 | 0.29 |
| | | CRS-$v_\varepsilon$ | 12.53 | 19.27 | 0.52 | 0.29 |
| | | CRS-$v_x$ | **10.66** | **16.54** | **0.55** | **0.32** |
| SDE-DPM-Solver++(2M) | 50 | Linear | 3.42 | 10.58 | 0.63 | 0.57 |
| | | EDM | 3.57 | 10.93 | **0.64** | 0.57 |
| | | CRS-$v_{\mathrm{FID}}$ | 3.32 | **9.19** | 0.62 | 0.58 |
| | | CRS-$v_\varepsilon$ | 3.14 | 9.65 | 0.63 | 0.58 |
| | | CRS-$v_x$ | **3.09** | 9.34 | 0.63 | **0.59** |
| | 30 | Linear | 3.38 | 10.09 | **0.64** | 0.55 |
| | | EDM | 4.20 | 11.69 | 0.63 | 0.54 |
| | | CRS-$v_{\mathrm{FID}}$ | 3.61 | 9.51 | 0.61 | 0.57 |
| | | CRS-$v_\varepsilon$ | **3.19** | 9.56 | 0.62 | **0.59** |
| | | CRS-$v_x$ | 3.20 | **9.14** | 0.62 | 0.58 |
| | 20 | Linear | **3.55** | 10.37 | **0.63** | 0.56 |
| | | EDM | 5.92 | 13.79 | 0.62 | 0.52 |
| | | CRS-$v_{\mathrm{FID}}$ | 4.39 | **9.33** | **0.63** | 0.48 |
| | | CRS-$v_\varepsilon$ | 3.58 | 9.93 | 0.61 | **0.57** |
| | | CRS-$v_x$ | 3.57 | 9.55 | 0.62 | 0.56 |

Table 24: Performance of latent-space diffusion models on LSUN Church 256×256 using deterministic samplers, evaluated under different sampling schedules. The training schedule was fixed to CRS-$v_x$. Bold values indicate the best sampling schedule for each evaluation metric.

| Sampler | NFE | Sampling schedule | Metrics | | | |
|---|---|---|---|---|---|---|
| | | | FID ↓ | sFID ↓ | Precision ↑ | Recall ↑ |
| DDIM | 50 | Linear | 4.52 | 11.03 | **0.60** | 0.54 |
| | | EDM | 5.95 | 13.37 | 0.59 | 0.50 |
| | | CRS-$v_{\text{FID}}$ | 4.28 | **10.07** | 0.59 | 0.56 |
| | | CRS-$v_{\varepsilon}$ | 4.17 | 10.66 | **0.60** | 0.56 |
| | | CRS-$v_x$ | **4.01** | 10.10 | **0.60** | **0.57** |
| | 30 | Linear | 5.67 | 12.33 | **0.59** | 0.51 |
| | | EDM | 9.15 | 16.86 | 0.56 | 0.44 |
| | | CRS-$v_{\text{FID}}$ | 5.35 | **10.70** | 0.57 | 0.53 |
| | | CRS-$v_{\varepsilon}$ | 4.95 | 11.01 | **0.59** | 0.53 |
| | | CRS-$v_x$ | **4.81** | 10.79 | **0.59** | **0.55** |
| | 20 | Linear | 7.85 | 14.66 | 0.56 | 0.47 |
| | | EDM | 15.74 | 22.86 | 0.48 | 0.34 |
| | | CRS-$v_{\text{FID}}$ | 7.47 | 12.74 | 0.54 | 0.48 |
| | | CRS-$v_{\varepsilon}$ | 6.49 | 12.55 | 0.56 | 0.49 |
| | | CRS-$v_x$ | **6.09** | **11.72** | **0.57** | **0.50** |
| PNDM | 50 | Linear | 3.52 | 10.42 | **0.61** | 0.58 |
| | | EDM | 3.71 | 10.59 | 0.60 | 0.57 |
| | | CRS-$v_{\text{FID}}$ | 3.49 | 9.95 | 0.60 | **0.61** |
| | | CRS-$v_{\varepsilon}$ | **3.48** | 10.18 | 0.60 | 0.59 |
| | | CRS-$v_x$ | **3.48** | **9.92** | 0.60 | 0.60 |
| | 30 | Linear | 3.56 | 10.37 | **0.61** | 0.58 |
| | | EDM | 3.88 | 10.76 | 0.60 | 0.57 |
| | | CRS-$v_{\text{FID}}$ | 3.59 | **9.69** | 0.59 | **0.61** |
| | | CRS-$v_{\varepsilon}$ | **3.48** | 9.79 | 0.59 | 0.59 |
| | | CRS-$v_x$ | 3.62 | 10.15 | 0.59 | **0.61** |
| | 20 | Linear | 3.74 | 10.24 | 0.60 | 0.59 |
| | | EDM | 4.57 | 11.49 | 0.59 | 0.56 |
| | | CRS-$v_{\text{FID}}$ | 4.06 | **9.54** | 0.57 | **0.61** |
| | | CRS-$v_{\varepsilon}$ | **3.64** | 9.98 | 0.59 | **0.61** |
| | | CRS-$v_x$ | 3.86 | **9.54** | **0.62** | 0.52 |
| DPM-Solver++(2M) | 50 | Linear | 3.62 | 10.44 | **0.61** | 0.58 |
| | | EDM | 3.90 | 10.85 | 0.60 | 0.56 |
| | | CRS-$v_{\text{FID}}$ | 3.59 | 9.68 | 0.59 | **0.60** |
| | | CRS-$v_{\varepsilon}$ | 3.46 | 9.82 | 0.60 | 0.59 |
| | | CRS-$v_x$ | **3.43** | **9.45** | 0.60 | 0.59 |
| | 30 | Linear | 3.69 | 10.23 | **0.60** | 0.59 |
| | | EDM | 4.31 | 11.31 | **0.60** | 0.56 |
| | | CRS-$v_{\text{FID}}$ | 3.98 | 9.61 | 0.58 | **0.60** |
| | | CRS-$v_{\varepsilon}$ | 3.63 | 9.91 | 0.59 | **0.60** |
| | | CRS-$v_x$ | **3.59** | **9.51** | 0.59 | **0.60** |
| | 20 | Linear | 3.92 | 10.03 | **0.59** | 0.59 |
| | | EDM | 5.25 | 12.19 | **0.59** | 0.55 |
| | | CRS-$v_{\text{FID}}$ | 4.95 | 9.92 | 0.56 | 0.58 |
| | | CRS-$v_{\varepsilon}$ | **3.89** | 9.73 | 0.58 | **0.60** |
| | | CRS-$v_x$ | 4.00 | **9.44** | 0.58 | 0.59 |

Table 25: Performance of latent-space diffusion models on LSUN Bedroom 256×256 using stochastic samplers, evaluated under different sampling schedules. The training schedule was fixed to CRS-$v_x$. Bold values indicate the best sampling schedule for each evaluation metric.

| Sampler | NFE | Sampling schedule | Metrics | | | |
|---|---|---|---|---|---|---|
| | | | FID ↓ | sFID ↓ | Precision ↑ | Recall ↑ |
| Stochastic DDIM | 50 | Linear | 3.95 | 8.98 | 0.56 | 0.48 |
| | | EDM | 7.43 | 16.05 | 0.54 | 0.38 |
| | | CRS-$v_{\text{FID}}$ | 3.09 | **7.17** | 0.57 | **0.50** |
| | | CRS-$v_\varepsilon$ | 3.25 | 7.81 | 0.57 | 0.49 |
| | | CRS-$v_x$ | **3.08** | 7.20 | **0.58** | **0.50** |
| | 30 | Linear | 7.28 | 15.70 | 0.53 | 0.38 |
| | | EDM | 17.03 | 33.60 | 0.40 | 0.26 |
| | | CRS-$v_{\text{FID}}$ | 5.17 | 10.62 | **0.57** | 0.43 |
| | | CRS-$v_\varepsilon$ | 5.58 | 12.33 | 0.56 | 0.42 |
| | | CRS-$v_x$ | **4.78** | **9.85** | 0.55 | **0.45** |
| | 20 | Linear | 14.53 | 29.25 | 0.42 | 0.28 |
| | | EDM | 38.01 | 68.62 | 0.21 | 0.13 |
| | | CRS-$v_{\text{FID}}$ | 9.18 | 16.67 | 0.50 | 0.35 |
| | | CRS-$v_\varepsilon$ | 9.70 | 19.40 | 0.49 | 0.34 |
| | | CRS-$v_x$ | **8.11** | **15.96** | **0.51** | **0.36** |
| SDE-DPM-Solver++(2M) | 50 | Linear | 2.47 | 6.44 | 0.57 | 0.54 |
| | | EDM | 2.48 | 6.32 | 0.57 | 0.54 |
| | | CRS-$v_{\text{FID}}$ | **2.21** | 5.87 | 0.57 | 0.54 |
| | | CRS-$v_\varepsilon$ | 2.22 | 5.76 | 0.57 | **0.55** |
| | | CRS-$v_x$ | 2.23 | **5.67** | 0.57 | 0.54 |
| | 30 | Linear | **2.45** | 6.70 | 0.56 | **0.55** |
| | | EDM | 2.90 | 7.41 | **0.57** | 0.52 |
| | | CRS-$v_{\text{FID}}$ | 2.46 | 6.19 | **0.57** | 0.54 |
| | | CRS-$v_\varepsilon$ | 2.50 | 5.93 | **0.57** | 0.53 |
| | | CRS-$v_x$ | 2.48 | **5.73** | 0.56 | 0.54 |
| | 20 | Linear | **2.78** | 6.79 | 0.56 | **0.53** |
| | | EDM | 4.00 | 9.57 | 0.55 | 0.50 |
| | | CRS-$v_{\text{FID}}$ | 3.19 | 6.70 | 0.55 | **0.53** |
| | | CRS-$v_\varepsilon$ | 2.87 | **6.22** | **0.57** | **0.53** |
| | | CRS-$v_x$ | 3.07 | 6.34 | 0.56 | 0.52 |

Table 26: Performance of latent-space diffusion models on LSUN Bedroom 256×256 using deterministic samplers, evaluated under different sampling schedules. The training schedule was fixed to CRS-$v_x$. Bold values indicate the best sampling schedule for each evaluation metric.

| Sampler | NFE | Sampling schedule | Metrics | | | |
|---|---|---|---|---|---|---|
| | | | FID ↓ | sFID ↓ | Precision ↑ | Recall ↑ |
| DDIM | 50 | Linear | 3.02 | 6.59 | 0.53 | 0.54 |
| | | EDM | 4.17 | 7.64 | 0.52 | 0.51 |
| | | CRS-$v_{\text{FID}}$ | 2.66 | 6.03 | **0.54** | **0.55** |
| | | CRS-$v_\varepsilon$ | 2.71 | 6.03 | **0.54** | **0.55** |
| | | CRS-$v_x$ | **2.62** | **5.86** | **0.54** | **0.55** |
| | 30 | Linear | 3.76 | 7.50 | 0.53 | 0.51 |
| | | EDM | 6.81 | 9.98 | 0.47 | 0.46 |
| | | CRS-$v_{\text{FID}}$ | 3.23 | 6.61 | 0.53 | **0.53** |
| | | CRS-$v_\varepsilon$ | 3.22 | 6.65 | 0.53 | **0.53** |
| | | CRS-$v_x$ | **3.10** | **6.43** | 0.54 | **0.53** |
| | 20 | Linear | 5.38 | 9.27 | 0.49 | 0.48 |
| | | EDM | 12.57 | 14.55 | 0.38 | 0.40 |
| | | CRS-$v_{\text{FID}}$ | 4.39 | 7.86 | **0.51** | 0.49 |
| | | CRS-$v_\varepsilon$ | 4.28 | 7.97 | **0.51** | 0.49 |
| | | CRS-$v_x$ | **4.10** | **7.54** | **0.51** | **0.51** |
| PNDM | 50 | Linear | 2.51 | 6.02 | **0.54** | 0.56 |
| | | EDM | 2.64 | 6.16 | 0.53 | **0.57** |
| | | CRS-$v_{\text{FID}}$ | 2.50 | 5.66 | **0.54** | **0.57** |
| | | CRS-$v_\varepsilon$ | **2.44** | 5.73 | **0.54** | **0.57** |
| | | CRS-$v_x$ | 2.45 | **5.61** | **0.54** | 0.56 |
| | 30 | Linear | 2.57 | 6.03 | **0.54** | 0.56 |
| | | EDM | 2.93 | 6.45 | 0.52 | **0.57** |
| | | CRS-$v_{\text{FID}}$ | 2.64 | 5.59 | 0.53 | **0.57** |
| | | CRS-$v_\varepsilon$ | **2.48** | 5.58 | **0.54** | 0.56 |
| | | CRS-$v_x$ | 2.65 | **5.53** | **0.54** | **0.57** |
| | 20 | Linear | 2.77 | 6.13 | 0.53 | **0.57** |
| | | EDM | 3.76 | 7.25 | 0.52 | 0.54 |
| | | CRS-$v_{\text{FID}}$ | 3.02 | 5.66 | 0.52 | **0.57** |
| | | CRS-$v_\varepsilon$ | **2.71** | **5.53** | 0.54 | 0.56 |
| | | CRS-$v_x$ | 3.63 | 6.23 | **0.61** | **0.57** |
| DPM-Solver++(2M) | 50 | Linear | 2.59 | 6.15 | **0.54** | 0.56 |
| | | EDM | 2.86 | 6.44 | 0.52 | **0.57** |
| | | CRS-$v_{\text{FID}}$ | 2.46 | 5.63 | **0.54** | 0.56 |
| | | CRS-$v_\varepsilon$ | 2.46 | 5.63 | **0.54** | 0.56 |
| | | CRS-$v_x$ | **2.37** | **5.42** | **0.54** | 0.56 |
| | 30 | Linear | 2.68 | 6.18 | **0.54** | **0.56** |
| | | EDM | 3.40 | 7.09 | 0.51 | **0.56** |
| | | CRS-$v_{\text{FID}}$ | **2.60** | 5.63 | **0.54** | **0.56** |
| | | CRS-$v_\varepsilon$ | **2.60** | 5.53 | **0.54** | **0.56** |
| | | CRS-$v_x$ | 2.63 | **5.45** | **0.54** | 0.55 |
| | 20 | Linear | 2.89 | 6.24 | 0.53 | **0.56** |
| | | EDM | 4.14 | 8.07 | 0.52 | 0.53 |
| | | CRS-$v_{\text{FID}}$ | 3.03 | 5.81 | 0.52 | **0.56** |
| | | CRS-$v_\varepsilon$ | **2.88** | **5.59** | 0.53 | **0.56** |
| | | CRS-$v_x$ | 3.00 | 5.63 | **0.54** | 0.55 |

Table 27: FID scores on LSUN Bedroom 256×256 in latent-space diffusion models at small NFE settings. Bold values indicate the best FID score at each NFE. Results for rows other than "Ours" are taken from Zheng et al. (2023a). As with other evaluations in this paper, we set $\xi = 1$ by default for CRS-$v_x$ and CRS-$v_\varepsilon$, except at NFE = 5, where $\xi$ was manually tuned to further improve performance. In this experiment, we observed that the importance of tuning $\xi$ increases as the NFE decreases. We attribute this behavior to the discretization error introduced by the continuous-time approximation used in Eq. (7).

| Model | Sampler | Sampling schedule | FID ↓ | | |
| --- | --- | --- | --- | --- | --- |
| | | | NFE = 5 | NFE = 10 | NFE = 20 |
| LDM | DPM-Solver++ | Linear | 18.59 | 3.63 | 3.16 |
| | UniPC | | 12.24 | 3.56 | 3.07 |
| | DPM-Solver-v3 Zheng et al. (2023a) | | 7.54 | 3.16 | 3.05 |
| Ours | iPNDM | Linear | 10.81 | 3.71 | 2.64 |
| | | CRS-$v_\varepsilon$ | 7.88 | **3.12** | **2.48** |
| | | CRS-$v_x$ | 8.07 | 3.45 | 2.50 |
| | | CRS-$v_\varepsilon$: $\xi = 1.4$ | 7.64 | – | – |
| | | CRS-$v_x$: $\xi = 2.2$ | **6.96** | – | – |

Table 28: Detailed evaluation results on LSUN Bedroom 256×256 in latent-space diffusion models at small NFE settings. Bold values indicate the best sampling schedule for each metric.

| Sampler | NFE | Sampling schedule | Metrics | | | |
| --- | --- | --- | --- | --- | --- | --- |
| | | | FID ↓ | sFID ↓ | Precision ↑ | Recall ↑ |
| iPNDM | 20 | Linear | 2.64 | 6.15 | 0.53 | **0.57** |
| | | CRS-$v_\varepsilon$ | **2.48** | 5.48 | 0.54 | 0.56 |
| | | CRS-$v_x$ | 2.50 | **5.41** | **0.55** | 0.56 |
| | 10 | Linear | 3.71 | 6.88 | 0.52 | 0.54 |
| | | CRS-$v_\varepsilon$ | **3.12** | **5.69** | 0.52 | **0.56** |
| | | CRS-$v_x$ | 3.45 | 6.15 | **0.54** | 0.53 |
| | 5 | Linear | 10.81 | 13.82 | 0.42 | 0.49 |
| | | CRS-$v_\varepsilon$ | 7.88 | 9.84 | **0.45** | 0.50 |
| | | CRS-$v_x$ | 8.07 | 10.43 | **0.45** | 0.48 |
| | | CRS-$v_\varepsilon$: $\xi = 1.4$ | 7.64 | 9.23 | **0.45** | **0.52** |
| | | CRS-$v_x$: $\xi = 2.2$ | **6.96** | **8.65** | **0.45** | 0.49 |

Table 29: Comparison between our trained model and LDM-4 (Rombach et al., 2022) on ImageNet under class-conditional generation without CFG. Nparams denotes the number of parameters in the U-Net model. Our model, trained with CRS-$v_x$ as the training schedule, achieves a better FID score than LDM-4 despite having fewer U-Net parameters.

| Model | Training schedule | Sampling schedule | Sampler | NFE | FID(↓) | Nparams |
| --- | --- | --- | --- | --- | --- | --- |
| LDM-4 | Linear | Linear | Stochastic DDIM | 250 | 10.56 | 400M |
| Ours | CRS-$v_x$ | | | | 9.67 | 296M |

Table 30: Performance of latent-space diffusion models on ImageNet 256×256 at NFE = 30, evaluated under various training and sampling schedules using stochastic samplers. Two samplers were used: Stochastic DDIM and SDE-DPM-Solver++(2M). Gray-shaded rows indicate variations in training schedules (with fixed sampling), while unshaded rows show variations in sampling schedules (with fixed training). Bold values indicate the best result for each evaluation metric.

| Sampler | Training schedule | Sampling schedule | Metrics FID ↓ | sFID ↓ | Precision ↑ | Recall ↑ |
|---|---|---|---|---|---|---|
| Stochastic DDIM | Linear | Linear | 27.31 | 21.39 | 0.46 | 0.55 |
| | VDM++ | | 29.00 | **17.67** | 0.46 | **0.57** |
| | CRS-$v_{\mathrm{FID}}$ | | **25.50** | 21.41 | **0.48** | 0.56 |
| | CRS-$v_\varepsilon$ | | 26.97 | 20.71 | 0.47 | 0.56 |
| | CRS-$v_x$ | | 25.83 | 19.92 | 0.47 | **0.57** |
| | CRS-$v_x$ | Linear | 25.83 | 19.92 | 0.47 | 0.57 |
| | | EDM | 49.72 | 50.08 | 0.31 | 0.51 |
| | | CRS-$v_{\mathrm{FID}}$ | 20.85 | 13.01 | 0.52 | 0.56 |
| | | CRS-$v_\varepsilon$ | 20.43 | 13.14 | 0.53 | 0.58 |
| | | CRS-$v_x$ | **17.88** | **10.14** | **0.55** | **0.59** |
| SDE-DPM-Solver++(2M) | Linear | Linear | 11.12 | 5.68 | **0.62** | 0.62 |
| | VDM++ | | 29.00 | 17.67 | 0.46 | 0.57 |
| | CRS-$v_{\mathrm{FID}}$ | | **10.22** | 5.81 | **0.62** | **0.64** |
| | CRS-$v_\varepsilon$ | | 11.18 | 5.84 | **0.62** | 0.63 |
| | CRS-$v_x$ | | 10.25 | **5.64** | **0.62** | **0.64** |
| | CRS-$v_x$ | Linear | 10.25 | 5.64 | 0.62 | **0.64** |
| | | EDM | 11.52 | 6.13 | 0.60 | **0.64** |
| | | CRS-$v_{\mathrm{FID}}$ | 10.49 | 5.39 | **0.63** | 0.63 |
| | | CRS-$v_\varepsilon$ | **10.24** | **5.22** | **0.63** | 0.63 |
| | | CRS-$v_x$ | 10.56 | 5.32 | **0.63** | 0.63 |

Table 31: Performance of latent-space diffusion models on ImageNet 256×256 at NFE = 30, evaluated under various training and sampling schedules using deterministic samplers. Three samplers were used: DDIM, PNDM, and DPM-Solver++(2M). Gray-shaded rows indicate variations in training schedules (with fixed sampling), while unshaded rows show variations in sampling schedules (with fixed training). Bold values indicate the best result for each evaluation metric.

| Sampler | Training schedule | Sampling schedule | Metrics FID ↓ | sFID ↓ | Precision ↑ | Recall ↑ |
|---|---|---|---|---|---|---|
| DDIM | Linear | Linear | 17.33 | 6.79 | 0.55 | 0.62 |
| | VDM++ | | 19.53 | **5.97** | 0.54 | **0.64** |
| | CRS-$v_{\mathrm{FID}}$ | | **15.52** | 6.95 | **0.56** | 0.63 |
| | CRS-$v_{\varepsilon}$ | | 17.02 | 6.94 | 0.55 | 0.63 |
| | CRS-$v_x$ | | 16.28 | 6.27 | **0.56** | **0.64** |
| | CRS-$v_x$ | Linear | 16.28 | 6.27 | 0.56 | **0.64** |
| | | EDM | 22.17 | 9.22 | 0.49 | 0.62 |
| | | CRS-$v_{\mathrm{FID}}$ | 15.51 | 6.72 | 0.56 | 0.63 |
| | | CRS-$v_{\varepsilon}$ | **15.26** | 5.79 | **0.57** | **0.64** |
| | | CRS-$v_x$ | 15.28 | **5.67** | **0.57** | **0.64** |
| PNDM | Linear | Linear | 13.21 | 5.35 | **0.59** | 0.65 |
| | VDM++ | | 16.01 | 6.31 | 0.57 | **0.66** |
| | CRS-$v_{\mathrm{FID}}$ | | **11.42** | **5.31** | **0.59** | **0.66** |
| | CRS-$v_{\varepsilon}$ | | 12.69 | 5.45 | **0.59** | 0.65 |
| | CRS-$v_x$ | | 12.29 | 5.38 | **0.59** | **0.66** |
| | CRS-$v_x$ | Linear | 12.29 | 5.38 | **0.59** | **0.66** |
| | | EDM | 12.86 | 5.62 | 0.58 | **0.66** |
| | | CRS-$v_{\mathrm{FID}}$ | **12.18** | **5.21** | **0.59** | 0.65 |
| | | CRS-$v_{\varepsilon}$ | 12.57 | 5.24 | **0.59** | **0.66** |
| | | CRS-$v_x$ | 12.65 | 5.38 | **0.59** | **0.66** |
| DPM-Solver++(2M) | Linear | Linear | 13.87 | 5.41 | **0.59** | 0.64 |
| | VDM++ | | 16.97 | 6.45 | 0.56 | **0.66** |
| | CRS-$v_{\mathrm{FID}}$ | | **12.09** | **5.40** | **0.59** | 0.65 |
| | CRS-$v_{\varepsilon}$ | | 13.41 | 5.54 | 0.58 | 0.65 |
| | CRS-$v_x$ | | 12.92 | 5.46 | **0.59** | **0.66** |
| | CRS-$v_x$ | Linear | **12.92** | 5.46 | **0.59** | **0.66** |
| | | EDM | 13.67 | 5.79 | 0.58 | **0.66** |
| | | CRS-$v_{\mathrm{FID}}$ | 13.02 | 5.19 | **0.59** | 0.65 |
| | | CRS-$v_{\varepsilon}$ | 13.02 | 5.17 | **0.59** | 0.65 |
| | | CRS-$v_x$ | 13.33 | **5.13** | **0.59** | 0.65 |

Table 32: Performance of latent-space diffusion models on ImageNet 256×256 using stochastic samplers, evaluated under different sampling schedules. The training schedule was fixed to CRS-$v_x$. Bold values indicate the best sampling schedule for each evaluation metric.

| Sampler | NFE | Sampling schedule | Metrics | | | |
|---|---|---|---|---|---|---|
| | | | FID ↓ | sFID ↓ | Precision ↑ | Recall ↑ |
| Stochastic DDIM | 50 | Linear | 15.61 | 8.97 | 0.57 | 0.60 |
| | | EDM | 25.66 | 19.17 | 0.46 | 0.57 |
| | | CRS-$v_{\text{FID}}$ | 13.67 | 6.86 | **0.60** | 0.60 |
| | | CRS-$v_\varepsilon$ | 13.44 | 7.56 | 0.59 | **0.61** |
| | | CRS-$v_x$ | **13.05** | **6.45** | **0.60** | **0.61** |
| | 30 | Linear | 25.83 | 19.92 | 0.47 | 0.57 |
| | | EDM | 49.72 | 50.08 | 0.31 | 0.51 |
| | | CRS-$v_{\text{FID}}$ | 20.85 | 13.01 | 0.52 | 0.56 |
| | | CRS-$v_\varepsilon$ | 20.43 | 13.14 | 0.53 | 0.58 |
| | | CRS-$v_x$ | **17.88** | **10.14** | **0.55** | **0.59** |
| | 20 | Linear | 44.15 | 42.96 | 0.34 | 0.53 |
| | | EDM | 78.05 | 99.25 | 0.20 | 0.44 |
| | | CRS-$v_{\text{FID}}$ | 30.28 | 24.26 | 0.44 | 0.54 |
| | | CRS-$v_\varepsilon$ | 31.86 | 26.34 | 0.43 | 0.55 |
| | | CRS-$v_x$ | **27.20** | **20.09** | **0.47** | **0.56** |
| SDE-DPM-DPM-Solver++(2M) | 50 | Linear | 9.96 | 5.62 | 0.63 | **0.64** |
| | | EDM | 9.72 | 5.35 | 0.63 | **0.64** |
| | | CRS-$v_{\text{FID}}$ | 10.26 | 5.43 | 0.63 | 0.63 |
| | | CRS-$v_\varepsilon$ | **9.68** | 5.33 | 0.63 | **0.64** |
| | | CRS-$v_x$ | 10.11 | **5.32** | 0.63 | **0.64** |
| | 30 | Linear | 10.25 | 5.64 | 0.62 | **0.64** |
| | | EDM | 11.52 | 6.13 | 0.60 | **0.64** |
| | | CRS-$v_{\text{FID}}$ | 10.49 | 5.39 | **0.63** | 0.63 |
| | | CRS-$v_\varepsilon$ | **10.24** | **5.22** | **0.63** | 0.63 |
| | | CRS-$v_x$ | 10.56 | 5.32 | **0.63** | 0.63 |
| | 20 | Linear | 11.47 | 5.81 | 0.61 | **0.63** |
| | | EDM | 16.26 | 9.38 | 0.55 | **0.63** |
| | | CRS-$v_{\text{FID}}$ | 11.10 | 5.50 | **0.63** | 0.62 |
| | | CRS-$v_\varepsilon$ | **10.83** | 5.36 | 0.62 | **0.63** |
| | | CRS-$v_x$ | 11.64 | **5.26** | **0.63** | 0.62 |

Table 33: Performance of latent-space diffusion models on ImageNet 256×256 using deterministic samplers, evaluated under different sampling schedules. The training schedule was fixed to CRS-$v_x$. Bold values indicate the best sampling schedule for each evaluation metric.

| Sampler | NFE | Sampling schedule | Metrics | | | |
| --- | --- | --- | --- | --- | --- | --- |
| | | | FID ↓ | sFID ↓ | Precision ↑ | Recall ↑ |
| DDIM | 50 | Linear | 13.91 | 5.37 | 0.58 | **0.65** |
| | | EDM | 16.25 | 6.24 | 0.55 | **0.65** |
| | | CRS-$v_{\text{FID}}$ | **13.45** | 5.36 | 0.58 | 0.64 |
| | | CRS-$v_\varepsilon$ | 13.64 | 5.20 | 0.58 | **0.65** |
| | | CRS-$v_x$ | 13.75 | **5.12** | **0.59** | **0.65** |
| | 30 | Linear | 16.28 | 6.27 | 0.56 | **0.64** |
| | | EDM | 22.17 | 9.22 | 0.49 | 0.62 |
| | | CRS-$v_{\text{FID}}$ | 15.51 | 6.72 | 0.56 | 0.63 |
| | | CRS-$v_\varepsilon$ | **15.26** | 5.79 | **0.57** | **0.64** |
| | | CRS-$v_x$ | 15.28 | **5.67** | **0.57** | **0.64** |
| | 20 | Linear | 20.49 | 8.82 | 0.51 | 0.62 |
| | | EDM | 33.08 | 15.78 | 0.40 | 0.60 |
| | | CRS-$v_{\text{FID}}$ | 18.43 | 7.52 | 0.54 | **0.63** |
| | | CRS-$v_\varepsilon$ | 18.30 | 7.57 | 0.54 | **0.63** |
| | | CRS-$v_x$ | **18.16** | **7.05** | **0.55** | 0.62 |
| PNDM | 50 | Linear | 12.10 | 5.32 | 0.59 | **0.66** |
| | | EDM | 12.53 | 5.49 | 0.59 | **0.66** |
| | | CRS-$v_{\text{FID}}$ | **11.86** | **5.20** | 0.59 | 0.65 |
| | | CRS-$v_\varepsilon$ | 12.35 | 5.28 | 0.59 | **0.66** |
| | | CRS-$v_x$ | 12.44 | 5.28 | 0.59 | **0.66** |
| | 30 | Linear | 12.29 | 5.38 | **0.59** | **0.66** |
| | | EDM | 12.86 | 5.62 | 0.58 | **0.66** |
| | | CRS-$v_{\text{FID}}$ | **12.18** | **5.21** | **0.59** | 0.65 |
| | | CRS-$v_\varepsilon$ | 12.57 | 5.24 | **0.59** | **0.66** |
| | | CRS-$v_x$ | 12.65 | 5.38 | **0.59** | **0.66** |
| | 20 | Linear | **12.83** | 5.52 | **0.59** | **0.66** |
| | | EDM | 13.99 | 5.99 | 0.58 | **0.66** |
| | | CRS-$v_{\text{FID}}$ | 12.97 | 5.38 | 0.58 | **0.66** |
| | | CRS-$v_\varepsilon$ | 12.92 | **5.30** | **0.59** | 0.65 |
| | | CRS-$v_x$ | 13.63 | 6.10 | 0.57 | **0.66** |
| DPM-Solver++(2M) | 50 | Linear | 12.42 | 5.41 | 0.59 | **0.66** |
| | | EDM | 12.90 | 5.57 | 0.59 | **0.66** |
| | | CRS-$v_{\text{FID}}$ | **12.27** | **5.15** | 0.59 | 0.65 |
| | | CRS-$v_\varepsilon$ | 12.64 | 5.23 | 0.59 | **0.66** |
| | | CRS-$v_x$ | 12.75 | 5.21 | 0.59 | 0.65 |
| | 30 | Linear | **12.92** | 5.46 | **0.59** | **0.66** |
| | | EDM | 13.67 | 5.79 | 0.58 | **0.66** |
| | | CRS-$v_{\text{FID}}$ | 13.02 | 5.19 | **0.59** | 0.65 |
| | | CRS-$v_\varepsilon$ | 13.02 | 5.17 | **0.59** | 0.65 |
| | | CRS-$v_x$ | 13.33 | **5.13** | **0.59** | 0.65 |
| | 20 | Linear | 13.71 | 5.45 | 0.58 | 0.65 |
| | | EDM | 15.31 | 6.23 | 0.57 | **0.66** |
| | | CRS-$v_{\text{FID}}$ | 14.10 | 5.25 | **0.59** | 0.65 |
| | | CRS-$v_\varepsilon$ | **13.70** | 5.13 | **0.59** | 0.65 |
| | | CRS-$v_x$ | 14.16 | **5.09** | **0.59** | 0.64 |

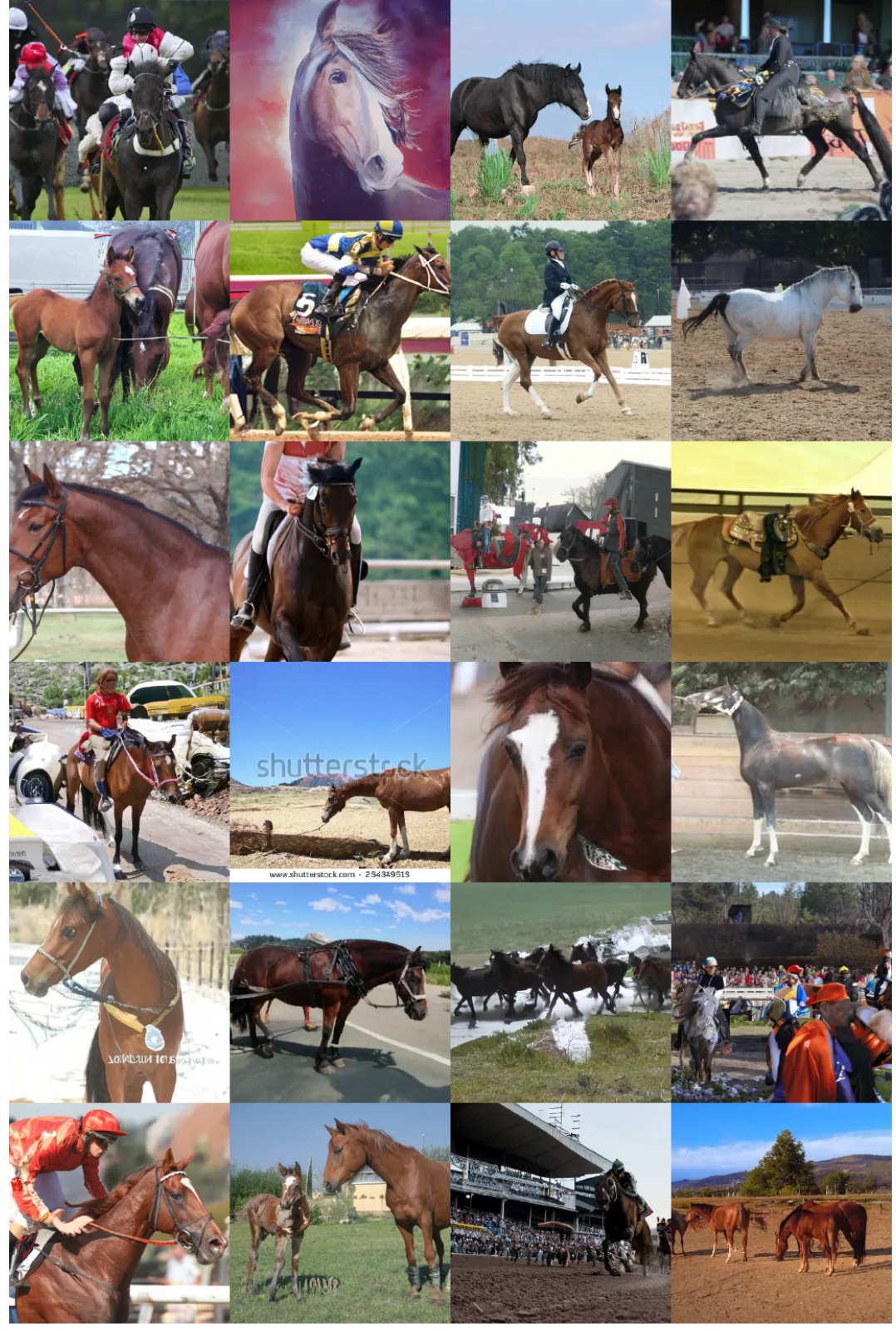

Figure 14: Generated samples of LSUN Horse 256×256 using a pixel-space diffusion model at NFE = 250 (FID = 2.03). The training and sampling schedules were optimized using CRS-$v_x$ and CRS-$v_x + v_{\mathrm{FID}}$, respectively. SDE-DPM-Solver++(2M) was used as the sampler.

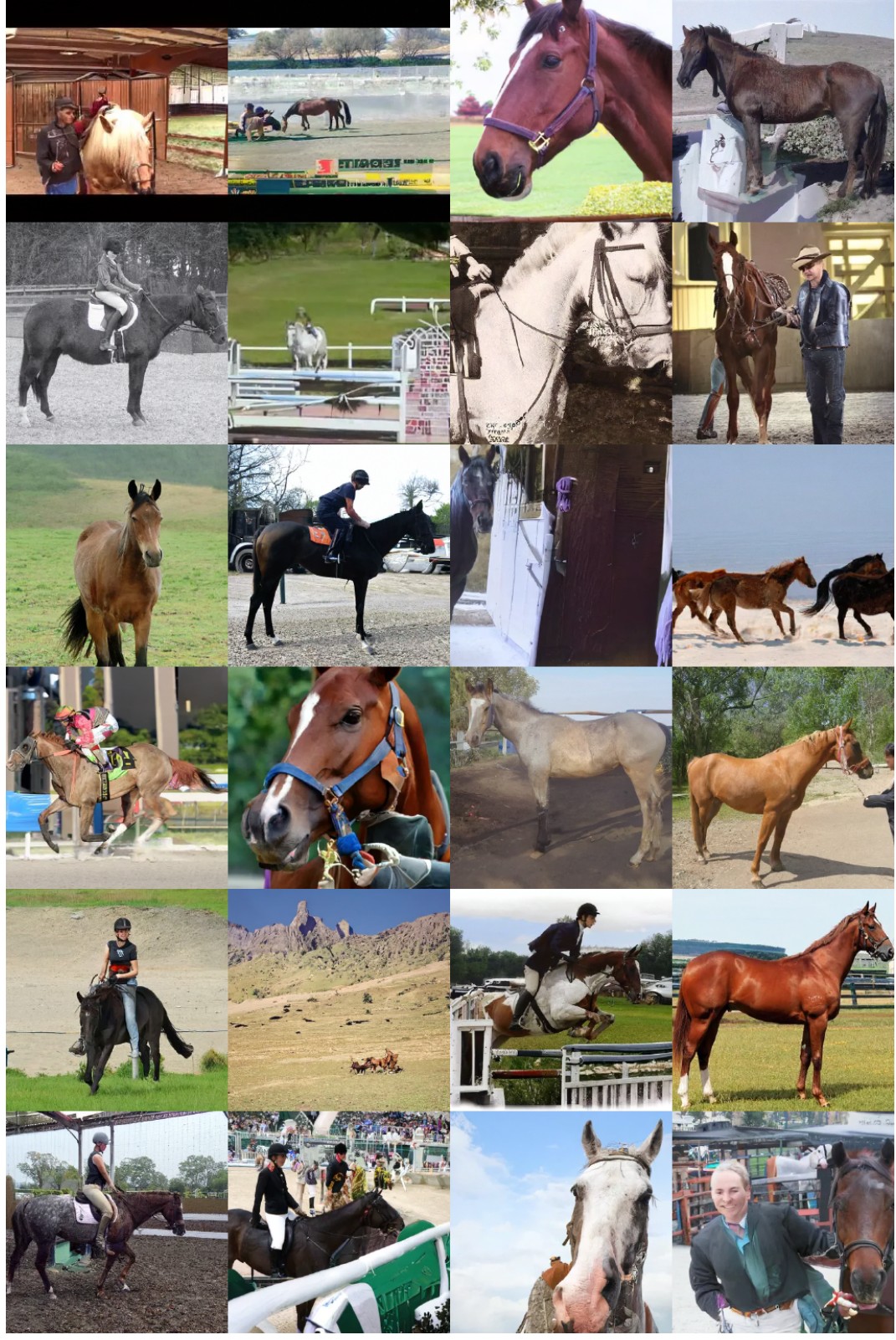

Figure 15: Generated samples of LSUN Horse $256\times256$ using a pixel-space diffusion model at NFE = 50 (FID = 2.04). The training and sampling schedules were optimized using CRS-$v_x$ and CRS-$v_x + v_{\mathrm{FID}}$, respectively. PNDM was used as the sampler.

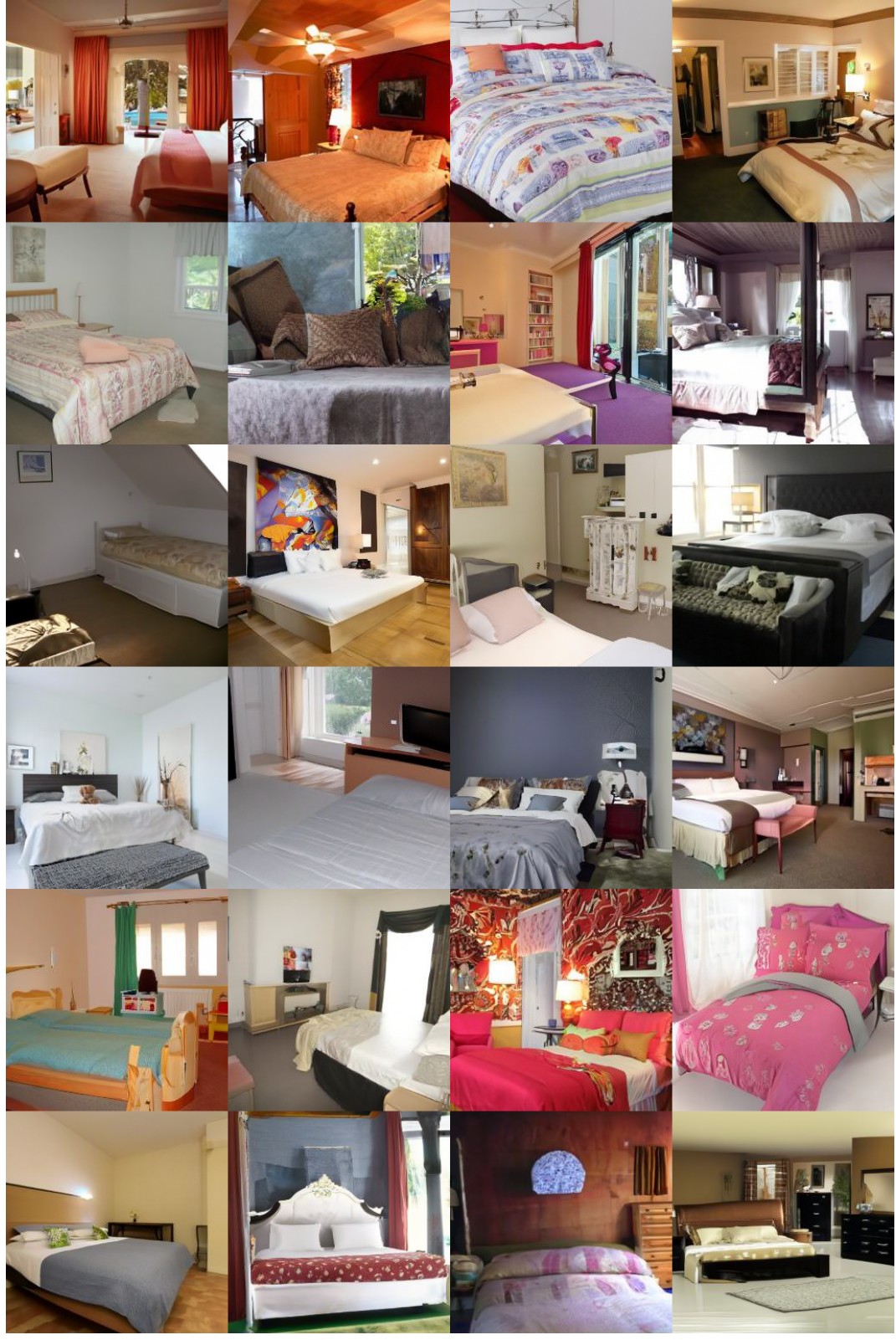

Figure 16: Generated samples of LSUN Bedroom 256×256 using a pixel-space diffusion model at NFE = 10 (FID = 3.30). The training and sampling schedules were optimized using CRS-$v_x$ and CRS-$v_x + v_{\text{FID}}$, respectively. UniPC was used as the sampler.

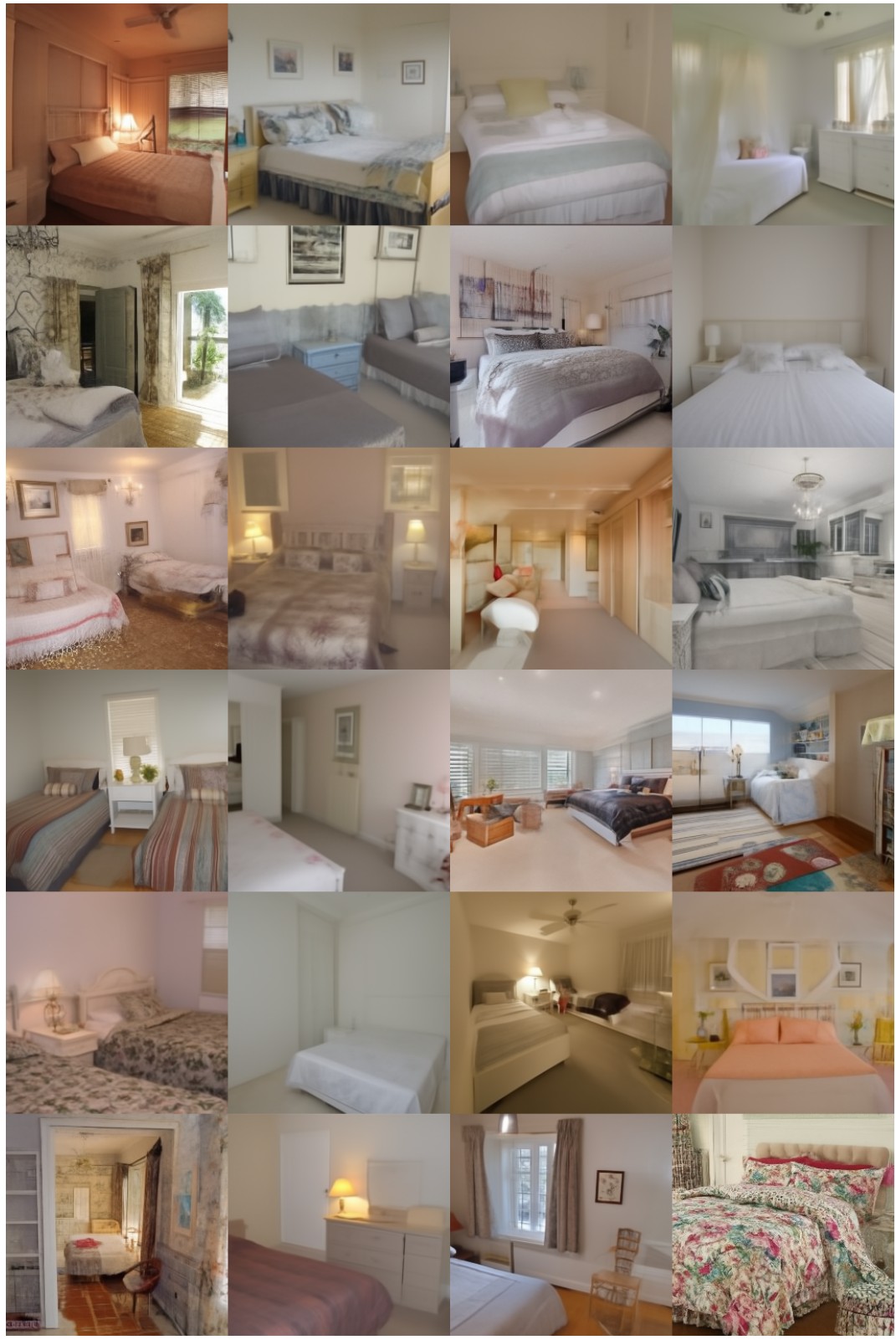

Figure 17: Generated samples of LSUN Bedroom 256×256 using a pixel-space diffusion model at NFE = 5 (FID = 14.02). The training and sampling schedules were optimized using CRS-$v_x$ and CRS-$v_x + v_{\text{FID}}$, respectively. DPM-Solver++(2M) was used as the sampler.

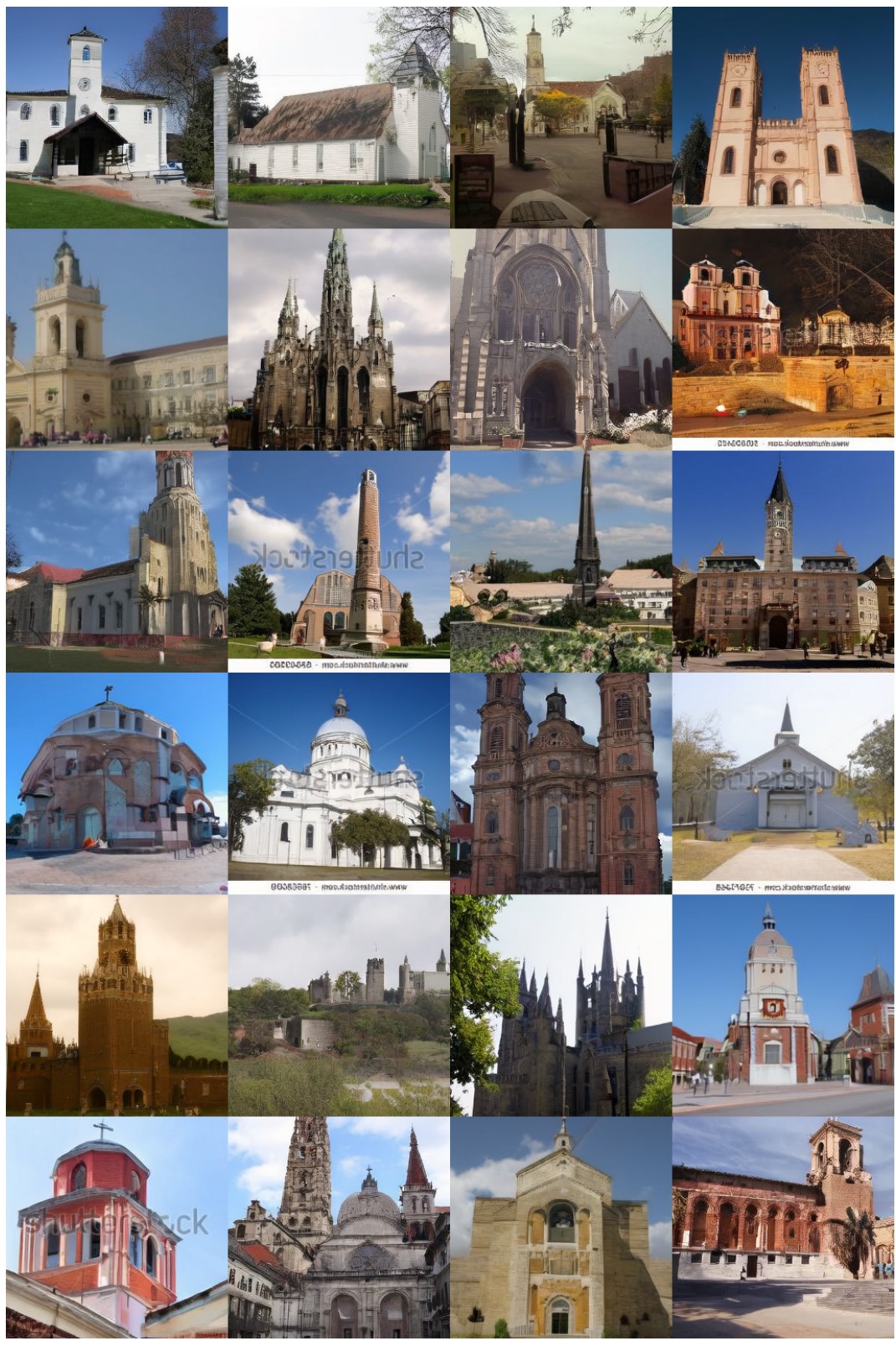

Figure 18: Generated samples of LSUN Church 256×256 using a latent-space diffusion model at NFE = 20 (FID = 3.89). The training and sampling schedules were optimized using CRS-$v_x$ and CRS-$v_\varepsilon$, respectively. DPM-Solver++(2M) was used as the sampler.

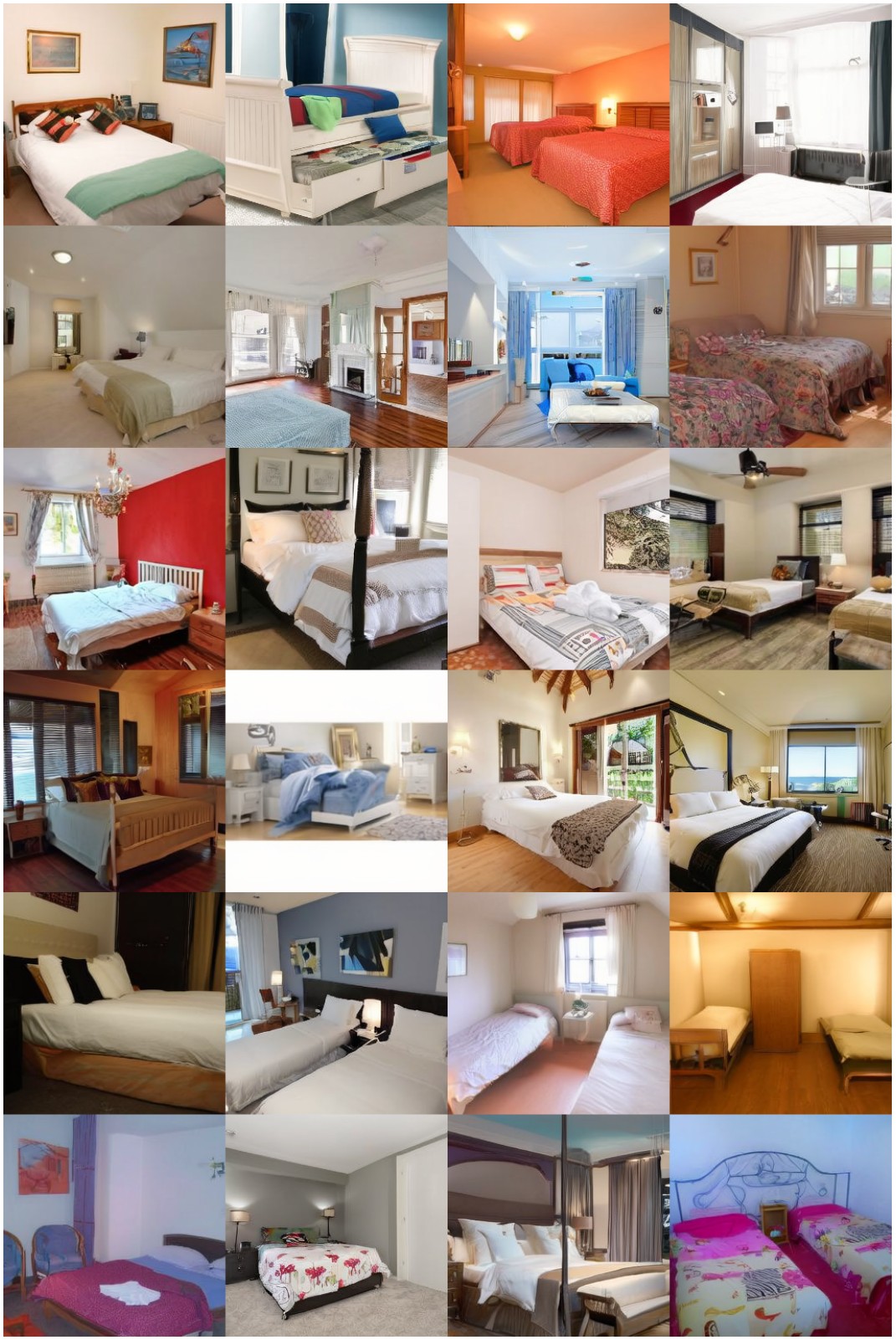

Figure 19: Generated samples of LSUN Bedroom 256×256 using a latent-space diffusion model at NFE = 20 (FID = 2.71). The training and sampling schedules were optimized using CRS-$v_x$ and CRS-$v_\varepsilon$, respectively. PNDM was used as the sampler.

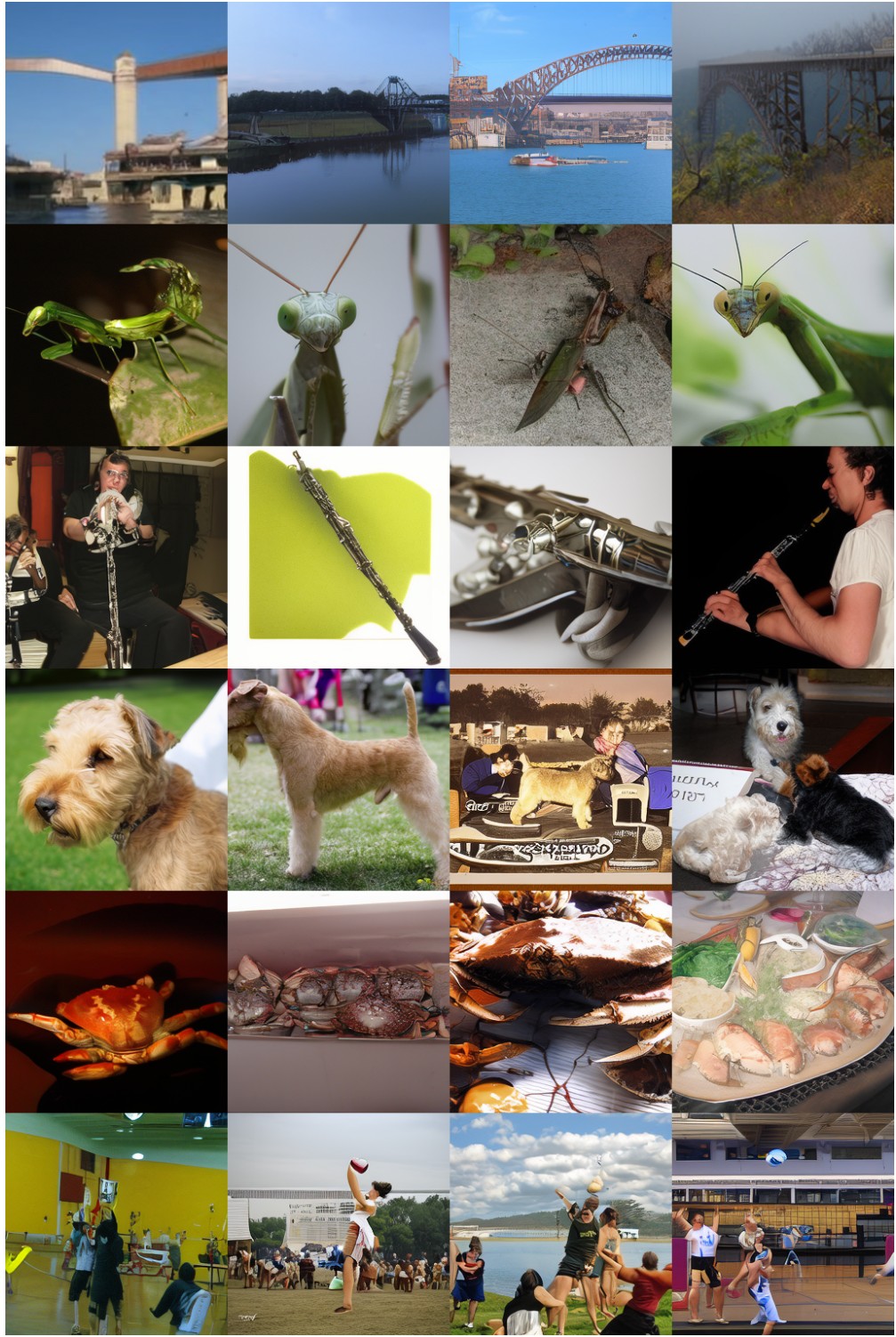

Figure 20: Generated samples of ImageNet 256×256 using a latent-space diffusion model at NFE = 20 with class condition (FID = 10.83). Six classes were randomly selected, and each row corresponds to one class. The training and sampling schedules were optimized using CRS-$v_x$ and CRS-$v_\varepsilon$, respectively. SDE-DPM-Solver++(2M) was used as the sampler.

