# OpenReview forum: "Constant Rate Scheduling: A General Framework for Optimizing Diffusion Noise Schedule via Distributional Change"
_TMLR — Accepted by TMLR_

### Review · Reviewer_DmXo · 2025-11-05

**Summary Of Contributions:**

The paper proposes Constant Rate Scheduling (CRS), a framework for designing both training and sampling noise schedules so that the rate of distributional change between adjacent timesteps is (approximately) constant. The authors minimize the discrepancy $D(t, t + \Delta t)$ between diffused data distributions such that the final schedule $ -d\alpha(t)/dt = C v(\alpha)^{-\xi}$  with $\xi>0$. The authors demonstrated several ways to instantiate $v(\alpha)$ (using vFID, vx, and v$\epsilon$). The authors applied the framework to LSUN horses/church, and they claim fairly consistent gains.

Strengths:
1. The work's general direction is highly relevant to diffusion models and optimizing trajectories.
2. The proposed recipe is easy to follow and is unified in training and in inference. The authors adapted multiple complementary discrepancy measures which can be used in conjunction.
3. The empirical ablation studies are generally extensive (with a few catches) and show general improvements over baselines.

Weakness:
1. I think the method may not be broadly applicable (vFID is very expensive, and it appears the method is highly sensitive to hyperparameters)
2. The paper does not justify (from a theoretical perspective) why such distributional changes relate to sample quality.
3. Some of the baseline comparisons do not strictly match the previously published results.

**Audience:**

Yes

**Audience Explanation:**

There is a significant number of TMLR readers who work on diffusion models, schedulers, and efficient sampling/training, and I expect such an audience to have an interest in this paper. Generally, this paper is well-positioned within the community's ongoing effort to optimize denoising trajectories, with interesting empirical results and some potential for other practitioners to try it out.

**Broader Impact Concerns:**

None.

**Claims And Evidence:**

Yes

**Claims Explanation:**

The core claim is that $ -d\alpha(t)/dt = C v(\alpha)^{-\xi}$ improves training/sampling by allocating steps where distributional change is the fastest. This is reasonably convincing from both logical and empirical perspectives. Nevertheless, I would expect more rigorous justifications (e.g. one could argue that different regions have different importance, where high noise is for structure formation and low noise is for finer details, which would warrant very different rates). Additionally, I think there can be more/better apple-to-apple comparisons with AYS/GITS/LDS as the existing results appear heavily hand-tuned.

**Requested Changes:**

Critical changes:
1. The paper requires more comparisons with schedule-optimization baselines under similar training settings and across more tasks. At least one more baseline and one more task should be shown (e.g. AYS, GITS, and potentially LD3 (ICLR'25) with either CIFAR10 or FFHQ). I would also be interested in seeing more empirical examples compared against baselines.
2. The readers will require a better explanation on how to best choose the hyperparameters. Generally, I believe the tuning budgets should be standardized across baselines.
3. There are a series of minor mistakes, e.g. incorrect citations to Feller 2015 (retracted chapter?), eq. 7 has a small approximation error in the first-order Taylor expansion, especially at low NFEs.

Important but not critical:
1. Theoretical analysis beyond simple intuition will greatly strengthen the paper. I would be interested in seeing a formal connection between distributional change and sample quality.
2. I would appreciate additional discussion on scalability or broader applicability of the method, especially given so many knobs to tune.

---

### Review · Reviewer_T49E · 2025-11-09

**Summary Of Contributions:**

The paper proposes a general framework called Constant Rate Scheduling (CRS) for optimizing the noise schedules in diffusion models, applicable to both training and sampling. The core idea is to design a schedule that enforces a constant rate of change in the probability distribution of the diffused data throughout the diffusion process. This rate of change is quantified using a flexible, user-defined discrepancy measure.

The authors introduce three practical discrepancy measures: one based on FID ($v_{FID}$), one on data prediction ($v_x$), and one on noise prediction ($v_\epsilon$). These measures can also be combined to leverage their complementary strengths.

**Audience:**

Yes

**Audience Explanation:**

The paper addresses the optimization of noise schedules for diffusion models , which is an active, fundamental, and important area of research within the machine learning community. The TMLR audience is centrally concerned with such foundational improvements to state-of-the-art generative models.

The proposed Constant Rate Scheduling (CRS) framework is a novel contribution that provides a general and unified method for optimizing both training and sampling schedules. This unification is a conceptual step beyond prior work, which often focused on optimizing only one of the two.

The paper provides extensive empirical evidence that this framework is not just a theoretical curiosity but delivers practical, consistent performance gains across a wide variety of settings

**Broader Impact Concerns:**

None. This paper presents a foundational algorithmic improvement for generative models. This work does not introduce any new, specific ethical risks

**Claims And Evidence:**

Yes

**Claims Explanation:**

The primary claim—that optimizing noise schedules via the CRS framework leads to improved performance (e.g., FID, sFID, Precision/Recall) compared to conventional schedules—is convincingly supported by evidence. The authors provide extensive results across pixel-space and latent-space models, multiple datasets, and a wide range of samplers and NFE settings (from 5 to 250). The ablations on different discrepancy measures (Tables 1, 3, etc.) and their combinations (e.g., $CRS-v_{x}+v_{FID}$) effectively demonstrate the framework's utility.

**Requested Changes:**

1. Address the Discrepancy in the $\Delta t$ Assumption: The derivation of the CRS condition in Eq. (7-8) is based on a first-order approximation that assumes $\Delta t$ is very small. However, the method shows strong performance in low-NFE regimes (e.g., NFE=5 or 10), where $\Delta t$ is large, and the approximation error should be significant. This is a key theoretical gap. The authors must add a discussion addressing this point. Why does the framework, derived from a continuous-time perspective, still hold in the highly discrete, few-step sampling setting?

2. Expand Experimental Scope to Justify "Generality": The paper claims to be a "general framework", but its generality is not sufficiently proven. The validation is limited to $\epsilon$-prediction models with U-Net backbones. To better support this central claim, we request the authors.
  a. Investigate Classifier-Free Guidance (CFG): The class-conditional experiments (e.g., on ImageNet ) presumably use CFG, but this is not mentioned. The interaction between noise schedule and guidance strength is a critical factor in practice. The authors should at least conduct a study showing how the optimal schedule and its performance gains are affected by varying the CFG scale
  b. Test on Other Parameterizations/Architectures: To truly demonstrate generality, the framework should be validated on other modern setups, such as $v$-prediction and Diffusion Transformer (DiT) architectures.

3. Analyze Training Convergence Speed: The motivation suggests that by making the distribution change constant, CRS provides more reliable local updates, which implies an "easier" learning problem. A natural hypothesis from this is that models trained with CRS (e.g., $CRS-v_x$) might converge faster than those trained with standard schedules (like linear or VDM++). It would be a strong addition to the paper to include a plot of FID or loss curves over training steps/epochs to confirm if this is the case.

4. Clarify CIFAR10 Results: The FID of 1.89 reported for CIFAR10 in Table 29  (NFE=50, DPM-Solver++) seems to underperform the original EDM paper, which reported an FID of 1.79

5. Mathematical Phrasing (Eq. 7): Please refine the language around Eq. (7). As written, it implies Eq. (7) is the solution to Eq. (6). More accurately, Eq. (7) (i.e., $D(t, t+\Delta t) = const.$) is the condition that the optimal schedule must satisfy to solve the min-max problem in Eq. (6)

6. Autoencoder Model (Sec 5.3): The paper states a "pre-trained $VQ-4$ autoencoder from LDMs" was used. This is incorrect; the LDM paper used a KL-regularized autoencoder (not VQ). Please correct this to KL-f4 (or the appropriate term). As a follow-up, please add a brief justification for using an f=4 autoencoder ($64 \times 64$ latent) for $256 \times 256$ images, as f=8 ($32 \times 32$ latent) is also a very common choice.

7. ImageNet Performance: The FID scores on ImageNet seem relatively high for a 500-epoch training run with this model size. This does not invalidate the relative improvement shown by CRS, but it would be good to briefly acknowledge if the baseline models are not fully converged and how this might affect the conclusions.

Missing References
- Scaling Rectified Flow Transformers for High-Resolution Image Synthesis
- On the Importance of Noise Scheduling for Diffusion Models
- Flow Matching for Generative Modeling
- Improved Noise Schedule for Diffusion Training

---

### Review · Reviewer_CJ5W · 2025-11-19

**Summary Of Contributions:**

This paper studies the sampling efficiency of diffusion models without sacrificing image quality or mode coverage. The authors propose Constant Rate Scheduling (CRS), a method to optimise the noise schedule (the variance added at each diffusion step) so that the probability distribution of the data undergoing diffusion changes at a constant rate throughout the process. The paper presents theoretical motivations for the proposed method. Through extensive experiments on multiple datasets (LSUN, ImageNet, FFHQ) and both pixel-space and latent-space diffusion models, the paper finds that CRS consistently improves generation performance (lower FID, higher precision/recall), demonstrating the approach’s effectiveness.

**Audience:**

Yes

**Audience Explanation:**

1. This paper studies diffusion models which are central in machine learning.

2. It addresses a real, widely felt pain point in diffusion models.

3. The paper presents conceptual/theoretical interest, not just experiments. Also, the experimental results are impressive.

**Claims And Evidence:**

Yes

**Claims Explanation:**

1. The paper presents strong evidence on performance improvement. This is supported by extensive results on a variety of datasets and settings. We can see consistency across multiple samplers (Stochastic DDIM, SDE-DPM-Solver++(2M), DDIM, PNDM, DPM-Solver++(2M), UniPC), and the paper reports precision/recall alongside FID rather than cherry-picking a single metric.

2. The paper claims that CRS helps both training and sampling, which is well supported in both latent models (LSUN Church, Bedroom, ImageNet), and pixel models (LSUN Horse, Bedroom).

3. The paper claims about generality and arbitrary distance metrics, which is a bit complecated. No non-image domains were tested, and only three metrics were tried. So the “any distance metric” / “expected to work beyond images” parts are unvalidated; however, the authors relegate this to future work in the Limitations section.

4. The code is not submitted / released. I encourage the author to expand on this in their response.

**Requested Changes:**

1. Reproducibility and hyperparameter specification for CRS. CRS introduces several hyperparameters, some of which are scattered in the appendix and/or briefly mentioned but not fully specified per experiment. Please include:

- How v(α) was computed (T, number of samples, feature network used).
- Exact ξ, w_x, w_FID, any clipping/thresholds, EMA parameters.
- Dataset-specific tuning rules (e.g., different R or ξ for ImageNet).

2. Computational overhead. The paper mentions that computing v_FID and v_x adds offline and/or training-time cost, but the discussion is fairly high-level. So, please add a short subsection (or expand the current discussion) that reports approximate compute overhead for:
- v_FID estimation (per dataset, relative to one training epoch).
- v_x estimation during training (e.g., % increase in training time).

3. Discuss choice of distance metric. CRS heavily relies on the metric it uses. It would be great to discuss how to find or learn better metrics/embeddings that can capture distribution changes of diffused data. The authors suggest using self-supervised feature learning on noised data as one idea; please expand on this.

4. CRS does not guarantee an optimal final model in theory – it’s a heuristic guided by distribution change, not a direct optimisation of the model’s loss or sample quality metric. While it correlates well with better performance (as seen empirically), there’s no formal proof.

---

> ### Author Response · Authors · 2025-11-22
> **Official Comment by Authors**
>
> We thank the reviewer for the thoughtful and constructive feedback.
> We appreciate the opportunity to clarify several points and improve the clarity and reproducibility of the manuscript.
> Below we address each requested change.
>
> ### **Requested change 1: Reproducibility**
>
> As the reviewer pointed out, the current manuscript contains the key implementation details in Appendix C and D,
> but the main text does not provide sufficient cross-references, which may make it difficult for readers attempting to reproduce the results.
>
> In the revised manuscript, we will make the following changes:
> - Computation of $v(\alpha)$: We will add a consolidated table summarizing all hyperparameters ($T$, number of samples, feature network, clipping/EMA settings) for ease of reference.
> - Hyperparameters used in all CRS-$v_{x}+v_{\mathrm{FID}}$ experiments: We will explicitly refer to Table 7 in Appendix D from Section 4.4 .
> - Dataset-specific tuning rules for CRS-$v_{x}+v_{\mathrm{FID}}$: We will explicitly refer to Appendix D from Section 4.4.
>
> Regarding the reviewer’s mention of the hyperparameter $R$, we would like to clarify that CRS does not include any parameter named $R$.
> We believe this may refer to a parameter used in other schedule-optimization works; in CRS, all dataset-specific tuning is performed through $w_{x}$, $w_{\mathrm{FID}}$, $\xi_{x}$, and $\xi_{\mathrm{FID}}$.
> We will state this explicitly in the revision to avoid confusion.
>
> ### **Requested change 2: Computational overhead**
>
> We thank the reviewer for pointing out that the current discussion on computational overhead is too high-level.
> In the revised manuscript, we will add a new subsection to Section 5 summarizing the computational cost (in terms of wall-clock time) of $v_{\mathrm{FID}}$ and $v_{x}$.
>
> **(1) Cost of computing $v_{\mathrm{FID}}$**
>
> We will report both the cost relative to one training epoch and the cost relative to the total training time.
> The results show that:
> - The cost relative to one epoch is nearly identical across all datasets for both pixel-space and latent-space diffusion models, because the time for computing $v_{\mathrm{FID}}$ and the time for one training epoch both scale proportionally with the number of training images.
> - In latent-space diffusion models, the cost relative to one training epoch appears larger than pixel-space diffusion models. This is because latent-space diffusion models require less time to train for one epoch than pixel-space diffusion models.
> - Even in the worst case, the overhead remains below 15% of the total training time.
>
> | Diffusion type | Dataset | Cost relative to one epoch | Cost relative to total training |
> | - | - | - | - |
> | Pixel space | LSUN Horse | ~5 epochs | ~10% |
> | | LSUN Bedroom | ~5 epochs | ~10% |
> | Latent space | LSUN Church | ~14 epochs | ~1.4% |
> | | LSUN Bedroom | ~14 epochs | ~14% |
> | | ImageNet | ~14 epochs | ~2.8% |
>
> **(2) Overhead of computing $v_{x}$ during training**
>
> We will also expand the discussion of the overhead of computing $v_{x}$.
> The table below summarizes the total training time when using Linear and CRS-$v_{x}$ as the training schedules.
> As CRS-$v_{x}$ requires evaluating $\bar{D_{x}^{2}}(\alpha, \alpha')$ during training, the total training time increases by 20-30% compared to using the linear schedule.
>
> | Diffusion type | Dataset | Linear | CRS-vx | Increase |
> | - | - | - | - | - |
> | Pixel space | LSUN Horse | 170 hours | 215 hours | +26.5% |
> | Latent space | LSUN Church | 85 hours | 105 hours | +23.5% |
> | | LSUN Bedroom | 190 hours | 245 hours | 28.9% |
> | | ImageNet | 430 hours | 525 hours | 22.1 % |
>
> We will summarize these results in Section 5, noting that CRS-$v_{x}$ often leads to faster FID convergence, which can largely compensate for the modest per-epoch overhead.

---

> > ### Comment · Reviewer_CJ5W · 2025-12-17
> > **Code**
> >
> > Thanks for your clarification. Most of my concerns have been cleared, except the one about the code. Will the authors release their code?

---

> ### Author Response · Authors · 2025-11-22
> **Official Comment by Authors**
>
> ### **Requested change 3: Discussion on discrepancy measures**
>
> We appreciate the reviewer’s suggestion to further discuss how to design or learn better discrepancy measures for CRS.
> We agree that the choice of discrepancy measure is central to the effectiveness of our framework.
>
> A promising direction is to build on advances in schedule-optimization theory, such as AYS.
> As shown in Appendix B, CRS-$v_{x}$ uses a simplified variant of the KLUB objective introduced in AYS as its discrepancy measure.
> Thus, future theoretical developments in schedule optimization may naturally yield new discrepancy measures applicable within the CRS framework.
>
> Although our experiments primarily relied on FID, many feature-based perceptual metrics have been proposed.
> LPIPS, for example, is a strong candidate for capturing visually meaningful differences.
> Because metrics such as FID and LPIPS depend on feature extractors, an important direction for CRS is to develop feature representations that remain robust and informative for noised data, which is crucial for accurately assessing distributional change.
>
> In this regard, self-supervised learning on noised samples is a promising possibility, as it requires no labels and can be designed to be noise-level aware (e.g., by predicting or conditioning on $\alpha$).
> Such learned embeddings may lead to more expressive and domain-adaptive discrepancy measures.
>
> We will add this discussion in the revised manuscript.
>
> ### **Requested change 4: Theoretical analysis**
>
> We agree that CRS does not provide any formal optimality guarantee.
> As noted in our Limitations section, establishing such guarantees is beyond the scope of this work, and we do not claim them in the paper.
>
> Our contribution is to offer a theoretically motivated and empirically effective framework that can incorporate a wide range of discrepancy measures.
> We agree that developing formal connections between discrepancy equalization and optimal sample quality is an important direction for future work, and we will make this point more explicit in the revision.
>
> Thank you again for the constructive feedback.
> We believe the proposed revisions will significantly strengthen the clarity, reproducibility, and positioning of our work.

---

> ### Author Response · Authors · 2025-12-18
> **Official Comment by Authors**
>
> Thank you for the question.
>
> Yes, we plan to release the code.
> While some additional internal steps and code cleanup are required, we will make the implementation publicly available upon acceptance of the paper, within a reasonable timeframe.
>
> The released code will include the implementation of CRS, including the computation of v(α), as well as training and sampling scripts necessary to reproduce the main experimental results.

---

### Author Response · Authors · 2025-11-25
**Official Comment by Authors**

We sincerely thank the reviewers and the Action Editor for their careful reading of our manuscript and for the constructive and insightful comments.
These comments have greatly helped us improve the clarity, rigor, and overall quality of the paper.
We have uploaded the revised version, with all changes highlighted in blue.

We summarize the revisions in the table below.
The following abbreviations are used in the table:
- CC: items marked as critical under Request Changes
- IC: items marked as important but not critical under Request Changes
- RC: general Request Changes items
- MR: missing references

| Reviewer | Comment | Revised Sections | Summary of Changes |
| - | - | - | - |
| DmXo | CC1 | Section 5.4 | Included comparisons against AYS, GITS, and LD3 on CIFAR-10 and FFHQ. |
| | CC2 | Section 4.4 | Inserted a pointer directing readers to Appendix D for the hyperparameter tuning procedure. |
| | | | Added a note that the default hyperparameter setting $w_{x} = w_{\mathrm{FID}} = 0.5$ is effective for CRS-$v_{x}+v_{\mathrm{FID}}$. |
| | | Section 5.4 | Clarified that although CRS takes longer to optimize sampling schedules than GITS and LD3, the schedule optimization is performed only once and remains insignificant relative to training time. |
| | CC3 | References | Modified the reference to Feller. |
| | | Section 4.2 | Clarified that the approximation error in Eq. (7) becomes significant when optimizing the sampling schedule at small NFEs, and noted that tuning $\xi$ can partially alleviate this effect. |
| | | Table 27 | Added an explanation that adjusting ξ can partially mitigate the impact of the approximation error. |
| | IC1 | Section 6 (1) | Stated explicitly that theoretical guarantees are beyond the scope of the current paper and constitute an important avenue for future work. |
| | IC2 | Section 6 (6) | Added a discussion on scalability with respect to dataset size and image resolution. |
| T49E | RC1 | Section 4.2 | Clarified that the approximation error in Eq. (7) becomes significant when optimizing the sampling schedule at small NFEs, and noted that tuning $\xi$ can partially alleviate this effect. |
| | | Table 27 | Added an explanation that adjusting ξ can partially mitigate the impact of the approximation error. |
| | RC2 | Section 5.4 | Clarified that EDM is based on the x-prediction parametrization. |
| | | Section 6 (5) | Added a discussion on the generality of CRS across prediction parameterizations and model architectures to clarify the current coverage of CRS. |
| | | Section 6 (2) | Included a note outlining future research directions concerning Classifier-Free Guidance (CFG). |
| | RC3 | Appendix F | Included additional experiments evaluating training convergence speed, presented in Figures 12 and 13. |
| | RC5 | Section 4.2 | Improved the explanation of Eq. (7) to enhance clarity and accuracy. |
| | RC6 | Section 5.3 | Revised the description of the autoencoder used in the latent-space diffusion models. |
| | RC7 | Appendix F.2 | Emphasized that the results correspond to class-conditional generation without CFG and noted that the reported FID scores are within a reasonable range compared to prior work. |
| | MR | Section 1 and 2 | Added the references listed under Missing References. |
| CJ5W | RC1 | Appendix C | Summarized the hyperparameters used for computing $v(\alpha)$ and the values used in the experiments in Tables 8, 9, and 10. |
| | | Section 5.1 | Added explanations on the hyperparameter values used for CRS and the tuning procedure for CRS-$v_{x}+v_{\mathrm{FID}}$. |
| | RC2 | Section 5.6 | Added a subsection describing the computational overhead of CRS. |
| | RC3 | Section 6 (4) | Discussed future research directions for improving the discrepancy measure. |
| | RC4 | Section 6 (1) | Stated explicitly that theoretical guarantees are beyond the scope of the current paper and constitute an important avenue for future work. |

---

### Decision · Action_Editor_ujHj · 2026-01-07

**Recommendation:** Accept as is

**Audience:**

Yes

**Audience Explanation:**

The paper is suggesting a new way to schedule the noise in diffusion models, used both in training and sampling from these models. This is one of the most key hyperparameters in a widely used model, so it will be of interest to a significant portion of TMLR's audience.

**Claims And Evidence:**

Yes

**Claims Explanation:**

The main claims pertain to empirical performance, which are well supported in the experiments.